# Multi-site calibration and validation of SWAT with satellite-based evapotranspiration in a data-sparse catchment in southwestern Nigeria

Abolanle E. Odusanya[1], Bano Mehdi[1, 2], Christoph Schürz[1], Adebayo O. Oke[3], Olufiropo S. Awokola[4], Julius A. Awomeso[5], Joseph O. Adejuwon[5] and Karsten Schulz[1]

[1]Institute of Hydrology and Water Management, University of Natural Resources and Life Sciences, Vienna (BOKU), 1190 Vienna, Austria
[2]Department of Crop Sciences, University of Natural Resources and Life Sciences, Vienna (BOKU), 3430 Tulln, Austria
[3]Institute of Agricultural Research and Training, Land and Water Resources Management Programme, Obafemi Awolowo University P.M.B 5029, Moor Plantation, Ibadan, Nigeria
[4]Department of Civil Engineering, College of Engineering, University of Agriculture, P.M.B. 2240, Abeokuta, Nigeria
[5]Department of Water Resources Management and Agrometeorology, College of Environmental Resources Management, University of Agriculture, P.M.B.2240, Abeokuta, Nigeria

*Correspondence to*: Bano Mehdi (bano.mehdi@boku.ac.at)

**Abstract.** The main objective of this study was to calibrate and validate the eco-hydrological model Soil and Water Assessment Tool (SWAT) with satellite-based actual evapotranspiration (AET) data from the Global Land Evaporation Amsterdam Model (GLEAM_v3.0a) and from the Moderate Resolution Imaging Spectroradiometer Global Evaporation (MOD16) for the Ogun River Basin (20 292 km$^2$) located in southwestern Nigeria. Three potential evapotranspiration (PET) equations (Hargreaves, Priestley-Taylor and Penman-Monteith) were used for the SWAT simulation of AET. The reference simulations were the three AET variables simulated with SWAT before model calibration took place. The Sequential Uncertainty Fitting technique (SUFI-2) was used for the SWAT model sensitivity analysis, calibration, validation, and uncertainty analysis. The GLEAM_v3.0a and MOD16 products were subsequently used to calibrate the three SWAT simulated AET variables, thereby obtaining six calibrations/validations at a monthly time scale. The model performance for the three SWAT model runs was evaluated for each of the 53 subbasins against the GLEAM_v3.0a and MOD16 products, which enabled the best model run with the highest performing satellite-based AET product to be chosen. A verification of the simulated AET variable was carried out by: (i) comparing the simulated AET of the calibrated model to GLEAM_v3.0b AET, this is a product that has a different forcing data to version of GLEAM used for the calibration, and (ii) assessing the long-term average annual and average monthly water balances at the outlet of the watershed. Overall, the SWAT model composed of Hargreaves PET equation and calibrated using the GLEAM_v3.0a data (GS1) performed well for the simulation of AET and provided a good level of confidence for using the SWAT model as a decision support tool. The 95% uncertainty of the SWAT simulated variable bracketed most of the satellite based AET data in each subbasin. A validation of the simulated soil moisture dynamics for GS1

was carried out using satellite retrieved soil moisture data, which revealed good agreement. The SWAT model (GS1) also captured the seasonal variability of the water balance components at the outlet of the watershed.

This study demonstrated the potential to use remotely sensed evapotranspiration data for hydrological model calibration and validation in a sparsely gauged large river basin with reasonable accuracy. The novelty of the study is the use of these freely available satellite derived AET datasets to effectively calibrate and validate an eco-hydrological model for a data-scarce catchment.

## 1. Introduction

Hydrological modelling in data sparse catchments has always been a challenging task due to lack of ground observations, and insufficient or poor-quality data. Data scarcity is the main limitation in tropical regions for setting up hydrological models for watershed simulations, which could be used as significant decision support tools for sustainable water resources management. Water resources globally are becoming increasingly vulnerable as a result of escalating water demand arising from population growth, expanding industrialisation, increased food production and pollution due to various anthropogenic activities, climate and land use change impacts (Carroll et al., 2013; McDonald et al., 2014; Goonetilleke et al., 2016). The situation is more evident and critical in many developing countries where no water resources monitoring plans or water management strategies are in place for the future. Like many developing countries, Nigeria cannot satisfy its domestic water needs as only 47% of the total population have access to water from improved sources (Ishaku et al., 2012).

The Ogun River is the main source of public water supply for the people living in the States of Lagos and Ogun in southwestern Nigeria. The prevalent situation of insufficient hydrological data associated with lack of up to date streamflow data (Sobowale and Oyedepo, 2013) and the poor level of data quality in this watershed can be attributed to a gradual decline in hydrological stations number and their management. Water management planners are facing considerable uncertainties in terms of future availability and quality of the water resource. Therefore, a clear understanding of the on-going challenges and innovative management approaches are needed. One of the many ways to tackle this task is by using hydrological models as tools coupled with the use of increasingly available global and regional datasets to run the models.

Numerous physically based distributed (PBD), continuous models that aim to describe which driving processes are present in a system and are able to make detailed predictions in both time and space are available to simulate water quantity and quality variables and these include, among others: the Soil and Water Assessment Tool (SWAT; Arnold et al., 1998), which is able to represent detailed agricultural management practices and simulate water quantity and quality variables; or the Hydrologic Simulation Program Fortran (HSPF; Bicknell et al., 1997) that is used in predicting hydrology with in-stream nutrient transport processes; or SHETRAN (Ewen et al., 2000), which has capabilities for modelling subsurface flow and transport. These PBD models attempt to explain hydrological phenomena through their underlying physical mechanisms, and explicitly represent (through mathematical equations) the biological, chemical and physical processes of a basin.

Schuol et al. (2008) have successfully applied the hydrological model SWAT to quantify the freshwater availability for the whole of Africa at a detailed subbasin level and on a monthly time scale. Using the SUFI-2 (Sequential Uncertainty Fitting Algorithm) program with three different objective functions, the model was calibrated and validated at 207 discharge stations. They reported the models' inability to simulate runoff adequately in some areas in the East and South Africa, but also reported that the model results were quite satisfactory for such a large-scale application although containing large prediction uncertainties in some areas. Many of the limitations reported within this continental modelling study in Africa were data related. Abaho et al. (2009) applied an uncalibrated SWAT model to evaluate the impacts of climate change on river flows and groundwater recharge in Sezibwa catchment, Uganda. They observed a 40% increase in groundwater recharge for the period of 2070-2100 and a 47% increase in average river flow. However, there are high levels of uncertainty associated with the model predictions since the model was not calibrated due to insufficient data.

In West Africa, the SWAT model has been widely applied to different river basins with satisfactory results. For example, Schuol and Abbaspour (2006) applied SWAT to model a $4 \times 10^6$ km$^2$ area; mainly the basins of the Niger, Volta and Senegal, addressing calibration and uncertainty issues. Measured river discharges at 64 stations to which many of these stations available data doesn't cover the whole simulation period were used for annual and monthly calibration using SUFI-2 algorithm. Although the results obtained are preliminary with basis for discussion of further improvement, Schuol and Abbaspour (2006) reported that the annual and monthly simulations with the calibrated SWAT model for West Africa showed promising results for the freshwater quantification despite the modelling shortcomings of lack of dams management operation long-term dataset. They also pointed out the importance of evaluating the conceptual model uncertainty as well as the parameter uncertainty. Laurent and Ruelland (2010) successfully calibrated SWAT for the Bani catchment ($1 \times 10^6$ km$^2$) in Mali, a major tributary of the upper Niger River. The calibration and validation results were satisfactory at the catchment outlet and also in various gauging stations located in tributaries. They showed the model performance by reporting discharge and biomass calibration results but did not assess the model prediction uncertainty.

In northwestern Nigeria, Xie et al. (2010) evaluated the SWAT model performance in a large watershed (30 300 km$^2$). Due to the short data period available, all the data obtained were used for calibration. In their study, the model parameters were first optimized with a genetic algorithm, and the uncertainty in the calibration was further analysed using the generalized likelihood uncertainty estimation (GLUE) method; the study presented a reasonably good calibrated model performance without validation. Adeogun et al. (2014) successfully calibrated and validated the SWAT model for the prediction of streamflow at the upstream watershed of Jebba reservoir (area: 12 992 km$^2$) located in north central Nigeria. The model results obtained were good with a Nash-Sutcliffe Efficiency (NSE) of 0.72 and coefficient of determination ($R^2$) of 0.76 for the calibration period, and for the validation period, an $R^2$ of 0.71 and NSE of 0.78 for monthly average streamflow, but the model prediction uncertainty was not quantified.

The findings from these past studies call for continued improvement in the hydrological model performances in Africa, especially in data-sparse regions. One solution is to use freely available global datasets to improve the model performance.

In the context of large-scale hydrological model simulation in data scarce areas, Lopez Lopez et al. (2017) investigated alternative ways to calibrate the large-scale hydrological model PCRaster GLOBAL Water Balance (PCR-GLOBWB) using satellite-based evapotranspiration (GLEAM) and surface soil moisture (ESA CCI) for the data poor catchments Oum er Rbia in Morocco with the aim to improve discharge estimates. In their study, different calibration scenarios are inter-compared. The

results show that GLEAM evapotranspiration and ESA CCI soil moisture used for model calibration resulted in reasonable discharge estimates (NSE ranges from -0.22 to 0.68 and -0.31 to 0.66, respectively). Although better model performance was achieved when the model was calibrated with in-situ streamflow observations resulting in NSE values from -0.15 to 0.75. Their results showed the possibility of using globally available Earth observation datasets in large scale hydrological models to estimate discharge at a river basin scale. Abera et al. (2017) developed a methodology that can improve the state of the art

by using available, but sparse, hydrometeorological data and satellite products to obtain the estimates of all the components of the hydrological cycle (precipitation, evapotranspiration, discharge, and storage) in the Upper Blue Nile Basin. To obtain a water-budget closure, Abera et al. (2017) used the JGrass-NewAge hydrological model calibrated with observed discharge (1994-1999) using particle swarm optimization. The simulation of each hydrological component by JGrass-NewAge was verified using available in-situ and remote sensing data. GLEAM (Miralles et al., 2011a) and MOD 16 AET were used as

independent data sets to assess the JGrass-NewAge estimated AET. Overall, the AET simulations showed the correlation and PBIAS obtained between JGrass-NewAge and GLEAM AET had a better agreement (very low bias and acceptable correlation) compared to JGrass-NewAge and MOD 16.

Recently, Ha et al. (2018) used remotely sensed precipitation, actual evapotranspiration (AET) and leaf area index (LAI) from open access data sources to calibrate the SWAT model for the Day Basin, a tributary of the Red River in Vietnam. The

calibration was performed in SWAT-CUP using the Sequential Uncertainty Fitting algorithm (SUFI-2). In this study simulated monthly AET correlations with remote sensing estimates showed an $R^2$ of 0.71. Pomeon et al. (2018) set up a hydrological modelling framework for sparsely gauged catchments in West Africa using SWAT model whilst largely relied on remote sensing and reanalysis inputs. In their study, validation of the model was conducted to further investigate its performance, where simulated actual evapotranspiration, soil moisture, and total water storage were evaluated using remote sensing data.

The validation result reveals good agreement between predictions and the remotely sensed data ($R^2$ calibration: 0.52 and 0.51; $R^2$ validation: 0.63 and 0.61)

Remote sensing technologies offer large scale spatially distributed observations and have opened up new opportunities for calibrating and validating hydrologic models. This advancement enables several global evapotranspiration products to be used. Extensive reviews of earth observation based methods for deriving AET have been carried out by several research groups

(Anderson et al., 2012; Bateni et al., 2013; Li et al., 2013; Savoca et al., 2013; Senay et al., 2013; Nouri et al., 2015; Wang-Erlandsson et al., 2016).

Two global-scale AET products derived from satellite observation have become available and these two AET products were used in this study. The Global Land Evaporation Amsterdam Model (GLEAM, http://www.gleam.eu) and Moderate Resolution Imaging Spectroradiometer Global Evaporation (MOD16). GLEAM is an evapotranspiration product developed

by the VU University of Amsterdam (Miralles et al., 2011a, 2011b) and contains a set of algorithms that separately estimate the different components of terrestrial evaporation (i.e. transpiration, interception loss, bare soil evaporation, snow sublimation and open water evaporation), as well as variables such as the evaporative stress factor, potential evaporation, root-zone soil moisture and surface soil moisture by using satellite-based climatic and environmental observations (Miralles et al., 2011a;

Martens et al., 2017). Recently, the GLEAM_v3.0 AET has been validated against measurements from 64 eddy-covariance towers and 2338 soil moisture sensors across a broad range of ecosystems with varying levels of success (Martens et al., 2017). In this study, GLEAM_v3.0a and v3.0b were used. These two datasets differ only in their forcing variables and spatial-temporal coverage. GLEAM_v3.0a is a dataset spanning the 35-year period 1980-2014 and is based on reanalysis net radiation and air temperature, a combination of gauged-based, reanalysis and satellite-based precipitation and satellite-based vegetation optical

depth. GLEAM_v3.0b is a dataset spanning the 13-year period 2003-2015 and is derived by satellite data only (Miralles et al., 2011a; Martens et al., 2017).

The MOD16 global evapotranspiration data is based on a  1 km$^2$ grid of land surface AET that was developed  with an energy balance model using satellite data as input (Mu et al., 2011).  The MOD16 product estimates evapotranspiration using Moderate Resolution Imaging Spectroradiometer, landcover, albedo, LAI, an Enhanced Vegetation Index (EVI), and a daily

meteorological reanalysis data set from NASA's Global Modelling and Assimilation office (GMAO). The non-satellite input data are NASA's MERRA GMAO (GEOS-5) daily meteorological reanalysis data. MOD 16 has been validated using measurement from eddy covariance station in different tropical sites ( Ruhoff et al., 2013; Ramoelo et al., 2014).  Ruhoff et al. (2013) validated MOD16 AET using ground-based measurements of energy fluxes obtained from eddy covariance sites installed in tropical sites in the Rio Grande basin Brazil. Likewise, Ramoelo et al. (2014) validated MOD16 using data from

two eddy-covariance flux towers installed in a savannah and woodland ecosystem within the Kruger National Park, South Africa.

The objective of our study was to obtain a high performing eco-hydrological model for the Ogun River Basin in southwestern Nigeria that can be used as a decision-support tool. To this effect, the specific objectives were: (i) to calibrate/validate the SWAT model with remotely sensed actual evapotranspiration products; namely the Global Land Evaporation Amsterdam

Model (GLEAM_v3.0a) and the Moderate Resolution Imaging Spectroradiometer Global Evaporation (MOD16), and ii) to use further independent products to validate the simulated soil moisture and verify the simulated water balance components. Although the three PET equations and the corresponding AET simulations from SWAT have been tested for their performance before (Wang et al. (2006); Franco and Bonumá (2017); Samadi (2017); Ha et al. (2018)), calibrating each of the three SWAT simulated AET variables with two remotely-sensed AET products for each delineated subbasin to determine the highest

performing model in a catchment has not been undertaken.

Hence, the contribution of this study include: (i) the calibration/validation of simulated AET from the three SWAT models using satellite derived AET data; (ii) the use of satellite-based AET data for calibration/validation of the SWAT model in each of the SWAT delineated subbasins; and (iii) the validation of simulated soil moisture dynamics of the highest performing

SWAT model run using European Space Agency Climate Change Initiative soil moisture (ESA CCI SM) in each of the SWAT delineated subbasins.

## 2. Materials and methods

### 2.1 Description of the study site

The study area is a sub-watershed (20 292 km$^2$) of the Ogun River Basin (23 700 km$^2$) located in southwestern Nigeria (Fig. 1), bordered geographically by latitudes 7° 7' N and 8° 59' N and longitudes 2° 4' E and 4° 9' E. About 2 % of the catchment area is located in the Benin Republic. The study area encompasses the Sepeteri, Iseyin, Olokemeji, Oyan and Abeokuta catchments and cut across the Oyo and Ogun state administrative boundaries. The Ogun River, which literarily means the River of Medicine, springs from Igaran Hills in Oyo state, near Saki, at an elevation of about 624 m above the mean sea level.

The elevation ranges from 624 m to 23 m. The mean annual rainfall (1984-2012) obtained from measured data in Ogun watershed is 1224 mm yr$^{-1}$ and the mean annual temperature (1984-2012) obtained from measured data is about 27°C. Mean annual potential evapotranspiration (PET) estimated by Hargreaves method (Hargreaves and Samani, 1985) using measured minimum and maximum temperature is 1720 mm yr$^{-1}$ and the mean AET obtained from SWAT output (1989-2012) for this study area is 692 mm yr$^{-1}$. Two seasons are distinguishable in the watershed, a dry season from November to March and a wet

season between April and October. The watershed area is characterized by strong climatic variation and an irregular rainfall (Eruola et al. 2012). The geology of the study area can be described as a rock sequence that starts with Precambrian Basement; which consists of quartzites and biotite schist, hornblende-biotite, granite and gneisses (Bhattacharya and Bolaji, 2010). The major soils of the basin are sandy clayey loam, sandy loam, clayey loam and silt loam. The landuse in the watershed is primarily forest (75 %), cropland (24 %), and urban (1 %).

The basin, in which two large dams (Oyan and Ikere Gorge dams) are located, is of great importance for the economic advancement both at the federal and state level. The dams are the main principal provider of water to Lagos and Ogun States Water Corporation for municipal drinking water production. The Oyan reservoir is located at the confluence of Oyan and Ofiki rivers at an elevation of 43.3 m above mean sea level and was built in 1984, it has a surface area of 40 km$^2$, and a catchment area of $9 \times 10^3$ km$^2$, with a dead storage capacity of $16 \times 10^6$ m$^3$, a gross storage capacity of $270 \times 10^6$ m$^3$, an embankment

crest length of 1044 m, a height of 30.4 m, four spillway gates (each 15 m wide and 7 m high) and three outlet valves (each 1.8 m diameter). The Ikere Gorge is an uncontrolled dam, which started operation in 1991. The dam crosses Ogun River in Iseyin local government area of Oyo state. Ikere Gorge has a capacity of $690 \times 10^6$ m$^3$. The reservoir is adjacent to the Old Oyo National Park, providing recreational facilities for tourists, and the river flows through the park (Oyegoke and Sojobi, 2012). Twenty-five local government areas fall within the study area. In densely populated areas, the Ogun River is used for

bathing, washing and drinking.

Fig.1

## 2.2 SWAT model description

The Soil and Water Assessment Tool (Arnold et al., 1998) is an open source eco-hydrological model developed for the USDA Agricultural Research Services. SWAT is a semi-distributed, process based, continuous model that uses weather, soil, topography and landuse for hydrologic modelling of a basin and runs at a daily time step. It was developed to predict the impact of agricultural land management practices on discharge, sediments, nutrients, bacteria, pesticides and biomass in large complex watersheds with varying soils, land use and management conditions over long periods of time. The SWAT model uses at its core the plant growth model EPIC (Williams et al., 1989) to simulate the growth (including nutrient and water uptake) of many types of crops and trees as land cover. SWAT categorizes plants into seven different types; warm season annual legume, cold season annual legume, perennial legume, warm season annual, cold season annual, perennial and trees. Plant growth is modeled by simulating leaf area development, light interception and conversion of intercepted light into biomass assuming a plant species-specific radiation-use efficiency. Hence, in SWAT, phenological plant development is based on the daily accumulated heat units. The plant growth model is used to assess removal of water and nutrients from the root zone, transpiration, and biomass/yield production.

For modelling purpose in SWAT, the watershed is divided into subbasins which are then further subdivided into hydrologic response units (HRUs) that consist of homogeneous landuse, soil types and slope (Arnold et al., 1998). The soil water balance (WB) is conducted for each HRU and the equation comprises six variables and is estimated in SWAT using the following Eq. (1):

$$SW_t = SW_0 + \sum_{i=1}^{t}(R_{day} - Q_{surf} - E_a - W_{seep} - Q_{gw}) \tag{1}$$

Where $SW_t$ is the final soil water content (mm of water), $SW_o$ is the initial soil water content on day $i$ (mm of water), $t$ is the time (days), $R_{day}$ is the amount of precipitation on day $i$ (mm of water), $Q_{surf}$ the amount of surface runoff on day $i$ (mm of water), $E_a$ the amount of evapotranspiration on day i (mm of water), $W_{seep}$ amount of water entering the vadose zone from the soil profile on day $i$ (mm of water), and $Q_{gw}$ is the amount of return flow on day $i$ (mm of water)

### 2.2.1 Evaporation estimation in SWAT

Evapotranspiration is a key process of the water balance and one of the more difficult components to determine. Although different empirical methods for the estimation of PET are widely adopted, AET is difficult to quantify and it usually requires the reduction of PET through a factor that describes the level of stress experienced by plants. This relationship has been described in detail by several researches (e.g. Morton, 1986; Hobbins et al., 1999; Wang et al., 2006). Numerous methods have been developed to estimate PET (Lu et al., 2005) and SWAT offers three PET estimation options from which the user can choose depending on e.g. the data availability: the Penman-Monteith method (P-M), the Priestley-Taylor method (P-T), or the Hargreaves method (HG). Any one of these three PET equations can be chosen to run in SWAT, but they vary in the amount

of input data required. The Hargreaves method (Hargreaves et al., 1985) is temperature-based and requires only average daily air temperature as input Eq. (2):

$$\lambda E_0 = 0.0023 \times H_0 \times (T_{max} - T_{min})^{0.5} \times (T_{mean} + 17.8) \tag{2}$$

Where $\lambda$ is the latent heat of vaporization (MJ kg$^{-1}$), $E_0$ is the potential evapotranspiration (mm d$^{-1}$), $H_0$ is the extra-terrestrial radiation (MJ m$^{-2}$ d$^{-1}$) , $T_{max}$ is the maximum air temperature for a given day (°C), $T_{min}$ is the maximum air temperature for a given day (°C), $T_{mean}$ is the mean air temperature for a given day (°C).

The Penman-Monteith method (Monteith, 1965; Allen, 1986; Allen et al., 1989) requires air temperature, solar radiation,
relative humidity and wind speed as input Eq. (3):

$$\lambda E = \frac{\Delta \times (H_{net} - G) + \rho_{air} \times C_P \times (e_z^o - e_z)/r_a}{\Delta + \gamma \times (1 + r_c/r_a)} \tag{3}$$

Where $\lambda E$ is the latent heat flux density (MJ m$^{-2}$ d$^{-1}$), $E$ is the depth rate evaporation (mm d$^{-1}$), $\Delta$ is the slope of the saturation
vapor pressure-temperature curve, de/dT (kPa °C$^{-1}$), $H_{net}$ is the net radiation (MJ m$^{-2}$ d$^{-1}$), $G$ is the heat flux density to the ground (MJ m$^{-2}$ d$^{-1}$), $\rho_{air}$ is the air density (kg m$^{-3}$); $C_p$ is the specific heat at constant pressure (MJ kg$^{-1}$ °C$^{-1}$), $e_z^o$ is the saturation vapor pressure of air at height z (kPa), $e_z$ is the water vapor pressure of air at height z (kPa), $\gamma$ is the psychrometric constant (kPa °C$^{-1}$), $r_c$ is the plant canopy resistance (s m$^{-1}$), and $r_a$ is the aerodynamic resistance (s m$^{-1}$).

The Priestley-Taylor equation (Priestley and Taylor, 1972) is a radiation-based method and it provides PET estimates for low advective conditions. The P-T method requires solar radiation, air temperature and relative humidity as input (Eq. 4):

$$\lambda E_o = \alpha_{pet} \times \frac{\Delta}{\Delta + \gamma} \times (H_{net} - G) \tag{4}$$

Where $\alpha_{pet}$ is a coefficient, $\Delta$ is the slope of the saturation vapour pressure-temperature curve, de/dT (kPa °C$^{-1}$), $\gamma$ is the psychometric constant (kPa °C$^{-1}$), $H_{net}$ is the net radiation (MJ m$^{-2}$ d$^{-1}$), and $G$ is the heat flux density to the ground (MJ m$^{-2}$ d$^{-1}$).

Once PET is determined, AET is estimated in SWAT, whereby first, SWAT evaporates any rainfall intercepted by the plant canopy. Second, it calculates the maximum amount of transpiration and sublimation/soil evaporation. Finally, the actual
amount of sublimation and evaporation from the soil surface is calculated. If snow is presented in the HRU, sublimation can occur. When there is no snow (such as this case study), only evaporation from the soil surface is calculated. A complete description of the SWAT  model and the model equations can be found in  Neitsch et al. (2002, 2005) and in Arnold et al. (1998).

**2.3 Model set-up**

The ArcView GIS interface for SWAT2012 (Winchell et al., 2013) was used to configure and parameterize the SWAT model. SWAT model inputs included a 30 m spatial resolution digital elevation model (DEM) with minimum, maximum and mean value of 23 m, 624 m and 289.1 m respectively (Fig 1), 17 soil classes, 17 landuse classes, 3 slope categories, meteorological data and landuse with its management (Table 1).

For the SWAT model set-up, the watershed was delineated into 53 subbasins, with the main outlet in Abeokuta. The minimum and maximum subbasin areas are 72.4 km$^2$ and 853.1 km$^2$ respectively, while the mean is 382.8 km$^2$. Daily precipitation data (1984-2012) and minimum and maximum temperature data (1984-2012) obtained from the Nigerian Meteorological Agency for four weather stations (Fig. 1) were used as observed input data. The weather stations are more or less evenly distributed in or around the watershed, and the weather data obtained from stations located in the same proximity show the same rise and fall dynamics. No orographic effect correction is needed for correcting the precipitation values.

The missing values of daily measured precipitation and minimum and maximum temperatures, were simulated by the WGEN _CFSR _World. The WGEN_CFSR_World weather database is an input into SWAT (ArcSWAT CSFR_World weather generator), containing long-term monthly weather statistics covering the entire globe and developed using the National Centres for Environmental Prediction (NCEP) Climate Forecast System Reanalysis (CFSR) global dataset. The ArcSWAT CSFR_World weather generator was used to simulate daily solar radiation, wind speed, relative humidity and wind speed. The simulated variables were used as input variables into the Penman-Monteith and Priestley-Taylor equations for obtaining the different PET estimates from SWAT.

The topHRU program (Strauch et al., 2017) was used to determine the optimum number of HRUs to use in the watershed. The topHRU program allows the identification of a pareto-optimal threshold which minimizes the spatial error to 0.01 ha for a given number of HRUs and thereby minimizes the trade-off between SWAT computation time and number of HRUs. In this case, topHRU determined the optimum number of HRUs to be 1397 for the Ogun River Basin. Thresholds of 150 ha for soil and 250 ha for slope were used in the SWAT set-up. The physical consequences of the thresholds is to improve the computational efficiency of simulations while keeping key landscape features and information of the watershed in the hydrologic modelling. Not selecting a threshold for landuse was based on our desire to retain all of the landuse classes for future landuse change research needs. The surface runoff in SWAT was estimated using the modified Soil Conservation Society Curve Number method. The SWATfarmR program (Schürz et al., 2017) was used to write the management files in SWAT. All SWAT simulations included a warm-up period of 5 years for the simulation period from 1984 to 2012.

The SWAT model was set-up once for the entire Ogun River Basin and then run three times, where each model run is composed of a different PET equation available in SWAT (HG, P-M or P-T). Figure 2 shows the framework in which the three SWAT model runs (SWAT_HG, SWAT_P-T, and SWAT_P-M) were used to evaluate the model performance by: (i) comparing the three uncalibrated SWAT simulations of AET with the two global AET products (GLEAM and MOD16), thus allowing for

six reference runs of SWAT (RGS1 through RMS6). SWAT_HG represents the SWAT run using the Hargreaves PET equation to simulate uncalibrated AET, these results were compare with the AET from GLEAM_V3.0a (RGS1) and MOD16 (RMS4). SWAT_P-T represents the SWAT run using the Priestley-Taylor PET equation to simulate uncalibrated AET and the results were compared with the AET from GLEAM_V3.0a (RGS2) and MOD16 (RMS5). SWAT_P-M represents the SWAT run using the Penman-Monteith PET equation to simulate uncalibrated AET and the results were compared with GLEAM_v3.0a (RGS3) and MOD16 (RMS6) and, (ii) comparing the calibrations/validations implemented with two global AET products (GLEAM and MOD16), thus allowing for six calibration results of SWAT (GS1 through MS6). SWAT_HG represents the SWAT run using the Hargreaves PET equation to simulate AET and that was calibrated and validated with the AET from GLEAM_v3.0a (GS1) and MOD16 (MS4). SWAT_P-T represents the SWAT run using the Priestley-Taylor PET equation to simulate AET and that was calibrated and validated with the AET from GLEAM_v3.0a (GS2) and MOD16 (MS5). SWAT_P-M represents the SWAT run using the Penman-Monteith PET equation to simulate AET and that was calibrated and validated with the AET from GLEAM_v3.0a (GS3) and MOD16 (MS6). This procedure enabled the SWAT model run with the highest performing simulated AET to be chosen for further use.

Table 1

Fig. 2

### 2.4 Satellite derived datasets

Due to unavailability of discharge measurements in the watershed, two satellites based AET products (GLEAM_v3.0a and MOD16) were used for the SWAT calibration and validation. The criteria for choosing GLEAM and MOD16 products are based on their temporal and spatial resolution and the fact that they are freely available and because these two AET data sets have been validated in several countries in Africa. To further assess the model performance in simulating other components of the water balance (e.g. soil moisture), a remotely sensed ESA CCI soil moisture v3.2 data was used to validate SWAT simulated soil moisture dynamics.

### 2.4.1 GLEAM

The Global Land Evaporation Amsterdam Model (GLEAM) combines a wide range of remote sensing observations from different satellites to separately estimate the different components of terrestrial evaporation and surface soil moisture through a process-based methodology (Martens et al., 2017). GLEAM developed in developed in 2011 has been continuously revised and updated. The Priestley and Taylor equation (1972) used in GLEAM calculates the potential evaporation (mm $d^{-1}$) based on remotely sensed observation of surface net radiation and near-surface air temperature (Eq. 4). Since GLEAM separately derives the different components of terrestrial evaporation (Eq. 5), the estimates of potential evaporation for the land fractions of bare soil, open water, tall canopy and short canopy derived are converted into actual evaporation using a multiplicative evaporative stress factor (Eq. 6) obtained from observations of microwave Vegetation Optical Depth (VOD) used as a proxy

for vegetation water content and simulations of root-zone soil moisture. Interception loss is estimated separately based on the Gash analytical model of rainfall interception driven by observations of precipitation and both vegetation and rainfall characteristics.

Two of the three version of the datasets produced in 2016 using GLEAM_v3.0 were downloaded for this study (GLEAM_v3.0a
and GLEAM_v3.0b). In this study, GLEAM_v3.0a was used for SWAT calibration and validation while GLEAM_v3.0b was used for the verification of the SWAT simulated AET.

$$E = Et + Eb + Ew + Ei + Es \tag{5}$$

$$S = \frac{(E - Ei)}{Ep} \tag{6}$$

Where $E$ is the actual evaporation (mm d$^{-1}$), $Et$ is the transpiration (mm d$^{-1}$), $Eb$ is bare-soil evaporation (mm d$^{-1}$), $Ew$ is the open-water evaporation (mm d$^{-1}$), $Ei$ is the interception loss (mm d$^{-1}$), $Es$ is the snow sublimation (mm d$^{-1}$), $S$ is the evaporative stress factor (-) and $Ep$ is potential evaporation (mm d$^{-1}$).

The datasets are provided on a $0.25^0$ by $0.25^0$ regular grid. For more information on GLEAM, its different forcing variables and the satellite data used in the GLEAM_v3.0a and GLEAM_v3.0b datasets, the reader is referred to Miralles et al. (2011b) and  Martens et al. (2017).

### 2.4.2 MOD16

The MOD16 retrieval algorithm (Mu et al., 2007, 2011) is based on the Penman–Monteith framework (Monteith, 1965) with modifications to account for parameters not readily available from space (Cleugh et al., 2007).  Terrestrial evapotranspiration includes evaporation from wet and moist soil, evaporation from rain water intercepted by the canopy before it reaches the ground, the sublimation of water vapor from ice and snow and the transpiration through stomata on plant leaves and stems (Mu et al., 2011).  Mu et al. (2007) derived actual evaporation from potential evaporation data by using multipliers to halt soil
evaporation and plant transpiration through transpiration flow that was limited by water stress and low temperatures and a complementary relationship which defines land-atmospheric interactions from relative humidity and vapour pressure deficit (Mu et al., 2007). Mu et al. (2011) apply the Penman-Monteith (P-M) equation (Eq. 3)  to calculate PET on a global scale by using variables and parameters needed from VIS/NIR remote sensing (land cover, LAI, albedo, FPAR) and from daily meteorological reanalysis data from NASA's global modeling and assimilation office (radiation, $T_{air}$, pressure, relative
humidity;).  In principle, the surface resistance ($r_s$) parameter in the P-M equation accounts for any direct effect on evapotranspiration due to limitations in available water.  The way $r_s$ is derived in the MOD16 evapotranspiration scheme only considers an indirect effect via a non-linear dependency of $r_s$ with the water vapor pressure deficit (VPD) in the atmosphere.

VPD under daytime conditions often represents a proxy for soil moisture conditions and therefore $r_s$. MOD16 AET is described in detail by Mu et al. (2007, 2011).

### 2.4.3 ESA CCI soil moisture

The European Space Agency Climate Change Initiative soil moisture data was generated by merging various available active and passive microwave based soil moisture data sets (Gruber et al., 2017; Wagner et al., 2012). The combined product used is ESA CCI SM v3.2 that was released in 2017 and was generated by blending passive and active microwave soil moisture retrieval generated by C- band scatterometers (ERS-1/2 AMS WS (TU Wien/WARP 5.5) scatterometer, ERS-2 AMI WS (TU Wien/WARPS5.4) scatterometer, MetOp-A + B ASCAT (H –SAF H109/H110/WARP 5.6) scatterometer) and multi-

frequency radiometers data (from SMMR, SSM/I, TMI, WinSat (all VUA/NASA LPRM v5); AMSR-E, AMSR2, SMOS (all VanderSat LPRM v6)) (Dorigo et al., 2017). The blending scheme of this product used a weighted average of measurements from all sensors that are available at a certain point in time to compute the merged soil moisture estimate (Dorigo et al., 2017). Since all input datasets used in generating the data have different dynamic ranges, they are rescaled through CDF-matching into a common climatology. Specifically, the soil moisture retrievals from multi-frequency radiometers products (SMMR,

SSM/I, TMI, and AMSR-E) were rescaled and merged on a pixel basis (Liu et al., 2012). ESA CCI SM v3.2 data is available at a daily resolution from 1978 to 2015, at a spatial resolution of $0.25^0$ and representing the upper soil depth from 0.5-2 cm. For more detail information on ESA CCI SM, the reader is referred to Wagner et al. (2012), Liu et al. (2012), Dorigo et al. (2014), Gruber et al. (2017), and Dorigo et al. (2017). The ESA CCI SM has been evaluated in West Africa using in-situ soil moisture data (Dorigo et al., 2014).

### 2.5 SWAT calibration, validation and uncertainty analysis

A multi-objective calibration and validation of SWAT simulated AET using satellite derived AET from GLEAM_v3.0a and MOD16 was implemented in SWAT-CUP (Abbaspour, 2015). SWAT-CUP is a package used to carry out sensitivity analysis, calibration and validation of the SWAT model. SUFI-2 (Abbaspour et al., 2004) is one of the programs available in SWAT-

CUP that is a multi-site, semi-automated, inverse modelling procedure used for calibrating parameters. SUFI-2 is based on a stochastic procedure for drawing independent parameter sets using Latin Hypercube sampling (LHS). In this paper, we followed the split-sample test as presented by Klemes (1986) and Gan et al. (1997), using a model calibration and validation approach that consists of equally splitting the available data, when the record is sufficiently long, to represent different climatic conditions i.e. wet, moderate, and dry years in both periods. An initial pre-selection of parameters based on literature research

(Bicknell et al., 1997; Wang et al., 2006; Rafiei Emam et al., 2016; Ha et al., 2018; Lopez Lopez et al., 2017) was undertaken to choose the most sensitive parameters to AET, and making sure that each of the hydrological processes (runoff, evaporation, interception, transpiration and percolation) are represented in the 50 parameters of the global sensitivity analysis. The initial

parameter ranges were based on Neitsch et al.(2002, 2005, 2011). The global sensitivity analysis based on multiple regression method (Abbaspour, 2015) was carried out in which parameter sensitivities are determined by numerous rounds of LHS to obtain the most sensitive parameters by examining the resulting p-value and the t-stat value. The p-value determines the significance of the sensitivity (a value close to zero has more significance) and the t-stat provides a measure of parameter

sensitivity (a larger absolute value is more sensitive). Based on the sensitivity analysis, 11 of the most sensitive parameters were selected and altered during calibration process using SUFI-2. The ranking and the calibrated values of the 11 parameters for each of the six calibration procedures are listed in Table 2. The equations written in SWAT theoretical documentation (Neitsch et al. (2011)) showing were the selected 11 sensitive parameters appear are presented in Appendix C.

In this study, the first three calibrations/validations GS1, GS2 and GS3 use the AET from GLEAM_v3.0a for SWAT

calibration (1989-2000) and validation (2001-2012). To compare SWAT simulated AET in each subbasin to the GLEAM_v3.0a and GLEAM_v3.0b AET pixel values and compute their NSE, $R^2$, PBIAS, KGE for each subbasin, a NetCDF raster layer was created in ArcGIS to view how many pixels of GLEAM covered each of Ogun River subbasin polygon (Fig. D1). GLEAM AET pixel value (daily resolution) was extracted for each subbasin by using "convert raster to points" and "Make NetCDF table view" tool in ArcGIS. The extracted daily data was aggregated to monthly data for each subbasin for

easy comparison with the monthly AET output from SWAT. We preferred and selected GLEAM_v3.0a AET for the calibration/validation because of its long-term availability that allows reasonably selection and splitting of calibration and validation periods, which are not substantially different in climatic condition i.e., wet, moderate, and dry years in both periods and which covers our SWAT simulation output period (1989-2012). GLEAM_v3.0a dataset spanning 24-year period 1989-2012 was used because the SWAT simulation output was from 1989-2012. The splitting of calibration period (1989-2000) and

validation period (2001-2012) for GLEAM_v3.0a AET followed the split-sample test as presented by Klemes (1986) and Gan et al. (1997).

The last three calibration/validation MS4, MS5 and MS6 use the MOD16 AET for SWAT calibration (2000-2006) and validation (2007-2012). Considering MOD16 AET available time-series period, the splitting of calibration and validation period also followed the split sample test as presented by Klemes (1986). Since MOD16 AET is a raster in geotiff format, to

compare the AET pixel value to SWAT simulated monthly AET values for each subbasin, an area-weighted averaging scheme was performed in ArcGIS to create aggregated monthly time-series of MODIS AET for each subbasin (Fig. D2).

The three model runs were calibrated (GS1 through MS6) by adjusting the 11 most sensitive parameters found in SUFI-2. In the calibration of SWAT with the AET from GLEAM a sample size of 1000 was chosen for the first iteration and a sample size of 500 for the second iteration, resulting in 1500 simulations. In the calibration of SWAT with AET from MOD16 a

sample size of 1000 was chosen for two iterations of LHS, resulting in 2000 simulations. The validation process involved running the model using parameter values that were determined during the calibration process and comparing the SWAT AET simulations to satellite based AET data.

In this study, we do not consider runoff-measured data for an independent validation because it is not available for the study basin and this is the main reason we considered AET derived from satellite products as an alternative option for the SWAT

model calibration and validation. We believe using available AET products (GLEAM & MOD16), that have been tested in the past in various calibration and validation studies undertaken by a number of scientists (Roy et al., 2017, Herman et al., 2017, Lopez Lopez et al., 2017, Ha et al., 2018, Pomeon et al., 2018) is one solution in setting up a hydrological model that will be used as a decision support tool in such a data scarce region.

The three SWAT model run calibrated and validated using GLEAM and MOD16 AET (GS1 through MS6) were evaluated with four objective functions, in which their mathematical formulations are presented in Appendix A. It should be noted again that the AET in the GLEAM and MOD16 products does not stem from measured data obtained from eddy covariance instruments, but instead are based on global earth observation products (satellite).

Presently, the general hydrologic model performance ratings for recommended statistics (NSE, PBIAS, $R^2$) performed at a
monthly time and mentioned by Moriasi et al. (2007, 2015) are mostly relevant for runoff, sediment and nutrients. For this study, the literature was searched on model evaluation methods for hydrologic model calibration using satellite or non-satellite derived evapotranspiration. The reviewed literature (Djman, 2016; Samadi et al., 2017; Lopez Lopez et al., 2017; Ha et al., 2018) showed that these studies also set their performance ratings for recommended statistics (NSE, PBIAS and, $R^2$) based on Moriasi et al. (2007, 2015) guidelines. In this study, we followed Lopez Lopez et al. (2017) and others to base our performance
rating criteria for judging the SWAT model performance (GS1 to MS6) by using Nash-Sutcliffe efficiency (NSE, (Nash and Sutcliffe, 1970)), Kling-Gupta efficiency (KGE,(Gupta et al., 2009)), the percent bias (PBIAS) and the coefficient of determination ($R^2$). NSE ranges from $-\infty$ to 1, where NSE > 0.5 indicates a good agreement (Moriasi et al., 2007, 2015) between simulated and satellite based evapotranspiration and NSE of 1 being the optimal value. $R^2$ ranges from 0 to 1 with higher values indicating less error variance and 1 being the optimal value. KGE ranges from $-\infty$ to 1, where KGE of 1 is the
optimal value. PBIAS ranges from $-\infty$ to $\infty$, where low magnitude values indicate better simulation. The optimum value of PBIAS is 0. In this paper, NSE is the selected objective function that was optimised during the calibration process.

The recommended statistics for a monthly time step based on Kouchi et al.( 2017) and Moriasi et al.( 2007, 2015), states that NSE>0.50, $R^2$>0.60, KGE≥0.50 and PBIAS ≤ ±25% are the required satisfactory threshold. SUFI-2 was also used for the uncertainty analysis of the AET modelling process. In this step, the procedure depicts the 95% prediction uncertainty (95PPU)
of the model compared with satellite based AET. The 95PPU was estimated at the 2.5% and 97.5% levels of the cumulative distribution of the AET simulated output variable derived through LHS. The uncertainties were quantified by two indices referred to as P-factor and R-factor (Abbaspour et al., 2004). The P-factor represents the percentage of observed data plus its error bracketed by the 95% predictive uncertainty (95PPU) band and varies from 0 to 1. Where 1 indicates a 100% bracketing of the observed data within model simulations. While the R- factor is the ratio of the average width of the 95PPU and the
standard deviation of the observed variable, this value ranges between 0 and infinity. These two indices were also used to judge the strength of the calibration and validation in which the ideal situation would be to account for 100% of the satellite AET data in the 95PPU while at the same time have an R-factor close to 0.

## 2.6 SWAT Model Verification

In some modelling studies (EPA, 2013; Faramarzi et al., 2017), the term model verification is used to refer to the examination of the numerical technique and computer code to ascertain that it truly represents the conceptual model and that there are no inherent numerical problems with obtaining a solution. In this study, to further examine the accuracy of the calibrated SWAT model, a verification of the simulated variables was carried out by: (i) a graphical comparison of calibrated SWAT simulated AET to GLEAM_v3.0b AET time-series (2003-2012). We considered GLEAM_v3.0b dataset for the verification of SWAT simulated AET because there are no ground truth AET data in the study area and, because of its different forcing variable, which categorizes it as an independent dataset not considered in the calibration and validation,; and (ii) assessment of the long-term average annual and average monthly water balances at the outlet of the watershed. The long-term average monthly and annual water balance assessment was based on SWAT simulated output with only precipitation and temperature as measured input data.  The SWAT water balance equations used for the assessment are:

$$WYLD = SURQ + LAT\_Q + GW\_Q - Q\_TLOSS \tag{7}$$

$$PRECIP = WYLD + AET + \Delta SW + PERC - GW\_Q \tag{8}$$

Where *PRECIP (mm)* is the observed precipitation; *AET (mm)* is the actual evapotranspiration; *WYLD (mm)* is the net amount of water that leaves the subbasin and contributes to stream flow in the reach; *SURQ (mm)* is the surface runoff contribution to stream flow; *GW_Q (mm)* is the groundwater contribution to stream flow; *ΔSW(mm)* is the change in soil water content; *PERC(mm)* is the water percolating past the root zone; *LAT_Q (mm)* is the lateral flow contribution to stream flow and *Q_TLOSS (mm)* is the transmission loss. The soil water content for both monthly and annual output is the average soil water content for the time period. Hence, the initial soil water content is the average for the time period of 25 years.

## 2.7 SWAT soil moisture validation

Soil moisture is a key driver for runoff, evaporation and infiltration processes in a catchment. The ESA CCI SM v3.2 was used for validation because it has a similar spatial resolution to the satellite-based AET products used for model calibration, the available time period (1978 -2015) fits the period of interest, and the product has achieved reasonable accuracy when evaluated at various sites in West Africa  (Dorigo et al., 2014). The SWAT simulated soil moisture was validated using SUFI-2 against ESA CCI SM v3.2 for each of the Ogun River subbasins using 500 simulations in one iteration from the period of 2001 to 2012. The $R^2$ statistics was computed between SWAT simulated soil moisture and ESA CCCI SM v3.2 for each pixel (Fig. E1) using the same approach described for GLEAM extraction in section 2.5. The ESA CCI SM product provides volumetric moisture in the shallow soil depth (0.5-2 cm) whereas SWAT provides plant available soil moisture for the total soil layer (0-200 cm), therefore a direct comparison was not possible. Hence, the proportion of the variance was considered to be an important criteria for evaluating the dynamics of soil moisture simulated, rather than evaluating the absolute values (Fig. 14).

## 3. Results

The results of global sensitivity analysis revealed that the SCS runoff curve number (CN2.mgt) is the most sensitive parameter to SWAT simulations of AET for all the six calibrations (Table 2). The sensitivity ranking of the remaining 10 parameters varies significantly.

Table 2

Figure 3 show the performance of the uncalibrated SWAT (RGS1, RGS2, RGS3, RMS4, RMS5, and RSM6), which represent the reference runs. Results indicate that the uncalibrated SWAT model underestimated AET (positive PBIAS) and has a high

10   percentage deviation from the GLEAM and MOD16 AET. The NSE, KGE, PBIAS and $R^2$ all depict a low model performance. Interestingly, the reference runs with MOD16 tend to have higher $R^2$ compared to reference runs with GLEAM. The results from all the six-reference runs justify the need to further improve the SWAT model performance.

Fig. 3

In this paper, only the spatial representations of the SWAT_HG with the highest (GS1) and the lowest (MS6) model performance are included for the purpose of showing the two extreme results obtained (Fig. 4, Fig. 5, Fig. 6 and Fig. 7). The calibration/validation results for GS1 show a model performance of NSE>0.50, KGE>0.50, $R^2$ >0.6 in more than half of the 53 subbasins and a PBIAS < ±15% in all of the 53 subbasins (Fig.4 and Fig 5). The calibration/validation results for MS6

20   (Fig.6 and Fig.7) show the lowest model performance.

Fig.4
Fig 5

Fig.6
Fig.7

30   The remaining calibration/validation results of GS2, GS3, MS4, MS5 (Fig. S1, Fig S2, Fig S3, Fig, S4, Fig S5, Fig S6, Fig S7 and Fig S8) show a lower model performance to GS1.

Figure 8 and 9 summarize the model performance results of the SWAT model runs when calibrated/validated with GLEAM_v3.0a (GS1, GS2 and GS3) and MOD16 AET (MS4, MS5 and MS6). Overall, results indicate that the

calibration/validation of GS1 (Fig.8 and Fig.9) exhibits a model performance superior to the remaining two model runs for AET simulation (through GS1 to MS6) judging by the four objective functions except for the validation period, where a lower NSE (average value of 0.45) was obtained (GS1, Fig. 9).  NSE>0.50 was achieved in 32 out of 53 subbasins during the model validation (GS1), meaning that, more than half of the 53 subbasin have a satisfactory model performance, therefore the average NSE value of 0.45 obtained can be considered acceptable.

Fig.8

Fig.9

## 3.1 Uncertainty analysis of SWAT model

The SWAT model performance results of the SWAT-HG when calibrated/validated with the AET from GLEAM_v3.0a (GS1) proved to be the most efficient of the three model runs (through GS1 to MS6), therefore, it was used to further predict the uncertainty associated with the AET simulations for each of the 53 subbasins to map error sources. In the calibration period, the values of the P-factor obtained were between 0.50 and 0.90 and the values of the R-factor were between 1.40 and 2.4.  In the validation period, the values of P-factor were between 0.6 and 0.88, and that of the R-factor were between 1.43 and 2.5. The P-factor values revealed that more than half of the earth observation derived AET plus its error are bracketed by the 95% predictive uncertainty. The predictive uncertainties were adequate in the 53 subbasins and had a satisfactory performance for monthly AET simulations using the Hargreaves equation, though the R-factor was quite large in all the 53 subbasins, indicating large model uncertainties. Extracts of the monthly calibration/validation results showing the 95% prediction uncertainty intervals along with the satellite based AET (GLEAM_v3.0a) are presented in Fig.10.

Fig.10

## 3.2 Model verification result

It was found that the AET from GLEAM_v3.0b was bracketed within the 95 percent uncertainty prediction (Fig. 10). The long-term average monthly water balance assessment performed at the outlet of the watershed shows a seasonal fluctuation which agrees with previous water balance studies conducted at the outlet of the of the study area located in Abeokuta (Ufoegbune et al., 2011; Eruola et al., 2012; Ufoegbune et al., 2012; Sobowale and Oyedepo, 2013), namely: (i) the study area is characterized by bimodal rainfall pattern, (ii) the AET increases in February from 55 mm to 76 mm as the wet season approaches and decreases in October from 72 mm to 54 mm in November as the dry season approaches (Ufoegbune et al., 2011), (iii) rainfall commences in March (66 mm) and is plentiful in June (165 mm) and September (167 mm), (iv) in August there is a decrease in precipitation to 96 mm  and a decrease in AET to 94 mm, the dry spell often experienced in August is termed "August break" (Ufoegbune et al., 2011), (iii) The dry period extends from November to March, the months of low

rainfall, AET, and soil moisture values (Ufoegbune et al., 2011), (v) with moderate rain in March soil water increases from 83 mm to 200 mm in July (Ufoegbune et al., 2011), (vi)  as dry season commences, the soil water gradually declines.

The differences in the long term mean monthly water balance values obtained in the past studies conducted within the catchment are due to the variation in duration of years considered. Also, Eruola et al. (2012) revealed the two peak rainfalls in July and September agree with the current study, while Ufoegbune et al. (2011) showed the two peak rainfall to be in  the month of June and September. All these previous studies and the current study water balance results are in the same range. Figure 12 shows the seasonal fluctuation of the SWAT estimated water balance components at the outlet of watershed, located in Abeokuta town. Our results show, the average long-term annual water balance estimated by SWAT to be within a reasonable percentage error of closure of ±15% (Table 3).

Table 3

Fig. 11

Fig. 12

**3.3 SWAT soil moisture validation result**

Only the spatial representation of the SWAT_HG with the highest AET calibration model result was validated against the ESA CCI SM. Overall, the average $R^2$ value obtained for the Ogun River Basin is 0.76 (Fig. 13). The results show 88.6% of the basin has $R^2 > 0.60$ and 11.3% of the basin has $R^2 < 0.60$. The graphical representation of the soil moisture dynamics from the highest and the lowest $R^2$ results obtained are presented in Fig. 14.

**4. Discussion**

The global sensitivity analysis revealed that for the three SWAT model set-up calibrations (GS1 to MS6), the same 11 SWAT hydrologic parameters governing AET were sensitive (Table 2). When different PET equations were tested in SWAT, different simulated AET values were obtained and the overall sensitivity ranking of the parameters varied significantly. Since parameters represent processes, the significant variation in the sensitivity ranking of the parameters implies that the impact of the selected PET methods in simulating the AET in the study area is relatively high. The SCS runoff curve number for moisture condition II (CN2.mgt), which is one of the parameters that controls the overland processes and is a function of the soil permeability, landuse and antecedent soil water was used to determine the surface runoff generation in the basin. The CN2.mgt is found to be the most sensitive parameter of the three model runs (through GS1 to MS6), indicating that it is also the dominant parameter controlling the AET processes in SWAT for the Ogun River Basin (Table 2). The soil evaporation compensation coefficient (ESCO.hru) controls soil evaporation and depends on soil characteristics. As the value of ESCO is reduced, more water is extracted from the lower layers to meet evaporative demand. ESCO is the second most sensitive parameter for GS1, with a very low value of 0.02, but for other runs (GS2 to MS6), the sensitivity ranking varies. The maximum canopy storage

(CANMAX.hru) accounts for the amount of water that can be trapped in the canopy when fully developed and affects infiltration, surface runoff, and evapotranspiration processes. It was more sensitive when the model was calibrated with GLEAM than when the model was calibrated with MOD16 AET. The soil bulk density (SOL_BD. sol) defines the relative amounts of pore space and its overall sensitivity ranking is between 2 and 4 (GS1 through MS6). It was more sensitive when

SWAT was calibrated with MOD16 than when SWAT was calibrated with GLEAM AET. The baseflow alpha factor (ALPHA_BF.gw) is the index of groundwater flow response to changes in recharge influences the baseflow simulation and its sensitivity varies (GS1 to MS6). The saturated hydraulic conductivity (SOL_K.sol) determines the shallow sub-surface flow and groundwater recharge and it affects the surface runoff response. In order to correctly account for volume of water lost to evaporation from the two reservoirs (Oyan dam and Ikere George dam), the evaporation coefficient (EVRSV.res) was

calibrated. Interestingly, we observed that GS1, GS3 and MS6 have higher EVRSV.res values that agree with the expected values in such a humid tropical region compared to EVRSV.res values obtained from GS2, MS4 and MS5. It is observed that GS1 tends to have the highest (4.7) maximum stomatal conductance (GSI.plant.dat) that denotes the maximum conductance of a leaf when the canopy resistance term is modified to reflect the impact of high vapour pressure deficit, during the calibration. The MS4 has the highest value (0.99) of initial soil water storage expressed as a fraction of field capacity water

content (FFCB). The MS4 also obtained the highest value (0.95) of plant uptake compensation factor (EPCO.hru), meaning that the calibration procedure makes  the model allows for more of the water uptake demand to be met by lower layers in the soil, and as EPCO approaches 0 (GS3 with the value 0.07).  The soil available storage capacity (SOL_AWC.sol) values for GS1 to MS6 were allowed to vary by a factor of 0.8 to 0.96, meaning there is an increase in SOL_AWC for the soil in the model set-up for Ogun River Basin.

Assessing the model performance with the objective function and their satisfactory threshold values used in this study, the calibration/validation with the AET from GLEAM_v3.0a showed an overall satisfactory SWAT model performance when the Hargreaves PET equation was used in SWAT to simulate AET (GS1), compared to the other model calibration results (GS2 to MS6). The calibration/validation with the AET from MOD16 yielded a lower SWAT model performance regardless which of the three PET equations was tested in SWAT.

Using the guidelines in Moriasi et al. (2007, 2015) and Kouchi et al. (2017) for evaluating the SWAT model performance at a monthly time-step, the PBIAS values showed a satisfactory model performance (PBIAS≤±25) in the six calibrations/validations of the three SWAT model runs (Fig. 8 and Fig.9). The positive PBIAS obtained in the calibration/validation of the three SWAT model run using the AET from MOD16 (MS4, MS5 and MS6) indicated a tendency for the SWAT model to underestimate monthly AET at the Ogun River Basin. An intuitive reason for this may be due to

transient water stress occurring in the basin, however, transient water stress is not the main challenge in the study area, which is located in the humid region of southwestern Nigeria with a mean Aridity Index of 0.75 (for the period 1989-2012). The careful consideration of equal wet and dry years in the calibration and validation years has accounted for similar climatic conditions in both periods.

The positive PBIAS obtained using MOD16 for calibrating agrees with previous studies conducted at a site in tropical region. Ruhoff et al. (2013) validated MOD16 AET using ground-based measurements of energy fluxes obtained from eddy covariance sites installed in tropical sites in the Rio Grande basin, Brazil from a hydrological model (MGB-IPH) at both local and regional scales and found that at the natural savannah vegetation site, the annual AET estimate derived by the MOD16 algorithm was

19% higher than the measured amount. Ruhoff et al. (2013) found that misclassification of land use and land cover was identified as the largest contributor to the error from the MOD16 algorithm. Ramoelo et al. (2014) validated MOD16 using data from two eddy covariance flux towers installed in a savannah and woodland ecosystem within the Kruger National Park, South Africa and found that one flux tower results showed inconsistent comparisons with MOD16 AET and the other site achieved a poor comparison with MOD16 ET. In their study, they found that, the inconsistent comparison of MOD16 and flux

tower-based AET can be attributed to the parameterization of the Penman-Monteith model, flux tower measurement errors, and flux tower footprint vs MODIS pixel. Also, Trambauer et al. (2014) compared different evaporation products in Africa and found that MOD16 evaporation does not show a good agreement with other products in most part of Africa, while other evaporation datasets (GLEAM, ECMWF reanalysis ERA-LAND and PCR-GLOBWB hydrological model simulated AET) are more consistent. From our results, we found that when the SWAT model was calibrated with MOD16 AET, the SWAT

simulations tend to underestimate AET.

The satisfactory SWAT model GS1 performance was achieved for all objective functions, except for the average NSE value of 0.45 in the validation period, however NSE values >0.50 were obtained in 60% of the subbasins. The KGE result revealed the SWAT-HG model validation (GS1) to be satisfactory (Fig 9). Also, the low PBIAS result of -0.02% and 0.45% (GS1, Fig 8 and Fig 9) corresponded to a performance rating "very good" indicating predictive capability of accurate model simulation.

The better SWAT model performance in GS1 is attributed to the selection of the Hargreaves equation, which is based on available observed precipitation and maximum and minimum temperature to obtained AET, while the Penman-Monteith and the Priestly-Taylor equations are driven by simulated variables (wind speed, relative humidity and solar radiation) in this study. Also the complex water balance model algorithm of GLEAM takes into account soil-water balance, bare-soil evaporation and open water evaporation, evaporative stress factor and rainfall interception, all of which  assist in simulating the dynamic

hydrological components, especially the AET.

The differences in GLEAM and MOD16 products are due to their input and forcing data (Trambauer et al. 2014). Our results agree with another study in which AET from GLEAM performed satisfactorily for the calibration of a large-scale hydrological model set up in Morocco (Lopez Lopez et al., 2017). The 95% predictive uncertainty of the highest SWAT model performance (GS1) was quantified, and the 95% predictive uncertainty bracketed most of the satellite based AET, although the R-factor

was quite large in all of the subbasins signifying a large model uncertainty which can be ascribed to the uncertainty in satellite derived AET, the forcing climate data, the conceptual model and the model parameters. The 95PPU are the combined outcome of the uncertainties, these uncertainty sources are not separately evaluated in SUFI-2 but attributed as a total model uncertainty to the parameters which are visualized through the simulated model output ranges. A first verification of the SWAT model run with the best model performance (GS1) was carried out using GLEAM_v3.0b as an independent dataset and found the results

to be bracketed within the 95PPU of GS1 (Fig. 11). The second verification of the SWAT model structure with the best model performance (GS1) was carried out by assessing the output of SWAT water balance components (Table 3 and Fig. 12). The results obtained from the long-term mean monthly water balance agrees with the previous water balance studies conducted within the study area. The differences in the water balance components values of the past and the current study are due to

variation in the length of years considered. The average long-term annual of the water balance at the outlet of the study area shows a satisfactory percentage error of closure of ±15% (Table 3).

At a monthly-time step, the dynamics of the SWAT simulated soil moisture (mm) for the whole soil profile compared to the ESA CCI SM (%) in the upper few centimetres of the soil profile fit very well in most of the basin. The lowest $R^2$ of 0.45 obtained in  subbasin 44 of Ogun River Basin (Fig. 14) also corresponded to the lowest result ($R^2 = 0.28$) obtained in subbasin

44 when the model was validated with AET from GLEAM_v3.0a (Fig. 10). Nevertheless, the high overall average soil moisture validation $R^2$ obtained agrees with a previous study that validated a hydrological model with ESA CCI SM in West Africa (Pomeon et al. 2018). The multi-calibration and validation results show the SWAT model to perform satisfactorily in the study area.

## 5. Conclusion

This study examined an alternative method to calibrate/validate the SWAT eco-hydrological model using available satellite-based AET products for the data-sparse Ogun River Basin in southwestern Nigeria. The approach opens up a new direction for calibration/validation of hydrological models in ungauged basins. Due to the different retrieval algorithms of both the satellite-based AET and SWAT simulated AET, two global evaporation products ((GLEAM and MOD16) were used to calibrate the three SWAT simulated AET on a monthly time scale. The use of Hargreaves, Priestley-Taylor and Penman-

Monteith equations in SWAT cause the different AET values obtained. Six different calibrations were implemented with the global AET products with the aim to obtain a high performing model for Ogun-River Basin. Overall, the results are promising, and show that global satellite-based AET data can be used as an alternative method to calibrate/validate the SWAT model in a tropical sparsely-gauged basin. Specifically, when SWAT model was used with the Hargreaves PET equation and was calibrated using the GLEAM_v3.0a AET product, the highest model performance was obtained with an acceptable predictive

uncertainty.

Statistical analysis of the model performance shows that global AET datasets used for the calibration were significantly different from each other, which was expected because of their different retrieval algorithms. Our findings suggest that the SWAT model run using the Hargreaves equation can be used as a potential decision support tool for further studies and predictions on basin hydrology in the Ogun River Basin.

There is still a need for further research on: (i) improving the model calibration performance in those subbasins where the performances are unsatisfactory and (ii) validation of other simulated variable (e.g. stream flow) of the calibrated SWAT model using observed datasets when these are available.

The results from this research contribute to a better understanding of the ease and suitability of using freely available satellite based AET datasets for model calibration in a tropical ungauged basins where the main limitation of setting-up hydrological models for discharge simulations is the lack of measured streamflow data. Furthermore, a new contribution of this study is the better understanding of calibration of the three different estimated AET in SWAT to derive the model with the best goodness

of fit and a satisfactory predictive capability.

We recommend testing the three PET equations in SWAT to simulate AET whenever SWAT calibration is carried out with any satellite-based AET products and to independently validate other water balance components. The work presented here is a first step in the hydrological modelling study that will set the basis for future modelling applications within the study basin.

**Author Contributions**: Abolanle E. Odusanya, Bano Mehdi and Karsten Schulz designed the methodological framework and advised and contributed to the entire strategic and conceptual framework of the study. Abolanle E. Odusanya performed the simulations, analyzed the results and prepared the manuscript under supervision of Bano Mehdi and Karsten Schulz. Christoph Schürz prepared the landuse and soil maps, wrote the SWATfarmR script, and modified topHRU code for this study. Adebayo O. Oke, carried out the field work for point source water pollution data collection used in the SWAT configuration. Olufiropo.

S. Awokola, Julius A. Awomeso and Joseph O. Adejuwon carried out the field work for the necessary data input for the two reservoirs used in the SWAT configuration.

**6. Acknowledgements**

The authors wish to thank those persons, institutions, authorities and agencies, especially Olusegun Orekoya (NIMET) that assisted in accessing research data and information. We would also like to thank Roger Moussa and the anonymous reviewers

for their constructive comments, which contributed to improving the overall quality of this paper.

**Conflicts of Interest**:  The authors declare no conflict of interest

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

5   **Table 1. Description and sources of input data used to configure SWAT model for the Ogun River Basin**

| Data type | Description/Resolution | Sources |
|---|---|---|
| Topography | 30m Resolution, Digital Elevation Model 1 arc-second global coverage | Shuttle Radar Topography Mission (SRTM, 2015) https://lta.cr.usgs.gov/SRTM1Arc |
| Soil | 250m Resolution, soil property maps of Africa | Soil property maps of Africa (Hengl et al., 2015) http://www.isric.org/projects/soil-property-maps-africa-250-m-resolution |
| Landuse | 300m Resolution, landuse classification Year 2010 | European Space Agency global land cover map (ESA CCI LC, 2014) https://www.esa-landcover-cci.org/?q=node/158 |
| Weather | Daily precipitation, max. and min. temperature (1984-2012) | Nigerian Metrological Agency |
| Reservoir outflow | Reservoir daily discharge (Oyan:2007-2012) | Ogun-Oshun River Basin Authority Nigeria |
| Reservoir Water level | Daily water level (Oyan:1984-2012) | Ogun-Oshun River Basin Authority Nigeria |
| Management practices | Major crop management practices | Ogun state Agricultural Development Authority, Nigeria Oyo state Agricultural Development Authority, Nigeria |

**Table 2. Sensitivity rank and calibrated parameters with their optimal value of the three SWAT model run through the six calibrations**

| SWAT Parameter | Description | Rank (optimal value) | | | | | |
|---|---|---|---|---|---|---|---|
| | | GS1 | GS2 | GS3 | MS4 | MS5 | MS6 |
| r__CN2.mgt | SCS runoff curve number for soil moisture condition II | 1 (-0.01) | 1 (-0.13) | 1 (0.08) | 1 (-0.48) | 1 (-0.48) | 1 (-0.47) |
| v__ESCO.hru | Soil evaporation compensation coefficient | 2 (0.02) | 4 (0.20) | 3 (0.20) | 4 (0.23) | 8 (0.33) | 5 (0.50) |
| v__CANMX.hru | Maximum canopy storage | 3 (6.96) | 2 (0.61) | 2 (3.86) | 5 (82.11) | 9 (33.9) | 4 (15.6) |
| r__SOL_BD,sol | Moist bulk density | 4 (-0.19) | 3 (0.11) | 4 (-0.20) | 3 (-0.82) | 3(-0.005) | 2 (-0.07) |
| v__ALPHA_BF.gw | Baseflow alpha factor | 5 (0.66) | 5 (0.62) | 7 (0.13) | 6 (0.42) | 6 (0.9) | 8 (0.14) |
| r__SOL_K.sol | Saturated hydraulic conductivity | 6 (0.23) | 10 (-0.26) | 8 (0.24) | 10 (0.49) | 10 (-0.19) | 10 (0.26) |
| v__EVRSV.res | Lake evaporation coefficient | 7 (0.59) | 7 (0.55) | 10 (0.62) | 8 (0.22) | 7 (0.23) | 7 (0.74) |
| v__GSI.plant.dat | Maximum stomatal conductance | 8 (4.7) | 11 (1.66) | 11 (3.4) | 7 (2.34) | 5 (1.9) | 6 (0.34) |
| v__FFCB.bsn | Initial soil water storage expressed as a fraction of field capacity water content | 9 (0.59) | 6 (0.82) | 5 (0.15) | 11 (0.99) | 11 (0.4) | 11 (0.83) |
| v__EPCO.hru | Plant uptake compensation factor | 10 (0.47) | 9 (0.61) | 9 (0.07) | 9 (0.95) | 4 (0.88) | 9 (0.47) |
| r__SOL_AWC.sol | Soil available water storage capacity | 11 (0.8) | 8 (0.92) | 6 (0.77) | 2 (0.96) | 2 (0.89) | 3 (0.93) |

"v_" means a replacement (initial or existing parameter value is to be replaced by a given value);

"r_" means a relative change (initial or existing parameter value is multiplied by 1+ given value within the range)

**Table 3. Average annual water balance at the outlet of the watershed in Abeokuta Town based on SWAT simulated output**

| Year | PRECP (mm) | AET (mm) | SW (mm) | PERC (mm) | SURQ (mm) | GW_Q (mm) | WYLD (mm) | LAT Q (mm) | ΔSW (mm) | *Estimated PRECP | Balance Year | PBIAS (%) |
|------|------|------|------|------|------|------|------|------|------|------|------|------|
| 1989 | 1357 | 941 | 57 | 188 | 294 | 147 | 456 | 5 | 5 | 1442 | -85 | -6 |
| 1990 | 1094 | 882 | 82 | 69 | 145 | 52 | 207 | 4 | -25 | 1081 | 13 | 1 |
| 1991 | 1161 | 881 | 54 | 117 | 228 | 84 | 321 | 4 | 27 | 1263 | -101 | -9 |
| 1992 | 1066 | 806 | 57 | 113 | 177 | 86 | 274 | 4 | -3 | 1104 | -38 | -4 |
| 1993 | 1185 | 862 | 63 | 55 | 305 | 38 | 351 | 4 | -5 | 1225 | -41 | -3 |
| 1994 | 870 | 768 | 47 | 34 | 96 | 17 | 118 | 3 | 16 | 918 | -48 | -6 |
| 1995 | 1166 | 858 | 55 | 116 | 225 | 83 | 317 | 4 | -8 | 1200 | -34 | -3 |
| 1996 | 1457 | 885 | 45 | 201 | 460 | 148 | 621 | 5 | 10 | 1569 | -112 | -8 |
| 1997 | 1341 | 851 | 110 | 151 | 342 | 122 | 478 | 5 | -65 | 1292 | 50 | 4 |
| 1998 | 1107 | 767 | 81 | 124 | 290 | 93 | 394 | 4 | 29 | 1222 | -114 | -10 |
| 1999 | 1515 | 900 | 100 | 223 | 458 | 183 | 656 | 5 | -19 | 1577 | -62 | -4 |
| 2000 | 1198 | 814 | 55 | 175 | 306 | 143 | 463 | 4 | 45 | 1355 | -157 | -13 |
| 2001 | 841 | 738 | 35 | 27 | 108 | 12 | 128 | 3 | 20 | 900 | -60 | -7 |
| 2002 | 1241 | 758 | 64 | 146 | 375 | 108 | 492 | 4 | -29 | 1260 | -19 | -2 |
| 2003 | 1456 | 845 | 56 | 216 | 488 | 177 | 681 | 5 | 8 | 1572 | -117 | -8 |
| 2004 | 1156 | 922 | 44 | 90 | 186 | 69 | 265 | 4 | 12 | 1220 | -64 | -6 |
| 2005 | 915 | 792 | 41 | 27 | 114 | 14 | 134 | 3 | 3 | 942 | -27 | -3 |
| 2006 | 1153 | 804 | 46 | 128 | 263 | 94 | 365 | 4 | -5 | 1198 | -45 | -4 |
| 2007 | 1600 | 910 | 50 | 229 | 552 | 175 | 742 | 6 | -4 | 1702 | -103 | -6 |
| 2008 | 1395 | 832 | 55 | 221 | 416 | 174 | 605 | 5 | -4 | 1480 | -85 | -6 |
| 2009 | 1338 | 872 | 65 | 185 | 334 | 151 | 500 | 5 | -10 | 1397 | -59 | -4 |
| 2010 | 1609 | 928 | 91 | 232 | 519 | 189 | 722 | 6 | -26 | 1667 | -58 | -4 |
| 2011 | 1264 | 815 | 64 | 172 | 367 | 134 | 515 | 5 | 27 | 1395 | -130 | -10 |
| 2012 | 1409 | 839 | 60 | 265 | 386 | 205 | 609 | 6 | 4 | 1512 | -103 | -7 |

PRECIP: precipitation; AET: actual evapotranspiration; SW: soil water; PERC: percolation; SURQ: surface runoff; GW_Q: groundwater recharge; WYLD: water yield; LAT_Q: lateral flow; SW: change in soil moisture; * Estimated PRECP is WYLD+AET+ΔS+PERC-GW_Q, expressed in mm.

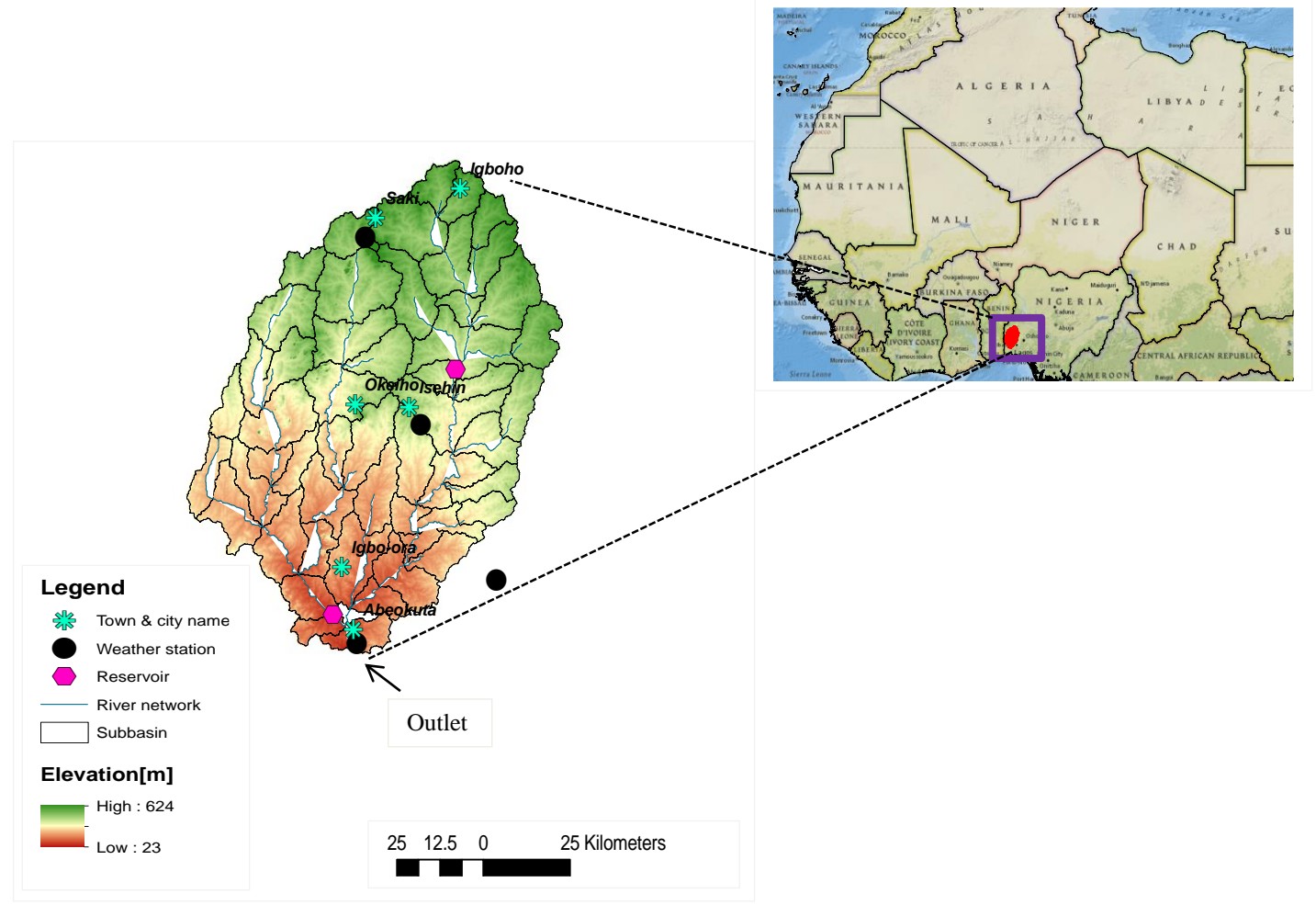

**Figure 1. The Ogun River Basin located in Nigeria showing the SWAT-delineated subbasins, weather stations and river network**

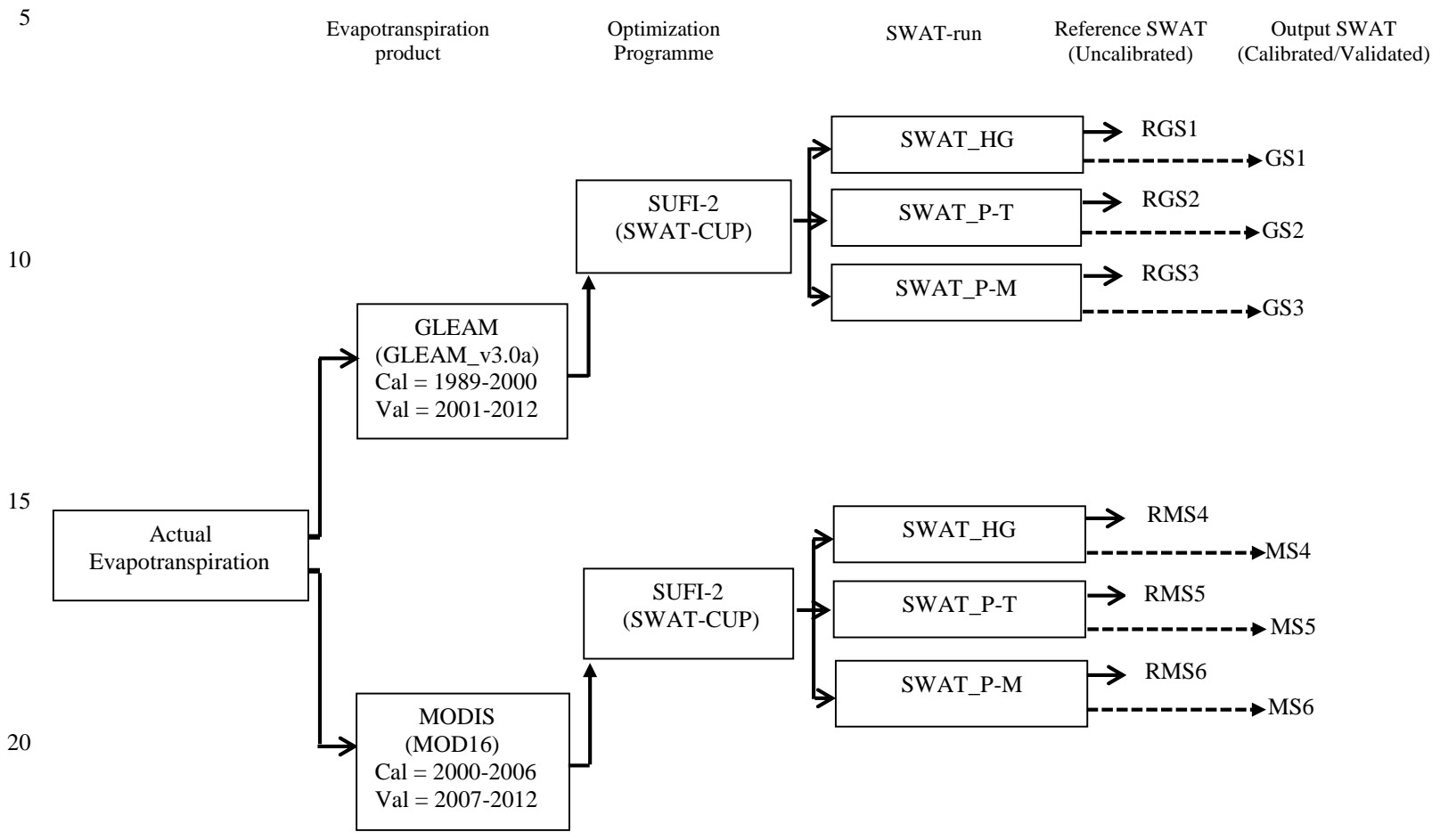

**Figure 2. Schematic diagram showing the set-up of the SWAT model, the two global AET products, the resulting six SWAT reference runs, and calibration and validation procedures for the Ogun River Basin.**

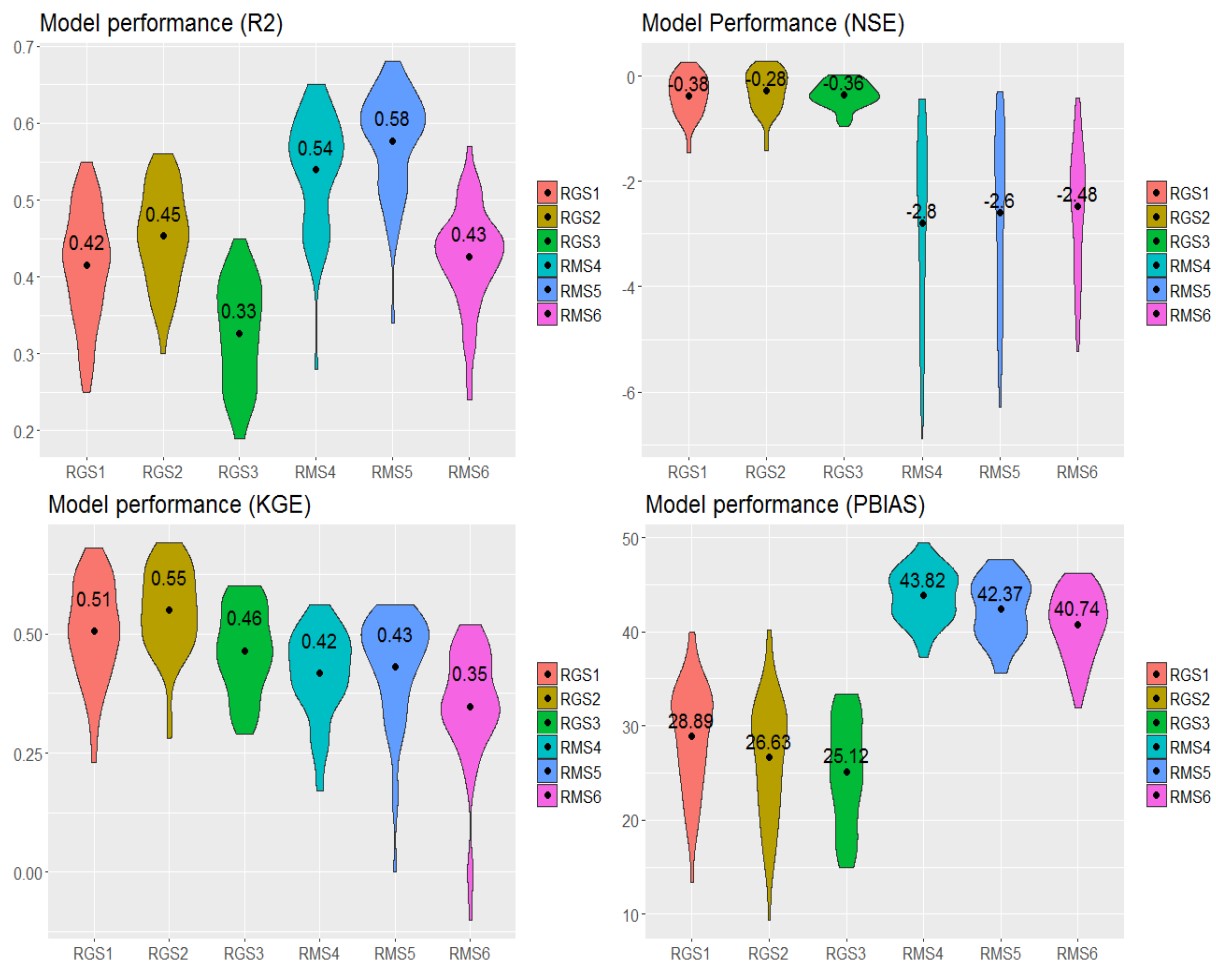

Figure 3. The plots of the performance results of the uncalibrated SWAT in simulating actual evapotranspiration. The values and the black dot symbol ("•") depicts the average value of, $R^2$, NSE, KGE and PBIAS obtained for each of the reference runs.

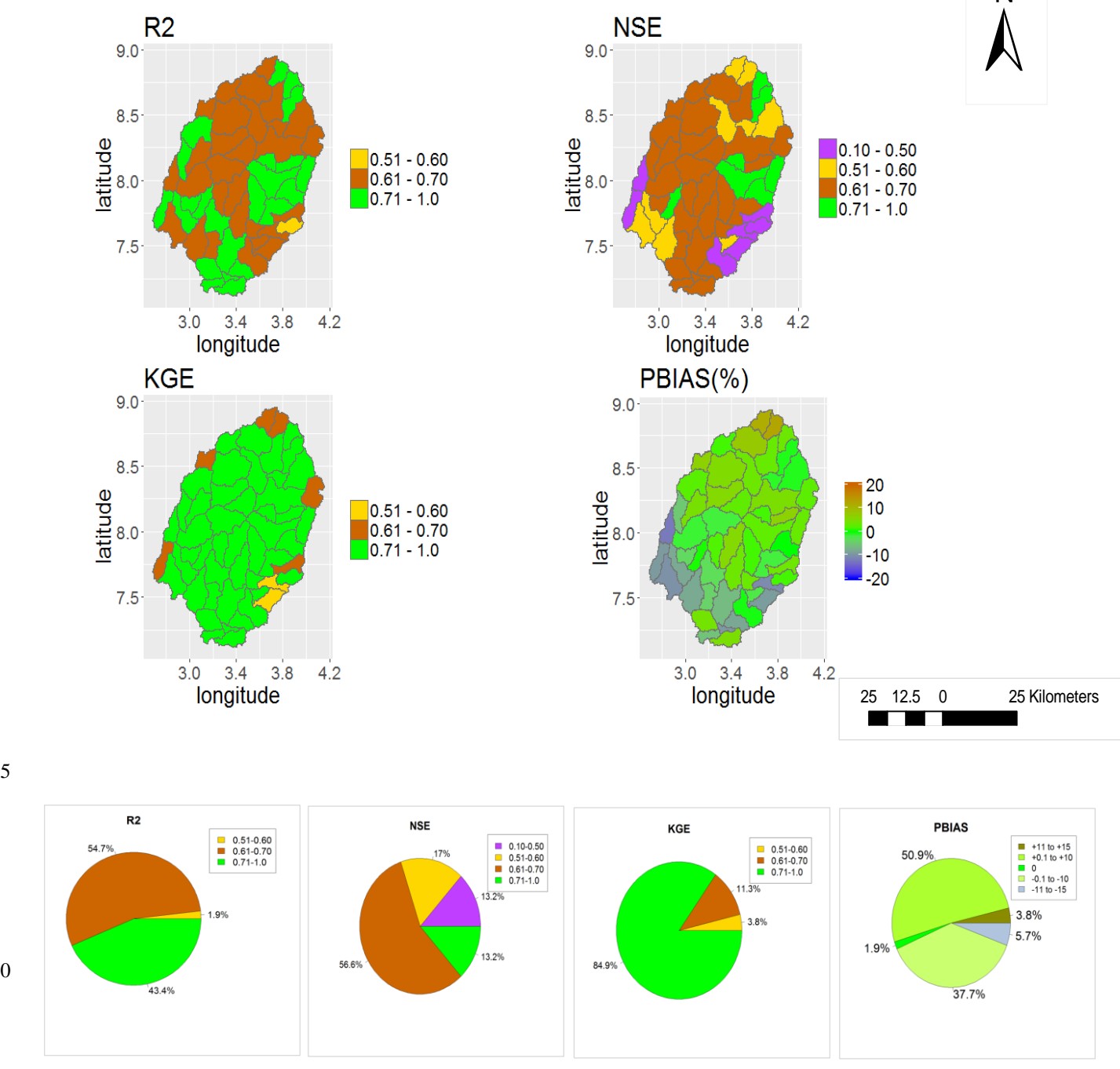

**Figure 4. Performance metrics (NSE, KGE, R², and PBIAS) of SWAT (SWAT_HG) when calibrated with GLEAM_v3.0a (GS1).**

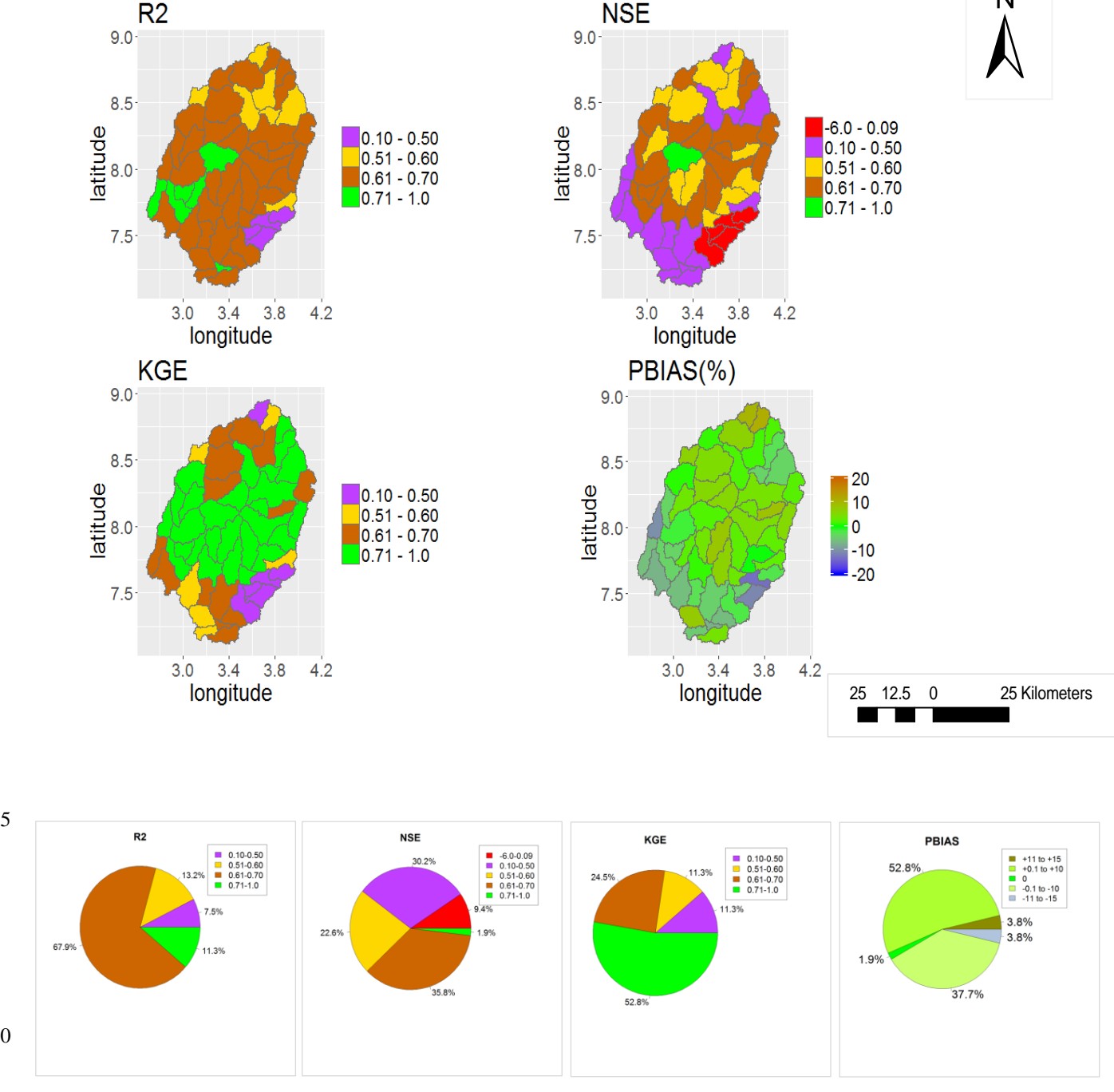

**Figure 5. Performance metrics (NSE, KGE, R², and PBIAS) of SWAT (SWAT_HG) when validated with GLEAM_v.3.0a (GS1)**

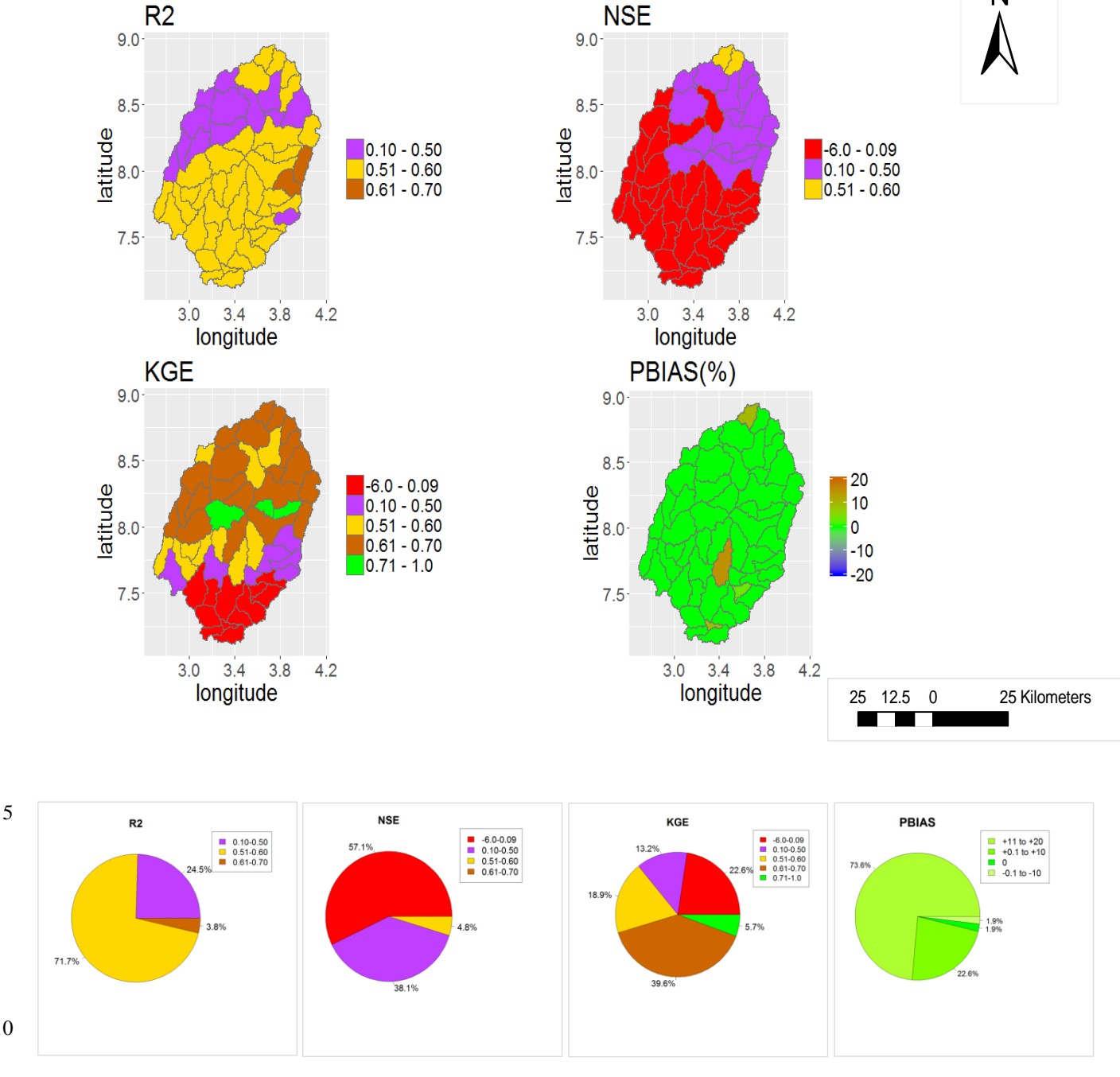

**Figure 6. Performance metrics (NSE, KGE, R², and PBIAS) of SWAT (SWAT_P-M) when calibrated with MOD16 (MS6)**

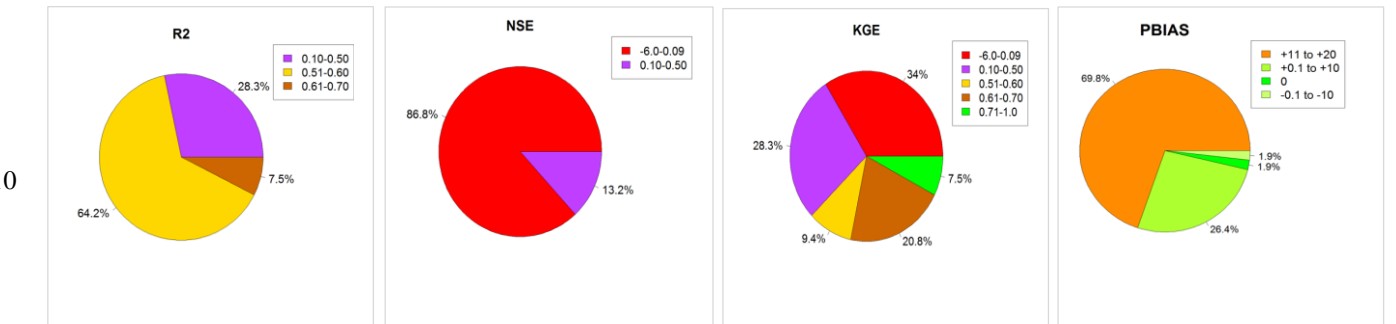

**Figure 7. Performance metrics (NSE, KGE, R², and PBIAS) result of SWAT (SWAT_P-M) when validated MOD16 (MS6)**

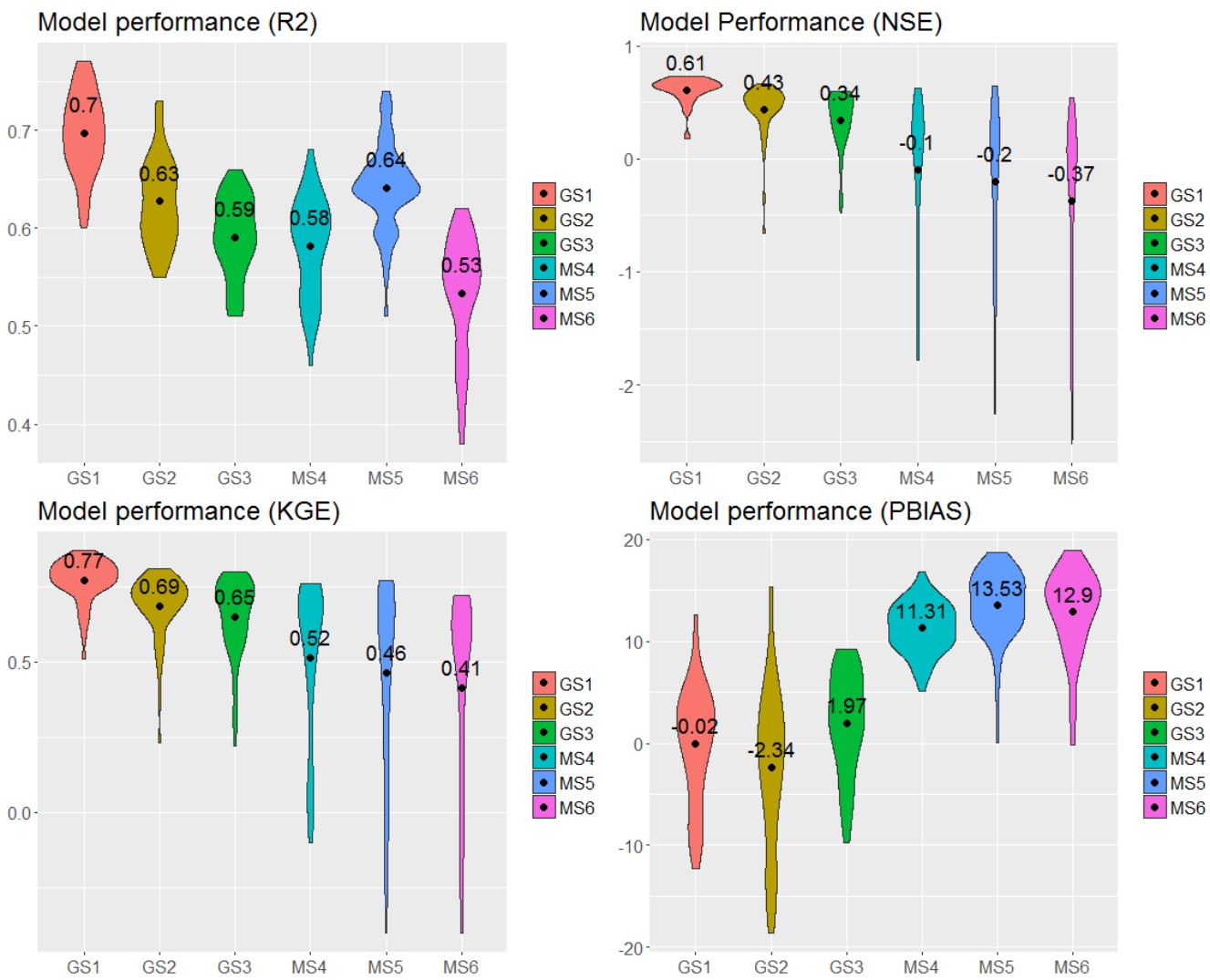

5    **Figure 8. The plots of the performance result of SWAT in simulating actual evapotranspiration. The values and the black dot symbol ("•") depicts the average value of, R², NSE, KGE and PBIAS obtained for each calibration.**

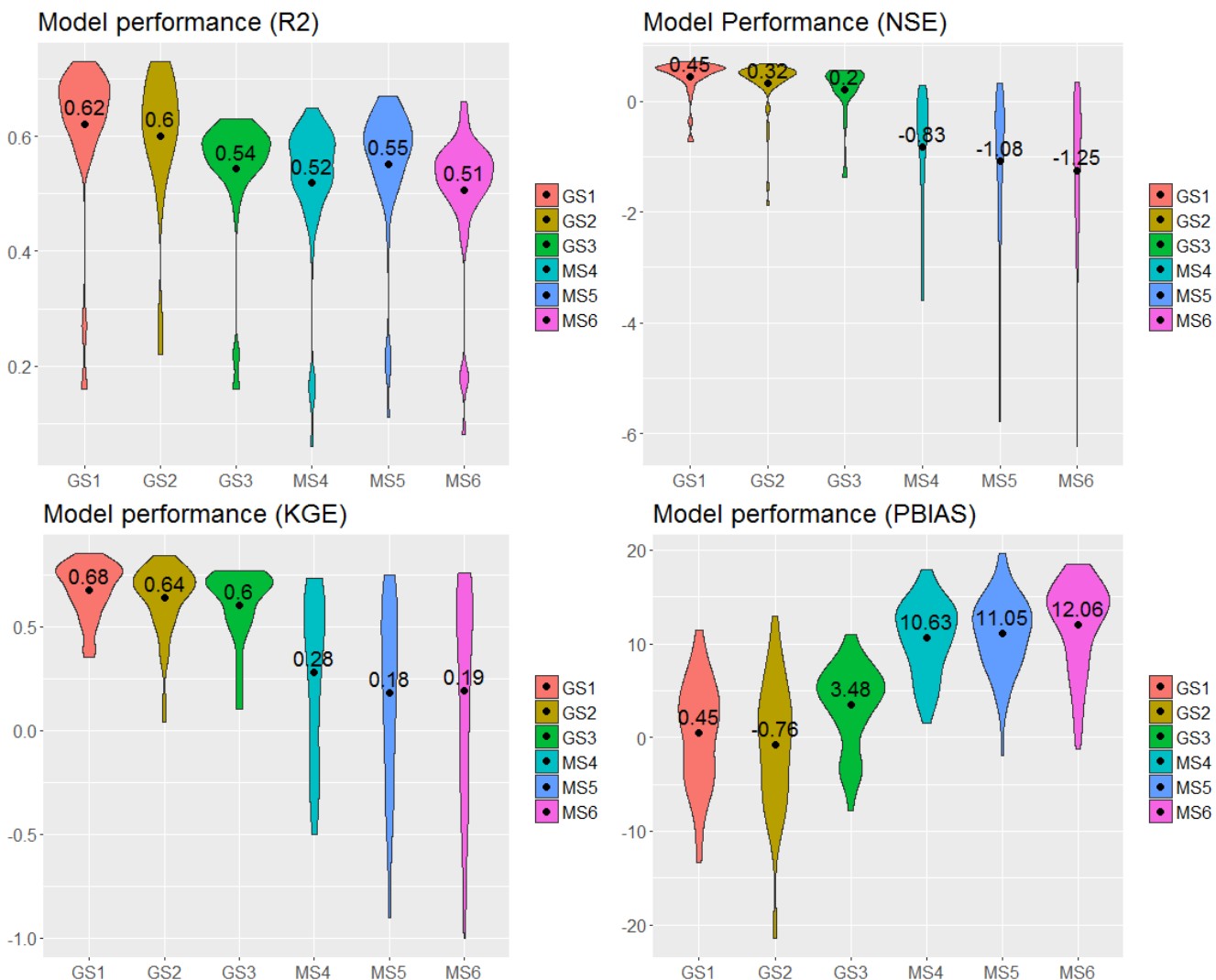

**Figure 9. The plots of the performance result of SWAT in simulating actual evapotranspiration. The values and the black dot symbol ("•") depicts the average value of, $R^2$, NSE, KGE and PBIAS obtained for each validation.**

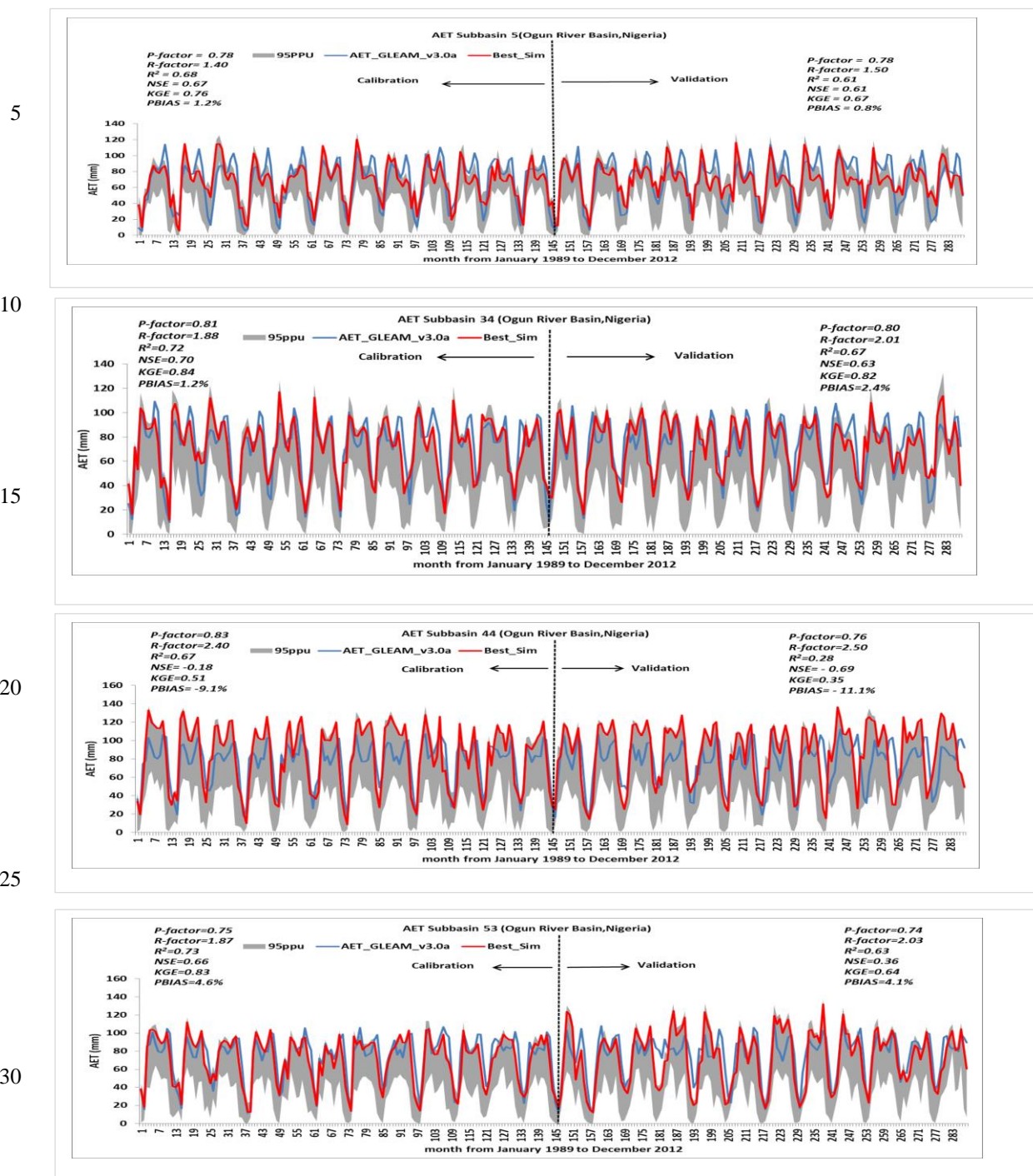

**Figure 10. Extracts of the monthly calibration and validation results (GSI) showing the 95% prediction uncertainty interval along with the best SWAT simulated actual evapotranspiration and the satellite based actual evapotranspiration (GLEAM-v3.0a).**

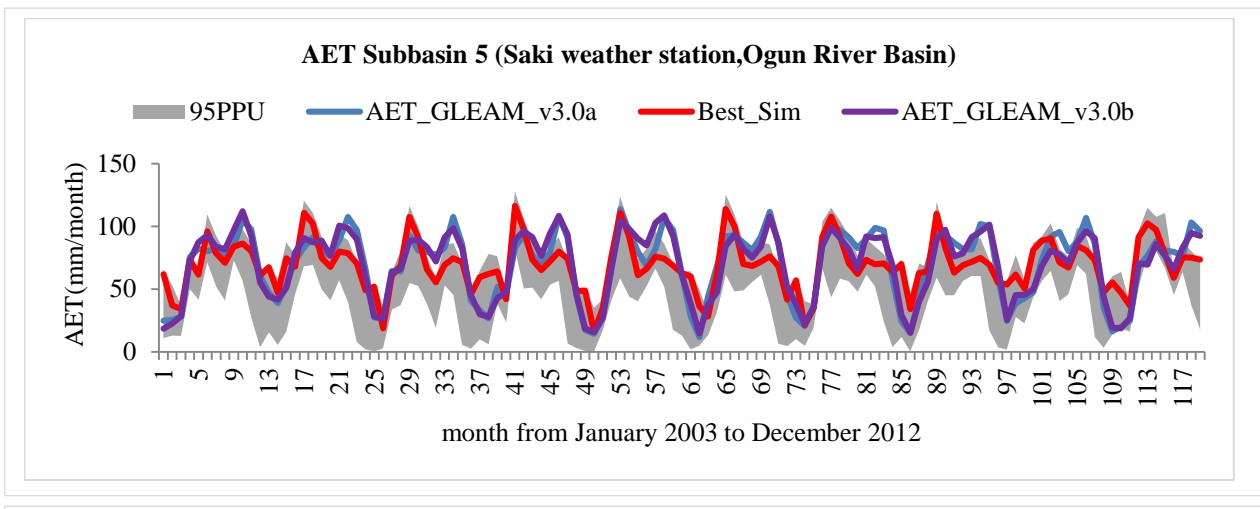

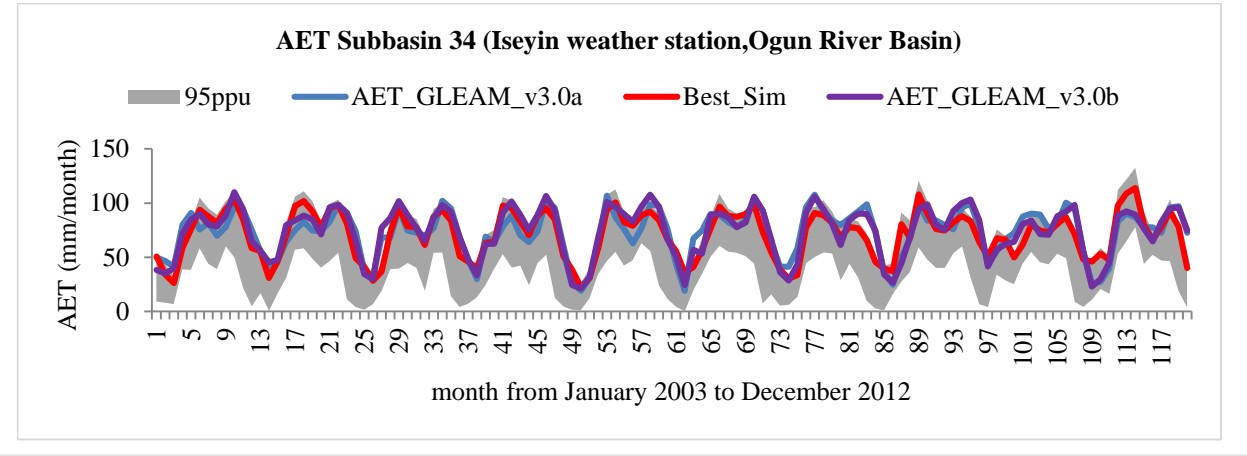

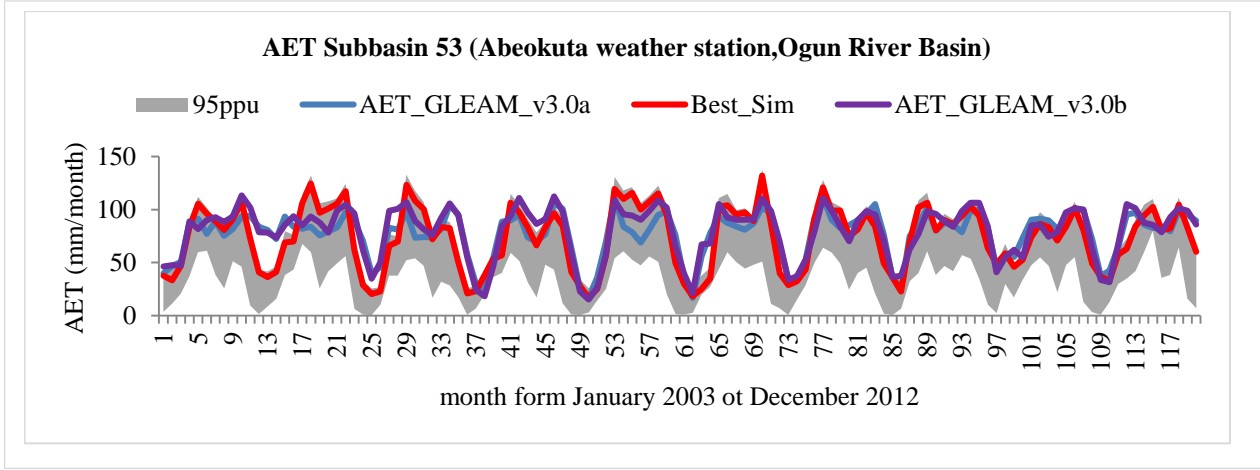

**Figure 11. SWAT model verification results showing the satellite based AET GLEAM_v3.0a used for the model calibration/validation, the best SWAT simulated actual evapotranspiration (GS1), and an independent GLEAM_v3.0b time series bracketed by 95% predictive uncertainty.**

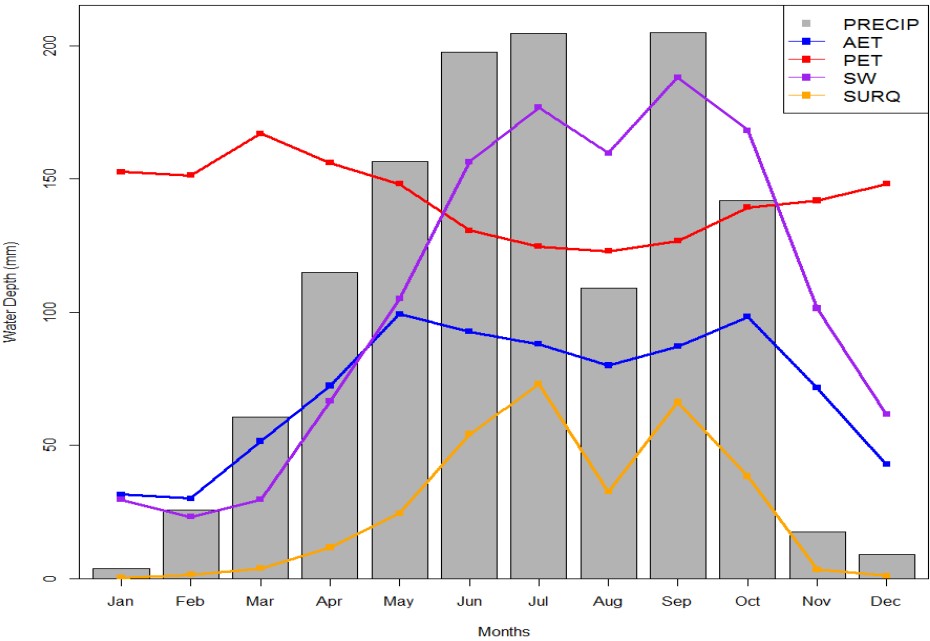

**Figure 12. Seasonal fluctuation of water balance components at the outlet of the watershed located in Abeokuta Town**

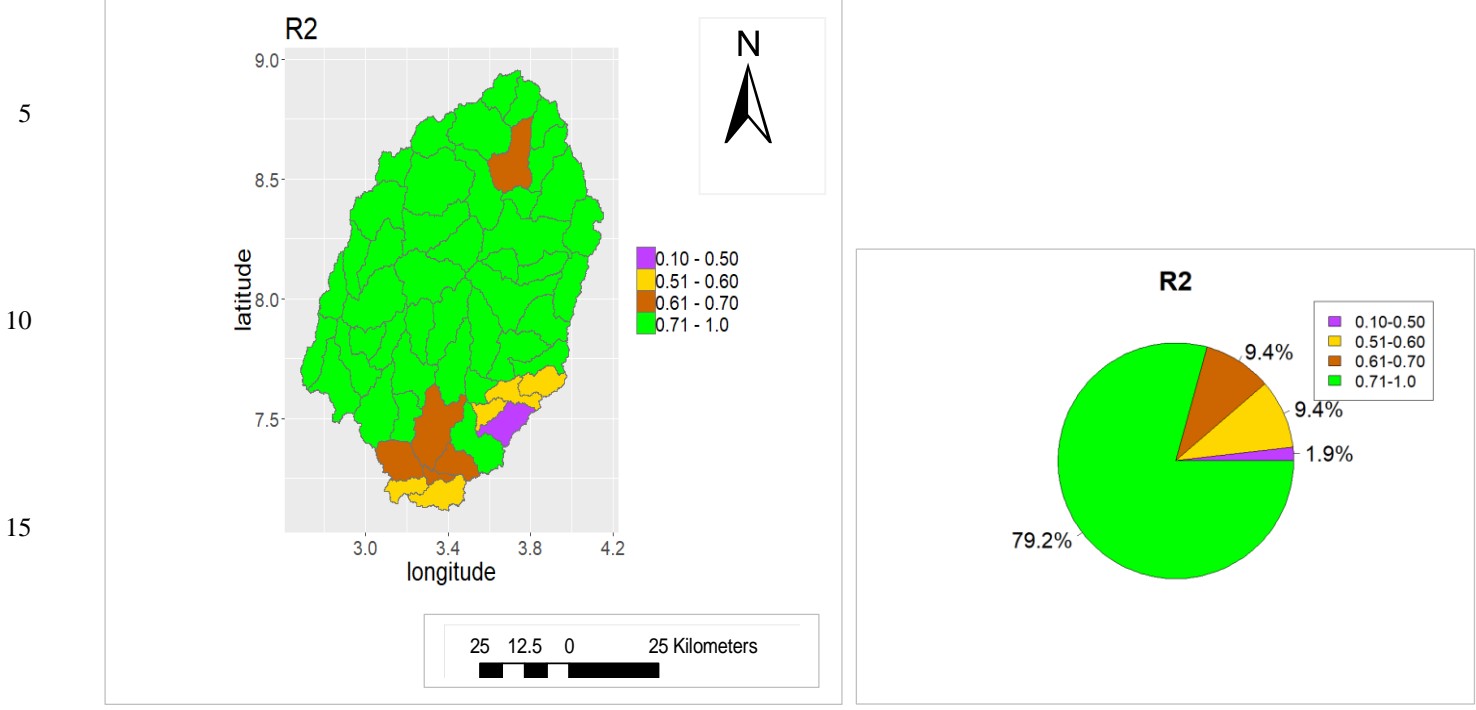

**Figure 13. Performance metric (R² ) of SWAT simulated soil moisture when validated with ESA CCI soil moisture v3.2**

 **Figure 14. Extracts of R² result of SWAT  monthly simulated soil moisture validation against ESA CCI SM v3.2 and the graphical representation of SWAT SM 95% prediction uncertainty band.**

**Appendix A: Performance metrics and their equations**

**Table A1. Table showing the performance metrics and their equations used to evaluate the model performance in this study.**

| Criterion | Mathematical equation | Description |
|-----------|----------------------|-------------|
| **$R^2$** | $$R^2 = \left( \frac{\sum_{i=1}^{n}(O_i - \overline{O})(P_i - \overline{P})}{\sqrt{\sum_{i=1}^{n}(O_i - \overline{O})^2 \sum_{i=1}^{n}(P_i - \overline{P})^2}} \right)^2$$ | The percent of variance explained by the model. It is a statistical measure of how close the data are to the fitted regression line. |
| **NSE** | $$NSE = 1 - \frac{\sum_{i=1}^{n}(O_i - P_i)^2}{\sum_{i=1}^{n}(O_i - \overline{O})^2}$$ | Quantifies the relative magnitudes of the residual variance (noise) compared to the observed data variance |
| **KGE** | $$KGE = 1 - \sqrt{(r-1)^2 + (\alpha-1)^2 + (\beta-1)^2}$$ | The goodness of fit measure provides an analysis of the relative importance of different components (correlation, bias and variability) in hydrologic simulation. |
| **PBIAS** | $$PBIAS = 100 \; x \frac{\sum_{i=1}^{n}(O_i - P_i)}{\sum_{i=1}^{n}O_i}$$ | The deviation of data being evaluated expressed in percentage. It measures the average tendency of the simulated data to be larger or smaller than the observation. Negative values indicate model overestimating (overprediction) and positive values indicate model underestimating (underprediction). |

$O_i$ are satellite based AET values; $P_i$ are simulated AET values; $\overline{O}$ are mean satellite based AET values; $\overline{P}$ are mean simulated AET values; r is the Pearson product correlation coefficient between satellite-based AET and the simulated AET, $\alpha$ is the standard deviation of the simulated AET over the standard deviation of the satellite-based AET, and β is the ratio of the mean simulated AET to the satellite based AET.

**Appendix B: SWAT model calibrated parameters**

**Table B1. Eleven parameters and their minimum and maximum range used in this study for the 1st iteration (1000 simulations) of all the six calibrations.**

| Parameter name | Minimum range | Maximum range |
|---|---|---|
| **v__ESCO.hru** | 0.00 | 1.00 |
| **v__EPCO.hru** | 0.00 | 1.00 |
| **v__CANMX.hru** | 0.00 | 100.00 |
| **v__GSI{2,4,5}.plant.dat** | 0.00 | 5.00 |
| **v__ALPHA_BF.gw** | 0.00 | 1.00 |
| **v__EVRSV.res________17,50** | 0.00 | 1.00 |
| **v__FFCB.bsn** | 0.00 | 1.00 |
| **r__CN2.mgt** | -0.25 | 0.85 |
| **r__SOL_AWC().sol** | 0.23 | 0.95 |
| **r__SOL_K().sol** | -0.06 | 0.95 |
| **r__SOL_BD().sol** | -0.41 | 0.95 |

**Maximum stomata conductance**

The canopy resistance term is modified to reflect the impact of high vapor pressure deficit on leaf conductance when calculating actual evapotranspiration (Stockle et al., 1992). The adjusted leaf conductance in which parameter GSI appears is calculated in SWAT using equation C1 and C2:

$$g_e = g_{e,mx} \times \left[1 - \Delta g_{e,dcl}(vpd - vpd_{thr})\right] \quad if \ vpd > vpd_{thr} \tag{C1}$$

$$g_e = g_{e,mx} \qquad\qquad\qquad if \ vpd \leq vpd_{thr} \tag{C2}$$

Where $g_e$ is the conductance of a single leaf (m s$^{-1}$); $g_{e,mx}$ is the parameter GS1 which is the maximum stomatal conductance of a single leaf ((m s$^{-1}$); $\Delta g_{e,dcl}$ is the rate of decline in leaf conductance per unit increase in vapor pressure

deficit (m s$^{-1}$ kPa$^{-1}$), vpd is the vapor pressure deficit (kPa), and vpd$_{thr}$ is the threshold vapor pressure deficit above which a plant will exhibit reduced leaf conductance (kPa). The rate of decline in leaf conductance per unit increase in vapor pressure is calculated by solving equation C1.

**The SCS curve number for soil moisture condition II**

Three antecedent moisture conditions are defined by SCS curve number: I -dry (wilting point), II -average moisture, and III -wet (field capacity). The SCS curve numbers II is calculated from either SCS moisture condition I or from SCS moisture III in equation C3 and C4:

$$CN_1 = CN_2 - \frac{20 \times (100 - CN_2)}{(100 - CN_2 + exp[2.533 - 0.0636 \times (100 - CN_2)])} \tag{C3}$$

$$CN_3 = CN_2 \times exp[0.00673 \times (100 - CN_2)] \tag{C4}$$

Where CN1 is the moisture condition I curve number; CN2 is the moisture condition II curve number, and CN3 is the moisture condition III curve number.

**Maximum canopy storage**

The maximum amount of water that can be held in canopy storage varies from day to day as a function of the leaf area index in SWAT model and is estimated with equation C5 in which CANMX parameter appears:

$$can_{day} = can_{mx} \times \frac{LAI}{LAI_{mx}} \tag{C5}$$

Where can$_{day}$ is the maximum amount of water than can be trapped in the canopy on a given day (mm H$_2$0), can$_{mx}$ is the CANMAX parameter and is the maximum amount of water than can be trapped in the canopy when the canopy is fully developed (mm H$_2$0), LAI is the leaf area index for a given day, and LAI$_{mx}$ is the maximum leaf area index for the plant.

**Bulk density**

Bulk density is calculated using equation C6:

$$\rho_b = \frac{M_S}{V_T} \tag{C6}$$

Where $\rho_b$ is the bulk density (Mg m$^{-3}$), Ms is the mass of solids (Mg) and VT is the total volume (m3). The total volume is calculated as:

$$V_T = V_A + V_W + V_S \tag{C7}$$

Where $V_A$ is the volume of air (m$^3$), $V_W$ is the volume of water (m$^3$), and $V_S$ is the volume of solids (m$^3$).

### Soil available water storage capacity

Soil available water storage capacity is calculated by subtracting the fraction of water present at permanent wilting point from that present at field capacity.

$$AWC = FC - WP \tag{C8}$$

Where AWC is the plant available water content, FC is the water content at field capacity, and WP is the water content at permanent wilting point.

### Saturated hydraulic conductivity

The equation in which the parameter saturated hydraulic conductivity (SOL_K) appears is given in C9:

$$TT_{perc} = \frac{SAT_{ly} - FC_{ly}}{K_{sat}} \tag{C9}$$

Where $TT_{perc}$ is the travel time for percolation (hrs), $SAT_{ly}$ is the amount of water in the soil layer when completely saturated (mm H$_2$0), $FC_{ly}$ is the water content of the soil layer at field capacity (mm H$_2$0), and $K_{sat}$ is the saturated hydraulic conductivity for the layer (mm h$^{-1}$).

### Baseflow alpha factor

The baseflow recession constant (Baseflow alpha factor) is $\alpha_{gw}$. The $\alpha_{gw}$ is calculated using equation C10:

$$\alpha_{gw} = \frac{1}{N} \times ln \left| \frac{Q_{gw,N}}{Q_{gw,0}} \right| \tag{C10}$$

Where $\alpha_{gw}$ is the ALPHA_BF parameter, N is the time lapsed since the start of the recession (days), $Q_{gw,N}$ is the groundwater flow on day N (mm H$_2$0), Qgw,0 is the groundwater flow at the of the start of the recession (mm H$_2$0).

### Lake evaporation coefficient

The equation in which the Reservoir evaporation coefficient (EVRSV.res) appears is shown in C11:

$$V_{evap} = 10 \times \eta \times E_0 \times SA \tag{C11}$$

Where $V_{evap}$ is the volume of water removed from the water body by evaporation during the day (m$^3$ H$_2$0), $\eta$ is an evaporation coefficient with a default value of 0.6 (EVRSV), $E_0$ is the potential evapotranspiration for a given day (mm H$_2$0), and SA is the surface a rea of the water body (ha).

### Plant uptake compensation factor

The equation in which plant uptake compensation factor (EPCO) appears (C13) is used to calculate the adjusted potential water uptake when the upper layers in the soil profile do not contain enough water to meet the potential water uptake (C12):

$$W_{up,ly} = W_{up,zl} - E_{up,zu} \tag{C12}$$

Where $W_{up,ly}$ is the potential water uptake for layer ly ( mm H$_2$0), $W_{up,zl}$ is the potential water uptake for the profile to the lower boundary of the soil layer (mm H$_2$0), $E_{up,zu}$ is the potential water uptake for the profile to the upper boundary of the soil layer (mm H$_2$0).

$$W'_{up,ly} = W_{up,ly} + W_{demand} \times epco \tag{C13}$$

Where $W'_{up,ly}$ is the adjusted potential water uptake for layer ly (mm H$_2$0), W$_{demand}$ is the water uptake demand not met by overlying soil layers (mm H$_2$0), and epco is the plant uptake compensation factor.

**Soil evaporation compensation coefficient**

The modified equation of the amount of evaporative demand for a soil layer which is determined by taking the difference between the evaporative demands calculated at the upper and lower boundaries of the soil layer incorporate a coefficient called ESCO for depth distribution modification. The modified equation is:

$$E_{soil,ly} = E_{soil,zl} - E_{soil,zu} \times esco \tag{C14}$$

Where $E_{soil,ly}$ is the evaporative demand or layer ly (mm H20), $E_{soil,zl}$ is the evaporative demand at the lower boundary of the soil layer (mm H20), $E_{soil,zu}$ is the evaporative demand at the upper boundary of the soil (mm H20) and esco is the soil evaporative compensation coefficient.

**Initial soil water storage expressed a fraction of field capacity water content**

The estimation of field capacity water content is the equation in which the initial soil water storage expressed as a fraction of field capacity water content (FFCB) appears. The equation is C15:

$$FC_{ly} = WP_{ly} + AWC_{ly} \tag{C15}$$

Where $FC_{ly}$ is the water content at field capacity expressed as a fraction of the total soil volume (FFCB), $WP_{ly}$ is the water content at wilting point expressed as a fraction of the total soil volume, and $AWC_{ly}$ is the available water capacity of the soil layer expressed as a fraction of the total soil volume.

**Appendix D: Ogun River Basin with its 53 subbasins intersected by the satellite-based AET pixels**

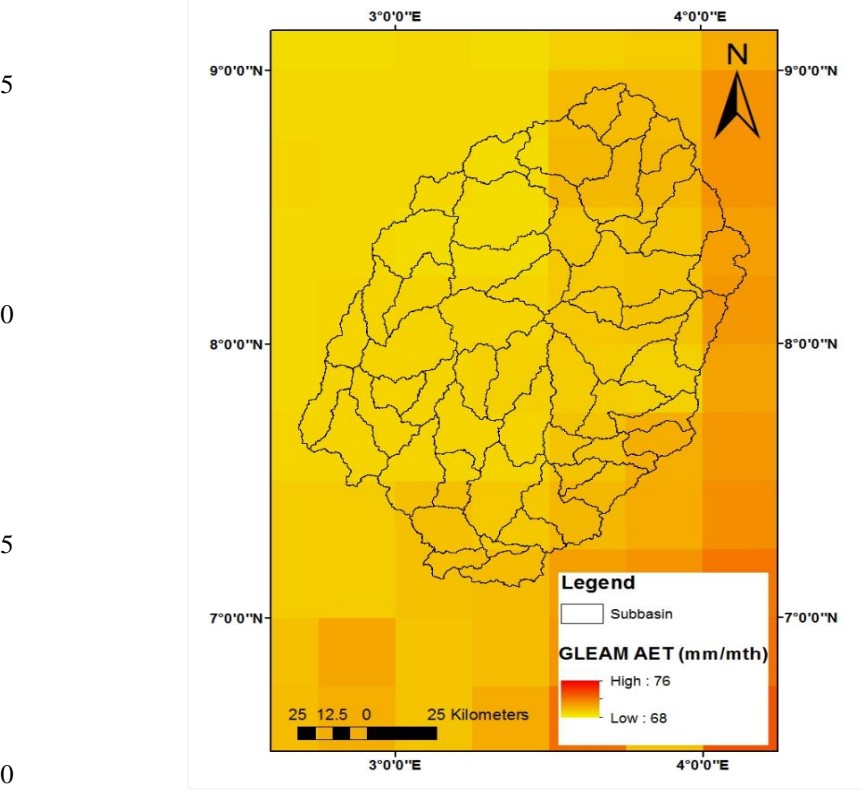

**Figure D1. Mean monthly actual evapotranspiration of GLEAM_v3.0a over the entire Ogun River Basin for 1989-2012.**

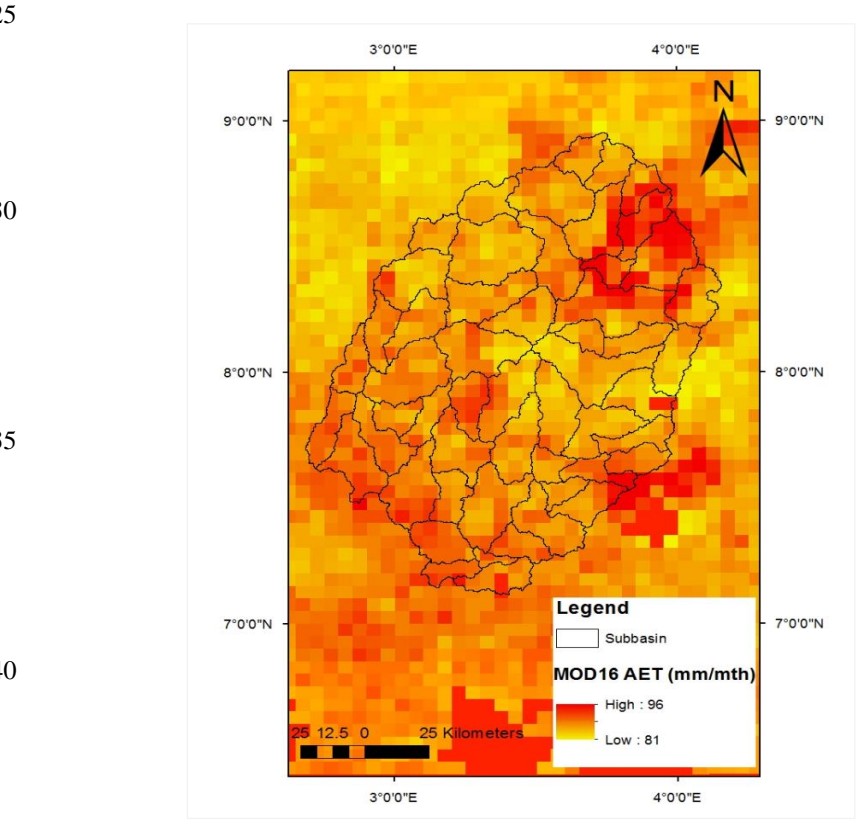

**Figure D2. Mean monthly actual evapotranspiration of MOD16 over the entire Ogun River Basin for 2000-2012.**

**Appendix E: Ogun River Basin with its 53 subbasins intersected by the ESA CCI soil moisture pixels**

**Figure E1. Mean monthly soil moisture of ESA CCI over the entire Ogun River Basin for 2001-2012.**