# Peer review of "Multi-site calibration and validation of SWAT with satellite-based evapotranspiration in a data-sparse catchment in southwestern Nigeria"

_Hydrology and Earth System Sciences, 2018_

## Referee Comment (RC1) · Anonymous Referee #1 · 29 Jun 2018

This paper calibrates the SWAT model using 2 available ET global products, a simple remote sensing ET equation (MOD16) and a more complex water balance model forced by remote sensing data (GLEAM).

Major comments: MOD16 does not explicitly account for transient water stress (as, say, derived from TIR data); how does this impact the results ? It is unclear to me whether the SWAT model used here uses the plant growth model. How is the vegetation taken into account ? Two additional important performance metrics are needed as a reference for the six calibrations: 1- A reference run with default (uncalibrated)

[Figure]

parameters – this is needed absolutely! 2- A focus on stressed/unstressed periods as defined by the GLEAM ET product, with metrics specific for each period; this would help analyse whether model improvement comes from a better ETP formulation or a better simulation of stress (S in GLEAM).

The description of the calibrated parameters (which, I assume, follow the SWAT terminology) is lacking: there is only a Table; equations showing where those paramterers appear should be provided in, say , an annex, to improve the paper standalone readability.

Minor comments:

Figure 2: why use an half-half split sample for MOD16 but only a 1/11 split sample for GLEAM ? Equation 5: the square root should extend to the third quadratic term. Page 10 Line 18: use the term "ratio" Page 13 Line 22: predicted > predict Page 15 Line 11: Runoff > Ruhoff ? Page 15 Line 33: "Therefore, the Heargraves ...periods": I don't understand this sentence

---

## Referee Comment (RC2) · Anonymous Referee #2 · 8 Jul 2018

General comments

The paper presents an interesting study on the calibration/validation/uncertainty analysis of the SWAT simulated actual evapotranspiration, based on freely available satellite data. The authors tested 2 different dataset and 3 different potential evapotranspiration models and evaluated their performances. They concluded that Hargreaves PET equation calibrated using the GLEAM_v3.0a data provided the most performing solution and gave a good level of confidence for using the SWAT model as a decision support tool. To my opinion the paper is worth to be published, but it needs some

major improvements for it to be sounder.

General comments

a) Literature Review: it lacks of significant contributions in the context of large-scale hydrological model simulation in data scarce area and it mainly focused on previous studies based on SWAT. It could be good to mention and discuss other approaches even if performed in different study areas but with the same problems (data scarce areas) (Kim et al., 2008; Kim and Kaluarachchi, 2009; Gebremicael et al., 2013; Tekleab et al., 2011; Abera et al., 2016 which applied a different hydrological modeling approach (Formetta et al., 2014))

b) I feel that the authors should acknowledge explicitly that the analysis presented needs to be tested against observed data and that the satellite data are them self based on modeling assumptions, which may or may not be plausible in some areas. Of course they provide a huge help and the way in which they are used in the paper nicely show it, but probably assuming them as "measured" can be misleading. At least can be specified once in the text that "measured AET" doesn't mean proper eddy-covariance data.

c) In the paper is claimed the importance of the Curve Number parameter but nothing is said about soil moisture evolution and runoff. I wonder why the authors do not use runoff-measured data as independent validation. This will show the effects of the different ET calibration on the runoff dynamic. The two processes are strongly related and the sensitivity of the CN parameter confirms this. This will be an important added value to the paper. Again, the authors claim: "The average long-term annual of the water balance at the outlet of the study area shows a satisfactory percentage error of closure". Is this referred to modeled data or modeled and measured? The use of measured streamflow data would help to better understand this part as well.

d) Because one of the main points in discussion/conclusion is the fact that: "Hargreaves equation had a superior model performance of the Penman Monteith and the

Priestly-Taylor" the authors should add their equations in the text. This would help to visualize the variables in input for each method, the variables that have been chosen for calibration and the variables that have been excluded.

Specific comments

Page 1 line 20: remove space in the number: River Basin (20 292 km2)

Page 1 line 21: "The novelty of the study is the use of freely available satellite derived AET data for calibration/validation of each of the SWAT delineated subbasins, thereby obtaining a better performing model at the local scale as well as at the whole watershed level": sounds like this is the first time the gleam dataset have been used to validate/calibrate swat, which is a strong sentence. May be in the study area?

Page 1 line 24: "Three different structures of the SWAT model were used in which each model structure was a set-up of SWAT with a different potential evapotranspiration (PET) equation": I would say that three different PET equations are tested: the model setup (in term of all the single components is the same except the pet)

Page 2: mechanistic, what the authors mean? Please explain.

Page 3 line 25: results showed a good Nash-Sutcliffe efficiency (NSE) and Coefficient of determination (R2) value for monthly average: quantify what good means for the authors and the values obtained.

Pag 5: The mean annual rainfall for the watershed is 1224 mm year-1 and the mean annual temperature is about 27o C. Mean annual potential evapotranspiration (PET) estimated by Hargreaves method (Hargreaves and Samani, 1985) is 1720 mm year-1 and the mean AET is about 692 mm year-1. Are this value based on measured or modeled data? Please specify it.

Page 5 typos: is 1224 mm year-1

Page 5: of 4103 ha, and please convert in km2 because all the other areas are in km

Page 7: 30 m spatial resolution digital elevation model (DEM), 17 soil classes, 17 landuse classes, 3 slope categories, meteorological data and landuse with its management (Table 1). Please specify if those data are available, from which web-site, and the accessed date.

Page 7 line 10: "The topHRU program allows the identification of a pareto-optimal threshold which minimizes the spatial error to 0.01 ha for a given number of HRUs and thereby minimizes the trade-off between SWAT computation time and number of HRUs. In this case, topHRU determined the optimum number of HRUs to be 1397 for the Ogun River basin. Thresholds of 0 ha for landuse, 150 ha for soil and 250 ha for slope were used in the SWAT set-up". What are the physical consequences of the thresholds? What happens if you use larger or lower values? How you define them?

Page 7: "delineated into 53 subbasins, with the main outlet in Abeokuta". Can you please give some summary statistics about them: min max average area, elevation, etc.

Daily precipitation 5 data (1984-2012) and minimum and maximum temperature data (1984-2012) at four weather stations (Fig. 1) were used as observed input data. Are you only using 4 stations for the whole basin (20292 km2)? Why not considering satellite products for a variable (precipitation), which sometimes could be even more important than etp? The authors should include this in the discussion.

The missing values of daily precipitation and minimum and maximum temperatures, along with solar radiation, wind speed and relative humidity were simulated by the ArcSWAT CSFR_World weather generator: it is clear that the ArcSWAT CSFR_World is used for gap filling of precipitation and temperature. The authors should specify: 1) How did you use the dataset for solar radiation, wind speed and relative humidity? 2) At which time resolutions are you specifying that input? 3) For which hydrological processes did you use these "simulated" forcing variables and how this affects your results?

Page 7 line 22-27: it sounds slightly repetitive: please consider to write the full sentence only of one model structure and to generalize for the other 2.

Page 8 line 10: please explain what are the main differences between the two dataset GLEAM_v3.0a and GLEAM_v3.0b and justify why you selected one of the two.

Page 8 line 25: "was implemented in SWAT-CUP. SWAT-CUP (Abbaspour, 2015)" move the citation when you firstly introduce SWAT-CUP.

Pages 8 line 28 to page 9 line 6: Please specify the parameter set that you started the sensitivity analysis with, at least the processes to which they are related. Moreover specify the list of the parameters that resulted sensitive and how you define a parameter as "sensitive".

Page 9 line 16: A metric among the six can be considered an objective function if it is optimized in the calibration procedure; it can be considered as goodness of fit metric if it is used to quantify how well or bad the model reproduce the measured data. Are those goodness of fit metrics? Which one of these six metrics has been optimized in the calibration procedure? Have you used all of them also as objective function? This is not fully clear.

Page 9 line 20 –Page 10 line 20: Consider to: i) just spell in the text the statistics used, their ranges and their optimal values and ii) move in appendix the explanation of each statistics because they are well known.

Page 10 line 23-26: please specify how the uncertainty is quantified: what are the parameters that are changed/sampled the LHS, what are their ranges?

Page 11: please show a figure of the river basin with the subbasin polygons and the pixel of MODIS and GLEAM. This will help the reader to understand how many pixels of MODIS and GLEAM cover your basins.

Each sub-basins has its own model AET. How did you choose the MODIS or GLEAM pixel to compare with and compute the NSE, R2, etc.

Page 11: figure 3,4,5, and 6 please consider to add a 4th class for the range of KGE and NSE (<0). This indicates where just the mean of the observed data will be more performing than the model itself.

Page 10 line 15: equation is not correct please revise it.

Page 11: figure 3, 4, 5, and 6 please add the North symbol and the scale bar in the maps.

Page 11 line 23: "The results of global sensitivity analysis revealed that the SCS runoff": please specify from where the reader can see this.

Page 12: all the page can be summarized by just one figure reporting on the x-axis the model configurations and on the y axis the percentage of sub-basin for a given class of the goodness of fit index (NSE, R2, etc).

Page 13 the paper goes from section 3 to subsection 3.3: 3.1 and 3.2 are missing.

Page 14: increases in February from 55mm to 76mm as the space between 55 and mm

Page 15 line 5: "Using the guidelines in Moriasi et al. (2007, 2015) and Kouchi et al. (2017) for" probably these guidelines were drawn for runoff? Is it correct to use it for others hydrological processes? Is this been done in the past? If yes, please add a citation otherwise just clarify this aspect.

Page 15 line 20: "From our results, we agree that the AET from MOD16 tends to overestimate AET". Overestimate against what? This is a strong statement mainly because there is not direct comparison against measured AET data

Reference

Kim, U. and Kaluarachchi, J. J: Climate Change Impacts on Water Resources in the Upper Blue Nile River Basin, Ethiopia, J. Am. Water Resour. As., 45, 1361–1378, 2009.

Kim, U., Kaluarachchi, J. J., and Smakhtin, V. U.: Generation of Monthly Precipitation Under Climate Change for the Upper Blue Nile River Basin, Ethiopia, J. Am. Water Resour. As., 44, 1231– 1247, 2008.

Gebremicael, T., Mohamed, Y., Betrie, G., van der Zaag, P., and Teferi, E.: Trend analysis of runoff and sediment fluxes in the Upper Blue Nile basin: A combined analysis of statistical tests, physically-based models and landuse maps, J. Hydrol., 482, 57– 68, 2013. Tekleab, S., Uhlenbrook, S., Mohamed, Y., Savenije, H. H. G., Temesgen, M., and Wenninger, J.: Water balance modeling of Upper Blue Nile catchments using a top-down approach, Hydrol. Earth Syst. Sci., 15, 2179–2193, https://doi.org/10.5194/hess-15- 2179-2011, 2011

Abera, W., Formetta, G., Brocca, L., and Rigon, R.: Modeling the water budget of the Upper Blue Nile basin using the JGrass-NewAge model system and satellite data, Hydrol. Earth Syst. Sci., 21, 3145-3165, https://doi.org/10.5194/hess-21-3145-2017, 2017.

Formetta, G., Antonello, A., Franceschi, S., David, O., & Rigon, R. (2014). Hydrological modelling with components: A GIS-based open-source framework. Environmental Modelling & Software, 55, 190-200.

---

## Author Comment (AC1) · 26 Jul 2018

**Reply to comments of Anonymous Referee #1**

*We thank anonymous referee #1 for reviewing our manuscript. We are especially grateful for the many insightful comments and valuable suggestions, as these comments will lead us to improve the paper. We have to the best of our abilities responded to them and address the referee's comments in the following point by point response. Note the following conventions: RC = referee comments, AC = authors comments (replies) printed in italic.*

**Major comments**

RC 1: This paper calibrates the SWAT model using 2 available ET global products, a simple remote sensing ET equation (MOD16) and a more complex water balance model forced by remote sensing data (GLEAM). MOD16 does not explicitly account for transient water stress (as, say, derived from TIR data); how does this impact the results?

*AC1: We agree with the referee that MOD 16 does not explicitly account for transient water stress because it is not directly derived from Thermal Infrared Remote sensing (TIR) data. Some of the reasons for not fully using TIR data at the global scale for the MOD16 product are: a) a changing relationship between TIR based land surface temperature (LST) and NDVI when moving from mid to high latitudes; b) LST as derived from TIR is not equal to the aerodynamic surface temperature (which is driving the sensible heat flux), potentially leading to non-accurate ET estimations under various conditions (Mu et al., 2007, 2013). MOD16 applies the Penman-Monteith (PM) equation to calculate ET on a global scale by using variables and parameters needed from VIS/NIR remote sensing (land cover, LAI, albedo, FPAR) and from daily meteorological reanalysis data (radiation, $T_{air}$, pressure, rel. humidity; NASA's global modeling and assimilation office, GMAO). In principle, the surface resistance ($r_S$) parameter in the PM equation accounts for any direct effect on ET due to limitations in available water. The MOD16 scheme however, does not include any soil water content data directly. The way $r_S$ is derived in the MOD16 scheme only considers an indirect effect via a non-linear dependency of $r_S$ with the water vapor pressure deficit (VPD) in the atmosphere. VPD under daytime conditions often represents a proxy for soil moisture conditions and therefore $r_S$. The impact of the not-explicit consideration of transient water stress in the MOD16 product on our SWAT-model calibration is difficult estimate.*

*Transient water stress is not a main challenge in the present study area, which is located in the humid region of south western Nigeria with a mean Aridity Index of 0.75 from the period 1989 to 2012 (A.I. > 0.65 value, which is considered to be a humid region; UNEP, 1997).*

*We also reviewed the literature of the MOD16 ET with measured (EC) flux data at sites climatically similar to our catchment and found an agreement between our catchment and the result obtained between MOD16 ET and the measurements for a study area in the tropical region, Brazil (PDG site in the Rio Grande Basin) conducted by Ruhoff et al. (2013) also located in a tropical region. While not being comprehensive, the comparisons were an indication that MOD16 behave similarly having a positive PBIAS (MOD 16 overestimating AET in both sites). Also, Trambauer et al. (2014) compared different evaporation products for Africa, in their paper they stated that MOD16 evaporation do not show a good agreement with other products in most part of Africa, while the rest (GLEAM, ERAL, PCR-GLOBWB hydrological model simulated AET) are more consistence.*

*We will fully indicate this comparison in the manuscript and will discuss this point in more detail in the revised version of the manuscript, and also provide the same level of detail for the description of ET in the GLEAM product.*

RC2: It is unclear to me whether the SWAT model used here uses the plant growth model.

*AC2: The Soil and Water Assessment Tool (SWAT) is an eco-hydrological model that uses at its core the plant growth model EPIC (Williams et al., 1989) that is able to simulate the growth (including nutrient and water uptake) of many types of crops and trees as land cover. The plant growth component of SWAT is a simplified version of the EPIC plant growth model. We will include this sentence in the paper.*

RC2b: How is the vegetation taken into account?

*AC2b: SWAT is a physically based model that requires a land use map as one input data source that represents the spatial distribution of vegetation in the watershed. SWAT categorizes plants into seven different types: warm season annual legume, cold season annual legume, perennial legume, warm season annual, cold season annual, perennial and trees.*

*Plant growth is modeled by simulating leaf area development, light interception and conversion of intercepted light into biomass assuming a plant species-specific radiation-use efficiency. Hence, In SWAT, phenological plant development is based on daily accumulated heat units.*

*The plant growth model is used to assess removal of water and nutrients from the root zone, transpiration, and biomass/yield production. In SWAT the plant growth can be inhibited by a minimum or maximum temperature, available water, nitrogen and phosphorus stress. The potential biomass is based on a method developed by Monteith in 1977 (a radiation model, that uses solar radiation as its decisive factor of crop production, while temperature and water are two other important factors); and a harvest index is used to calculate the final yield. We will include this sentence in the revised manuscript for more clarity.*

RC3: Two additional important performance metrics are needed as a reference for the six calibrations:

1. A reference run with default (uncalibrated) parameters – this is needed absolutely!
2. A focus on stressed /unstressed periods as defined by GLEAM ET product, with metric specific for each periods; this would allowed help analyze whether model improvement comes from better ETP formulation or a better simulation of stress.

*AC3: 1. the referee is right to point out that a reference run with uncalibrated parameters is needed. We agree with the referee and will add the reference scenario results in the revised manuscript.*

*AC3: 2. it is true that stressed /unstressed periods should be considered. We included both dry and the wet periods of growth in our calibration and validation. In this paper, we followed the split-sample test as presented by Klemes (1986), which is a model calibration and validation approach that consists of equally splitting the available data, when the record is sufficiently long to represent different climatic conditions. Also, Gan et al. (1997) stated that data are most frequently split by time periods, carefully ensuring that the climate data used for both calibration and validation are not substantially different, i.e., wet, moderate, and dry years occur in both periods. In this study, this we achieved by comparing the absolute AET values of the calibration and validation years, with the aim of having a minimal difference by including wet, moderate, and dry years in both periods. We further checked the linear*

*regression of the calibration and the validation years. With these two indices, we successfully included wet, moderate and dry years in each calibration and the validation periods. Since the now newly added reference run will include both dry and wet periods, we believe having another separate performance metric accounting for stressed/unstressed periods will be of no need.*

RC4: The description of the calibrated parameters (which, I assumes, follow the SWAT terminology) is lacking: there is only a Table; equations showing where those parameters appear should be provided in, say, an annex, to improve the paper standalone readability.

*AC4: We believe that the list of the calibrated parameters and where they can be found, as well as the description of the selected parameters and their calibrated optimal values, are of importance and these are provided in Table 2. Including the equations showing where these parameters appear will be too ambitious, because several different equations are formulated for different hydrological conditions for most of the parameters, and this information will be difficult to present in a tabular form. We agree to direct the reader to SWAT documentation by stating that:*

*"For further reading on the model equations in which the calibrated parameters in SWAT appear, the reader is referred to the in SWAT theoretical documentation version 2009 by Nietsch et al. (2011) (http://swatmodel.tamu.edu)". We also agree to add equations related to the evapotranspiration estimations in the manuscript*

**Minor comments:**

RC5: Figure 2: why use a half-half split sample for MOD16 but only a 1/11 split sample for GLEAM?

*AC5: The referee is right to point out this issue and we agreed that this needs clarification in the manuscript. MOD16 is a global dataset spanning the 13-year period 2000-2012 and the splitting of calibration period (2000-2006) and validation period (2007-2012) followed the split-sample test as presented by Klemes (1986) and Gan et al. (1997). While the GLEAM_v3.0a is a global dataset spanning the 35-year period 1980-2014. For this study, we used GLEAM_v3.0a dataset spanning 24-year period 1984-2012 because the SWAT simulation output was from 1989-2012. The splitting of calibration period (1989-2000) and validation period (2001-2012) for GLEAM_v3.0a also followed the split-sample test as presented by Klemes (1986) and Gan et al. (1997). The splitting by time periods was carefully done by ensuring that both MOD16 and GLEAM AET available years dataset used for calibration and validation are not substantially different, i.e., wet, moderate, and dry years occur in both periods.*

RC6: Equation 5: the square root should extend to the third quadratic term.

*AC6: We agree with the referee and we will make the suggested changes.*

RC7: Page 10 line 18: use the term "ratio"

*AC7: Thank you! We will make the change on page 10 line 18.*

RC8: Page 13 line 22: predicted>predict

*AC8: Thank you! We will make the change on page 13 line 22*

RC9: Page 15 line 11: Runoff > Ruhoff?

*AC9: Thank you! We will make the change on page 15 line 11*

RC10:  Page 15 line 33: Therefore, the Hargreaves…….periods": I don't understand this sentence

*AC9: We agree with the referee that the sentence needs clarification. We will make the changes on page 15 from lines 28-33 to read in the following way:*

*"The Hargreaves PET equation uses ground measured minimum and maximum temperature to estimate the corresponding AET, whereas GLEAM uses an algorithm to convert PET into AET using a multiplicative evaporation stress factor (S). The derivation of S is based on microwave observation of the vegetation optical depth that is used as a proxy for the vegetation water content and used in simulations of root zone soil moisture. Therefore, the Hargreaves equation in SWAT directly uses measured data and radiation estimates of the area, whereas GLEAM algorithms separately estimate different component of terrestrial evaporation, which consist of evaporation, soil-water balance, stress and rainfall interception (Martens et al., 2016).*

*To this end, the good GS1 model performance is attributed to the use of Hargreaves equation (that uses the measured temperature data) and the complex water balance model algorithm of GLEAM that takes into account soil-water balance, bare-soil evaporation and open water evaporation, stress and rainfall interception. All of these components have temporal and spatial variations and assist in simulating the dynamic hydrological process, especially the AET, which is the variable of most concern in this study."*

Reference

Gan, T. Y., Dlamini, E. M., and Biftu, G. F.: Effects of model complexity and structure, data quality, and objective functions on hydrologic modeling. J. Hydrol. 192(1): 81-103, 1997.

Klemes, V.: Operational testing of hydrological simulation models. Hydrological Sciences Journal, v. 31, n. 1, p. 13-24, doi: 10.1080/02626668609491024, 1986.

Courault, D., Seguin,B., and  Olioso, A. : Review on estimation of evapotranspiration from remote sensing data: From empirical to numerical modeling approaches, Irrig. Drain. Syst., 19, 223–239      , 2005.

Martens, B., Miralles, D., Lievens, H., Fernández-Prieto, D., and Verhoest, N.: Improving terrestrial evaporation estimates over continental Australia through assimilation of SMOS soil moisture, Int. J. Appl. Earth Obs., 48, 146–162, 2016.

Monteith, J.L.: Climate and the efficiency of crop production in Britain. Philosophical Transactions of the Royal Society of London (Series B-Biological Sciences), 281, 277–294, 1977.

Moran, M. S.: Thermal infrared measurement as an indicator of plant ecosystem health, in Thermal Remote Sensing in Land Surface Processes. Taylor and Francis, Philadelphia, Pa, pp. 257–282, 2003.

Neitsch, S., Arnold, J., Kiniry, J., and Williams, J.: Soil & Water Assessment Tool Theoretical Documentation Version 2009, Texas Water Resour. Inst., 1–647, doi:10.1016/j.scitotenv.2015.11.063, 2011.

UNEP: United Nation Environmental Programme, world atlas of desertification 2ED.UNEP, London, 1997

Ruhoff, A. L., Paz, A. R., Aragao, L. E. O. C., Mu, Q., Malhi, Y., Collischonn, W., Rocha, H. R. and Running, S. W.: Assessment of the MODIS global evapotranspiration algorithm using eddy covariance measurements and hydrological modelling in the Rio Grande basin, Hydrol. Sci. J., 58(8), 1658–1676,

doi:10.1080/02626667.2013.837578, 2013.

Trambauer, P., Dutra, E., Maskey, S., Werner, M., Pappenberger, F., Van Beek, L. P. H. and Uhlenbrook, S.: Comparison of different evaporation estimates over the African continent, Hydrol. Earth Syst. Sci., 18(1), 193–212, doi:10.5194/hess-18-193-2014, 2014.

Williams J. R., Jones, C. A., Kiniry, J. R., and Spanel, D. A.: The EPIC Crop Growth Model, Transactions of the American Society of Agricultural Engineers, Vol. 32, No. 2, pp. 497-511, 1989.

---

## Author Comment (AC2) · 3 Aug 2018

**Reply to comments of Anonymous Referee #2**

*We thank anonymous referee #2 for reviewing our manuscript and for the positive comment. We are especially grateful for the many constructive comments and valuable suggestions, as these comments will lead us to improve the paper. We have to the best of our abilities responded to them and address the referee's comments in the following point by point response. Note the following conventions: RC = referee comments, AC = authors comments (replies) printed in italic.*

**Major comments**

RC (a): Literature Review: it lacks significant contributions in the context of large-scale hydrological model simulation in data scarce area and it mainly focused on previous studies based on SWAT. It could be good to mention and discuss other approaches even if performed in different study areas but with the same problems (data scarce areas) (Kim et al., 2008; Kim and Kaluarachchi, 2009; Gebremicael et al., 2013; Tekleab et al., 2011; Abera et al., 2016 which applied a different hydrological modeling approach (Formetta et al., 2014))

*AC (a): Agreed. We will add other related references and discuss other approaches in the context of large-scale hydrological model simulation in data scarce area as suggested.*

RC (b): I feel that the authors should acknowledge explicitly that the analysis presented needs to be tested against observed data and that the satellite data are them self-based on modeling assumptions, which may or may not be plausible in some areas. Of course, they provide a huge help and the way in which they are used in the paper nicely show it, but probably assuming them as "measured" can be misleading. At least can be specified once in the text that "measured AET" doesn't mean proper eddy-covariance data

*AC (b): We thank the reviewer for this suggestion. In the revised manuscript, we will strengthen the fact that we only make use of two satellite derived AET products. We will emphazise that we do not have any e.g. EC-based local measurments within our catchment. However, the satellite products have been tested elsewhere and we will briefly summarize study results that are relevant to our study (similar climate conditions) in a revised version of the manuscript. .*

RC (c): In the paper is claimed the importance of the Curve Number parameter but nothing is said about soil moisture evolution and runoff. I wonder why the authors do not use runoff-measured data as independent validation. This will show the effects of the different ET calibration on the runoff dynamic. The two processes are strongly related and the sensitivity of the CN parameter confirms this. This will be an important added value to the paper. Again, the authors claim: "The average long-term annual of the water balance at the outlet of the study area shows a satisfactory percentage error of closure". Is this referred to modelled data or modelled and measured? The use of measured streamflow data would help to better understand this part as well.

*RC (c): It is true that we emphases importance on the curve number parameter because it is found to be sensitive for all the six calibrations. As suggested by the referee, we agree, to mention and discuss other parameters relating to soil moisture and runoff considered in the calibration process.*

*We agree that AET calibration and runoff dynamics are strongly related. In the manuscript, we do not consider runoff-measured data as independent validation because it is not available for the study area and that is main reason we considered AET derived from satellite products as an alternative option for SWAT hydrologic model calibration. We believe using a freely available AET products (GLEAM & MOD16), that have been heavily tested in the past by calibration and validation experiments by a large number of scientist/research groups is one solution in setting up a hydrological model that will be used*

*as a decision support tool in such a data scarce region. These points will be emphasis in the revised manuscript.*

RC (d): Because one of the main points in discussion/conclusion is the fact that: "Hargreaves equation had a superior model performance of the Penman Monteith and the Priestly-Taylor" the authors should add their equations in the text. This would help to visualize the variables in input for each method, the variables that have been chosen for calibration and the variables that have been excluded.

*AC (d): We will include the Hargreaves, Penman-Monteith and Priestly-Taylor PET equations in the paper for visualizing of each variables input for each method.*

**Specific comments:**

RC1:  Page 1 line 20: remove space in the number: River Basin (20 292 km2)

*AC1: Thank you for this point. Although we haven't make the change yet, because checking the HESS manuscript preparation guidelines for authors, it is mentioned in the last sentence (h.) under heading "physical dimensions and units", that:*

*Numerals should also be typeset using upright fonts. The symbol for the decimal marker is the dot. To facilitate reading, numbers may be divided in groups of three using a thin space (e.g. 12 345.6), starting with the ten-thousand digit. Neither dots nor commas are permitted as group separators*

RC2:  Page 1 line 21: "The novelty of the study is the use of freely available satellite derived AET data for calibration/validation of each of the SWAT delineated subbasins, thereby obtaining a better performing model at the local scale as well as at the whole watershed level": sounds like this is the first time the gleam dataset have been used to validate/calibrate swat, which is a strong sentence. May be in the study area?

*AC2: We agree with the referee that it is a strong sentence and we acknowledge that the novelty of the study needs further clarification. To this effect we will rewrite the statement in the revised manuscript to read as follows:*

*"The novelty of the study is the use of two freely available satellite derived AET dataset to calibrate/validate SWAT setups using three different estimated AET method for each of the 53 SWAT delineated subbasins, to improve hydrological model performance for a data scarce watershed in Nigeria."*

*Although the 3 different PET methods and the corresponding actual evapotranspiration estimate in SWAT have been tested (Wang et al. (2006), Franco and Bonumá (2017), Samadi (2017), Ha et al. (2018)), but study of calibrating each of the 3 SWAT estimated AET with two different freely available remotely sensed derived AET for each delineated subbasin within SWAT framework in order to know the highest performing model for a particular region/basin is yet unknown.*

*We believe this is a new contribution both to research community and in the study area.*

RC3:  Page 1 line 24: "Three different structures of the SWAT model were used in which each model structure was a set-up of SWAT with a different potential evapotranspiration (PET) equation": I would

say that three different PET equations are tested: the model setup (in term of all the single components is the same except the pet).

*AC3: We agree with the referee. As suggested, we will make the changes throughout the manuscript.*

RC4:  Page 2: mechanistic, what the authors mean? Please explain.

*AC4: We agree with the referee that the terminology needs further clarification in the manuscript. In the context of this study mechanistic model such as of SWAT (Arnold et al., 1998), SHETRAN (Ewen et al., 2000), HSPF (Bicknell et al., 1997) means structural models that aim to describe which driving processes are present in a system and are able to make detailed predictions in both time and space. Hence, mechanistic is meant in a sense that relevant processes are described in some detail using physical and geochemical principles and process descriptions. The added statement will be inserted for more clarification in the revised manuscript.*

RC5:  Page 3 line 25: results showed a good Nash-Sutcliffe efficiency (NSE) and Coefficient of determination (R2) value for monthly average: quantify what good means for the authors and the values obtained.

*AC5: We agree with the referee that the statement needs further quantification and clarification. To this effect we will rewrite the statement in the revised manuscript to read as follows:*

*"The model results showed a good Nash-Sutcliffe efficiency (NSE) of 0.72 and Coefficient of determination ($R^2$) of 0.76 during the calibration periods.  For the validation periods, a good model performance result showing $R^2$ of 0.71 and NSE of 0.78 values for monthly average streamflow were also obtained.*

RC6:  Pag 5: The mean annual rainfall for the watershed is 1224 mm year-1 and the mean annual temperature is about 27o C. Mean annual potential evapotranspiration (PET) estimated by Hargreaves method (Hargreaves and Samani, 1985) is 1720 mm year-1 and the mean AET is about 692 mm year-

*AC6: Many thanks for highlighting this points for clarification. We will rewrite the statement in the revised manuscript to read as follows:*

*The mean annual rainfall (1984-2012) obtained from measured data of Ogun watershed is 1224 mm $yr^{-1}$ and the mean annual temperature obtained from measured data is about 27º C. Mean annual potential evapotranspiration (PET) estimated by Hargreaves method (Hargreaves and Samani, 1985) using measured minimum and maximum temperature is 1720 mm $yr^{-1}$ and the mean AET derived from SWAT output(1989-2012) for this study area is  692 mm $yr^{-1}$.*

 RC7: Page 5 typos: is 1224 mm year-1

*AC7: Thank you. We will make the corrections throughout the manuscript.*

RC8: Page 5: of 4103 ha, and please convert in km2 because all the other areas are in km

*AC8: Thank you. We will make the changes as suggested in the revised manuscript*

RC9: 30 m spatial resolution digital elevation model (DEM), 17 soil classes, 17 landuse classes, 3 slope categories, meteorological data and landuse with its management (Table 1). Please specify if those data are available, from which web-site, and the accessed date

*AC9: Thank you. We will add the additional information to Table 1 in the revised manuscript as suggested.*

RC10: Page 7 line 10: "The topHRU program allows the identification of a pareto-optimal threshold which minimizes the spatial error to 0.01 ha for a given number of HRUs and thereby minimizes the trade-off between SWAT computation time and number of HRUs. In this case, topHRU determined the optimum number of HRUs to be 1397 for the Ogun River basin. Thresholds of 0 ha for landuse, 150 ha for soil and 250 ha for slope were used in the SWAT set-up". What are the physical consequences of the thresholds? What happens if you use larger or lower values? How you define them?

*AC10: As explained in the manuscript SWAT uses HRUs, whereby a watershed is subdivided into homogenous hydrologic response units having unique soil, slope, and land use properties as the basic unit of all SWAT model calculations. For SWAT, threshold specification of landcover, soil and slope is allowed and the physical consequences of the thresholds is to improve the computational efficiency of simulations while keeping key landscape features and information of a watershed in the hydrologic modelling. In the paper, we selected thresholds of 0 ha for landuse, 150 ha for soil and 250 ha for the slope. This means that HRUs should be created for all the area occupied by the landuse classes. For soil it means that any homogenous soil class occupying less than 150 ha should not be considered when determining HRUs. For slope, it means that any homogenous slope class that occupies less than 250 ha should not be considered. This allows us to define how detail the watershed will be represented by selecting the desired threshold values.*

*If we select larger values, then we eliminate key landscape features and their processes out of the system which may lead to considerable loss of information about the watershed landscape, resulting in model output that are less representative of the watershed as a whole and if we select lower values then we retain as many landscape features (spatial data) in the model thereby increasing the computational time of SWAT.*

*Therefore, we defined this threshold by selecting the desired threshold values using topHRU tool and its concept as explained in the paper while minimizing the spatial error to 0.01 ha for a given number of HRUs. Moreover, the criterial for selecting 0 ha for landuse was based on our desired to retain all the landuse area for all the landuse classes without losing out any information for future research needs.*

*We will try to explain the process of threshold selection more clearly in the revised manuscript.*

RC11: Page 7: "delineated into 53 subbasins, with the main outlet in Abeokuta". Can you please give some summary statistics about them: min max average area, elevation, etc. Daily precipitation 5 data (1984-2012) and minimum and maximum temperature data (1984-2012) at four weather stations (Fig. 1) were used as observed input data. Are you only using 4 stations for the whole basin (20292 km2)? Why not considering satellite products for a variable (precipitation), which sometimes could be even more important than etp? The authors should include this in the discussion. The missing values of daily precipitation and minimum and maximum temperatures, along with solar radiation, wind speed and relative humidity were simulated by the ArcSWAT CSFR_World weather generator: it is clear that the ArcSWAT CSFR_World is used for gap filling of precipitation and temperature. The authors should specify: 1) how did you use the dataset for solar radiation, wind speed and relative humidity? 2) At

which time resolutions are you specifying that input? 3) For which hydrological processes did you use these "simulated" forcing variables and how this affects your results?

*AC11: Many thanks for raising this comment for clarification.*

*In SWAT, the Ogun River Basin was delineated into 53 subbasin for this study. The summary statistics of the 53 delineated subbasins is as follows; the minimum and maximum elevation are 23 m and 624 m respectively, while the mean elevation is 289.1 m. The minimum and maximum subbasin area are 72.4 km$^2$ and 853.1 km$^2$ respectively, while the mean is 382.8 km$^2$. The minimum and maximum subbasin length are 72633 m and 269744 m while the mean length is 153528 m. We will add this information in the manuscript.*

*Four weather station data from the Nigerian meteorological agency were used as observed input data because that is the only available ground weather stations in the study area. Initially we considered using satellite precipitation products but comparing the observed with the satellite data downloaded yielded a huge bias. Thus we focused on the limited ground truth stations weather data. The weather stations are distributed evenly within the watershed, and one station is located outside the basin. All of the stations show high consistency. Looking at the elevation, no orographic effect needed to be considered for correcting the precipitation values. The reason for using only 4 weather stations in this study will be included in the revised manuscript.*

*SWAT requires daily values of solar radiation, relative humidity and wind speed in addition to the daily precipitation, minimum and maximum temperature as weather input in SWAT. One out of many options in SWAT to generate this input variables, is to use " WGEN_CFSR_World (ArcSWAT CSFR_World weather generator)" which is an MS Access file containing long-term monthly weather statistics covering the entire globe. In this study, the long-term monthly weather statistics developed using The National Centres for Environmental Prediction (NCEP) Climate Forecast System Reanalysis (CFSR) global dataset were used to simulate daily solar radiation, relative humidity and wind speed using the WGEN CFSR World. It is also used for filling gaps in measured climate data.*

*The simulated variables were used as input variables into Penman-Monteith and Priestly-Taylor equations for obtaining the different PET estimates from SWAT.*

*The simulated variables allow options for different evaporation estimates which actually affect the results of the model performance during the calibration and validation period as shown in the manuscript.*

RC12: Page7 line22-27: it sounds slightly repetitive: please consider to write the full sentence only of one model structure and to generalize for the other 2.

*AC12: We agreed to make the suggested changes in page 7 from line 16-27. We will rewrite the statement in the revised manuscript to read as follows:*

*The SWAT model was run three times whereby three different PET equations were tested in SWAT: the model setup (in term of all the single components were the same except the PET). The SWAT model in which three different AET estimates were obtained respectively with each PET equation (SWAT_HG, SWAT_P-M, SWAT_P-T) were used to evaluate the model performance by comparing the calibrations, validations, and the reference run (uncalibrated SWAT) with two global AET products (GLEAM_3.0a and MOD16), thus allowing for six calibrations of SWAT (GS1 through MS6) and six reference runs with SWAT default parameters (RGS1 through RMS6). Calibration GS1 through GS3 refers to SWAT simulated AET when SWAT_HG, SWAT_P-M, SWAT_P-T were selected respectively and were calibrated and validated with the AET from GLEAM_v3.0a. Calibration MS4 through MS6 refers to SWAT simulated*

*AET when SWAT_HG, SWAT_P-M, SWAT_P-T were selected respectively and were calibrated and validated with the AET from MOD16. Reference run (RGS1 through RGS3) refers to SWAT simulated AET obtained with SWAT default parameters when SWAT_HG, SWAT_P-M, SWAT_P-T were selected respectively and were compared with GLEAM_v3.0a. Reference run (RMS4 through RMS6) refers to SWAT simulated AET obtained with SWAT default parameters when SWAT_HG, SWAT_P-M, SWAT_P-T were selected respectively and were compared with MOD16 AET.*

*The changes will be made in the revised manuscript as requested. The figure 2 as shown in the manuscript will also be updated.*

RC13:  Page 8 line 10: please explain what are the main differences between the two dataset GLEAM_v3.0a and GLEAM_v3.0b and justify why you selected one of the two.

*AC13: Many thanks for raising this comment for more clarification.*

*The two datasets differ in their forcing variables and their temporal coverage (Martens et al., 2016).*

*GLEAM_v3.0a is a global dataset that is based on reanalysis net radiation and air temperature, satellite-based vegetation optical length, and a combination of gauge-based, reanalysis satellite-based precipitation. It is a dataset spanning the 35-year period 1980-2014.  For this study, we preferred and selected GLEAM_v3.0a dataset spanning 24-year period 1984-2012 because of its long-term availability that allows reasonably selection and splitting of calibration and validation periods that are not substantially different in climatic condition i.e., wet, moderate, and dry years occur in both periods and which covers our SWAT simulation output period (1989-2012).*

*GLEAM_v3.0b is a global dataset driven by satellite data only and spanning 13-year period 2003-2015. We considered this dataset for the verification of SWAT simulated AET because there are no ground truth AET data in the study area and also, because of its different forcing variable, which categories it as an independent dataset not considered in the calibration and validation period.*

*The added statement will be used to clarify further the differences between GLEAM_v3.0a and GLEAM_v3.0b in the revised manuscript.*

RC14:  Page 8 line 25: "was implemented in SWAT-CUP. SWAT-CUP (Abbaspour, 2015)" move the citation when you firstly introduce SWAT-CUP.

*AC14: Thank you. We will make the changes as suggested*

RC15:  Pages 8 line 28 to page 9 line 6: Please specify the parameter set that you started the sensitivity analysis with, at least the processes to which they are related. Moreover, specify the list of the parameters that resulted sensitive and how you define a parameter as "sensitive".

*AC15: We started the global sensitivity analysis for each of the six-calibration run with the same 50 parameters as shown in the table below. The table below only represent the result of the GS1 parameter sensitivity analysis but the same 50 parameters were used for GS2 through MS6 sensitivity analysis.*

*In this study as described in the manuscript, the parameter sensitivity is determined by numerous rounds of Latin Hypercube sampling and we defined a parameter as being sensitive when (considering the absolute values) large values of t-stat and smaller values of p-value were obtained, then, the more sensitive the parameter.*

*The processes to which the parameters are related hydrological are runoff, evaporation, interception, transpiration. In short, each of the hydrological parameters are represented in the 50 parameters we started the initial global sensitivity with, as shown in table 1.*

*Table 1: The 50 parameters consider in the initial global sensitivity analysis and their relative significance. The table below only show the global sensitivity analysis result of the GS1*

| Parameter Name | t-Stat | P-Value |
|---|---|---|
| 45:V__BMX_TREES{..}.plant.dat | 0.68 | 0.62 |
| 23:R__SOL_CBN(..).sol | -0.99 | 0.50 |
| 47:V__TMPMX(..).wgn | 1.84 | 0.32 |
| 48:V__PCPMM(..).wgn | -2.27 | 0.26 |
| 35:V__LAI_INIT.mgt | 2.28 | 0.26 |
| 31:V__SURLAG.bsn | -3.03 | 0.20 |
| 28:R__ALPHA_BF_D.gw | 3.21 | 0.19 |
| 19:A__GWQMN.gw | -3.32 | 0.19 |
| 24:R__SOL_ALB(..).sol | -3.82 | 0.16 |
| 37:V__FLOWFR.mgt | -3.86 | 0.16 |
| 40:R__SOL_ZMX.sol | -4.06 | 0.15 |
| 50:R__SOL_Z(..).sol | 4.08 | 0.15 |
| 39:V__TLAPS.sub | 4.27 | 0.15 |
| 15:V__ESCO.hru | -4.35 | 0.14 |
| 27:V__RCHRG_DP.gw | 4.55 | 0.14 |
| 42:V__ALPHA_BF_D.gw | -4.70 | 0.13 |
| 25:R__USLE_K(..).sol | -5.73 | 0.11 |
| 26:V__SOLARAV(..).wgn | -5.96 | 0.11 |
| 38:V__TDRAIN.mgt | 6.02 | 0.10 |
| 34:V__BIO_MIN.mgt | 6.03 | 0.10 |
| 46:V__TMPMN(..).wgn | -6.18 | 0.10 |
| 3:V__REVAPMN.gw | -6.19 | 0.10 |
| 43:V__RADINC(..).sub | -6.20 | 0.10 |
| 36:V__BIO_INIT.mgt | 6.25 | 0.10 |
| 13:V__EVRCH.bsn | -6.91 | 0.09 |
| 20:R__HRU_SLP.hru | 6.94 | 0.09 |
| 21:V__GW_DELAY.gw | 6.96 | 0.09 |
| 14:V__GW_REVAP.gw | -7.01 | 0.09 |
| 44:V__HUMINC(..).sub | -7.22 | 0.09 |
| 29:V__SHALLST.gw | -7.34 | 0.09 |
| 16:V__CH_N2.rte | 7.59 | 0.08 |
| 22:V__ALPHA_BNK.rte | 7.66 | 0.08 |
| 17:V__CH_K2.rte | -7.66 | 0.08 |
| 41:V__CH_N1.sub | 7.70 | 0.08 |
| 49:V__DEEPST.gw | 7.98 | 0.08 |
| 12:V__EVLAI.bsn | 8.03 | 0.08 |
| 11:V__OV_N.hru | 8.12 | 0.08 |
| 18:V__SFTMP.bsn | 8.49 | 0.07 |

| | | |
|---|---|---|
| 33:V__GWHT.gw | 8.74 | 0.07 |
| 10:V__FFCB.bsn | -8.78 | 0.07 |
| 7:R__SOL_BD(..).sol | -8.91 | 0.07 |
| 6:V__EVRSV.res | -8.95 | 0.07 |
| 32:V__GSI{..}.plant.dat | -9.78 | 0.06 |
| 2:V__EPCO.hru | 10.14 | 0.06 |
| 4:R__SOL_K(..).sol | 10.51 | 0.06 |
| 9:V__ALPHA_BF.gw | -10.52 | 0.06 |
| 1:V__ESCO.hru | -10.71 | 0.06 |
| 30:V__CANMX.hru | 11.52 | 0.06 |
| 5:R__SOL_AWC(..).sol | 12.37 | 0.05 |
| 8:R__CN2.mgt | 15.79 | 0.04 |

*After choosing the 11 most sensitive parameters based on the t-stat and p-values. We tried many combinations of the most sensitive analysis e.g. we started with P < 0.09 (37 parameters) the result was not as good at that of the 11 parameters combination. So we decided to take the 11 most sensitive parameters and we run another global sensitivity analysis to further identify the relative significance of each parameter before calibration. Using 11 parameters gave the most reasonable results. This methodology was extended to all the remaining calibration runs.*

*We will further clarify the number of parameters we started with, the process to which they are related and how we define the parameter sensitive in the revised manuscript.*

*Table 2: The 11 most sensitive parameters consider in the final calibration and validation of all the six calibration runs. The table below only show the sensitivity analysis of the GS1 calibration results.*

| Parameter Name | t-Stat | P-Value |
|---|---|---|
| 9:R__SOL_AWC(..).sol | 0.01 | 0.99 |
| 2:V__EPCO.hru | -0.21 | 0.83 |
| 7:V__FFCB.bsn | -0.27 | 0.79 |
| 4:V__GSI{..}.plant.dat | 0.40 | 0.69 |
| 6:V__EVRSV.res | -0.54 | 0.59 |
| 10:R__SOL_K(..).sol | -0.89 | 0.37 |
| 5:V__ALPHA_BF.gw | 1.43 | 0.15 |
| 11:R__SOL_BD(..).sol | -2.01 | 0.04 |
| 3:V__CANMX.hru | 2.19 | 0.03 |
| 1:V__ESCO.hru | -2.86 | 0.00 |
| 8:R__CN2.mgt | -23.93 | 0.00 |

*RC16:* Page 9 line 16: A metric among the six can be considered an objective function if it is optimized in the calibration procedure; it can be considered as goodness of fit metric if it is used to quantify how well or bad the model reproduces the measured data. Are those goodness of fit metrics? Which one of these six metrics has been optimized in the calibration procedure? Have you used all of them also as objective function? This is not fully clear.

*AC16: Many thanks for raising this comment for more clarification.*

*The Nash-Sutcliffe is the selected objective function that was optimized during the calibration procedure .This statement will be added for clarity in the revised manuscript.*

*RC17* Page 9 line 20 –Page 10 line 20: Consider to: i) just spell in the text the statistics used, their ranges and their optimal values and ii) move in appendix the explanation of each statistics because they are well known.

*AC17: Thank you. We will make the changes as suggested.*

*RC18:* Page 10 line 23-26: please specify how the uncertainty is quantified: what are the parameters that are changed/sampled the LHS, what are their ranges?

*AC18: Thank you for highlighting this point for further clarification*

*As described in the manuscript the uncertainty was quantified by: 1.) assessing the percentage of GLEAM_3.0a AET data bracketed by the model output 95% predictive uncertainty band, the index used for the quantification is P-factor, and 2.) assessing the ratio of the average width of the 95ppu and the standard deviation of the GLEAM_3.0a, the index used for the quantification is R-factor.*

*The parameters that are changed are listed in Table 2 above by implementing numerous rounds of Latin Hypercube sampling. The optimum values are presented in the manuscript but their ranges are presented in Table 3a-b below:*

*Table 3a: Eleven parameter set and their ranges used for the first 1000 simulations (1st iteration during calibration) in this study for all the six calibration runs.*

| Parameter name | Minimum range | Maximum range |
|---|---|---|
| v__ESCO.hru | 0.00 | 1.00 |
| v__EPCO.hru | 0.00 | 1.00 |
| v__CANMX.hru | 0.00 | 100.00 |
| v__GSI{2,4,5}.plant.dat | 0.00 | 5.00 |
| v__ALPHA_BF.gw | 0.00 | 1.00 |
| v__EVRSV.res_________17,50 | 0.00 | 1.00 |
| v__FFCB.bsn | 0.00 | 1.00 |
| r__CN2.mgt | -0.25 | 0.85 |
| r__SOL_AWC().sol | 0.23 | 0.95 |
| r__SOL_K().sol | -0.06 | 0.95 |
| r__SOL_BD().sol | -0.41 | 0.95 |

*Table 3b: Eleven parameters set and their ranges used for the second 500 simulations of the GS1 calibration run (2nd iteration during calibration) showing how the parameter range changes from the 1st iteration of GS1 calibration run.*

| Parameter name | Minimum range | Maximum range |
|---|---|---|
| v__ESCO.hru | 0.00 | 0.59 |
| v__EPCO.hru | 0.31 | 0.95 |
| v__CANMX.hru | 0.00 | 59.46 |
| v__GSI{2,4,5}.plant.dat | 2.35 | 5.00 |
| v__ALPHA_BF.gw | 0.38 | 1.00 |
| v__EVRSV.res_________17,50 | 0.43 | 0.60 |

| | | |
|---|---|---|
| v__FFCB.bsn | 0.36 | 1.00 |
| r__CN2.mgt | -0.47 | 0.40 |
| r__SOL_AWC().sol | 0.52 | 0.90 |
| r__SOL_K().sol | -0.12 | 0.59 |
| r__SOL_BD().sol | -0.39 | 0.50 |

*All the necessary additional information will be added in the revised manuscript for further clarification.*

*RC19:* Page 11: please show a figure of the river basin with the subbasin polygons and the pixel of MODIS and GLEAM. This will help the reader to understand how many pixels of MODIS and GLEAM cover your basins. Each sub-basins has its own model AET. How did you choose the MODIS or GLEAM pixel to compare with and compute the NSE, $R^2$, etc.

*AC19: Thanks for the suggestion, we prepared a figure showing the river basin with the subbasin polygon and the pixel of MOD16 and GLEAM AET. We will insert the figures as suggested in an Appendix.*

*To compare MODIS pixel value to SWAT simulated AET values from each subbasin for computing the NSE, $R^2$, PBIAS, KGE, an area-weighted averaging scheme was performed in ArcGIS to create aggregated monthly time-series of AET data of each subbasin.*

[Figure]

**MOD16 AET (1km)**

MOD16AET_intersect_ogun53_subbasin

● Polygon_to_Points_2000-20012

*Figure 1: Ogun River Basin with its 53-subbasin polygons intersected by the pixel of MOD16 AET.*

*The GLEAM_3.0a and GLEAM_3.0b AET is provided in netcdf format, with one file per year and variable. The datasets are available on a 0.25⁰ latitude- longitude regular grid and at daily temporal resolution.*

[Figure]

*Figure 2: Ogun River Basin with its 53-subbasin polygons intersected by the pixel of GLEAM AET*

*We used "make NetCDF raster layer" tool in ArcGIS to convert the NetCDF file into a raster layer to view how many pixels of GLEAM cover our subbasins (Figure 1). We realized some points from which the data will be extracted from the pixel are not located in some subbasins (Fig.3). Therefore, we decided to create a point at the center of each subbasin in ArcGIS (Fig. 4). The coordinates of each point at the center of the subbasins were obtained.*

[Figure]

*Figure 3: Ogun River Basin with its 53-subbasin polygons intersected by the pixel of GLEAM AET with a point at the center of each subbasin.*

*To enable us to compare GLEAM pixel value to SWAT simulated AET values for each subbasin for computing the NSE, R2, PBIAS, KGE, we went back to the NetCDF file of the daily time series (1989-2012) and extracted the GLEAM AET value for each subbasin using "Make NetCDF table view. The values extracted were aggregated to monthly values. The same procedure was carried out for GLEAM_3.0b version.*

*RC20:* Page 11: figure 3, 4, 5, and 6 please consider to add a 4th class for the range of KGE and NSE (<0). This indicates where just the mean of the observed data will be more performing than the model itself

*AC20: Many thanks for the suggestion that needs further clarification.*

*We divided the results into 5 classes. Wish we think the performance rating classes represented with figure 3,4,5,6 for each subbasins are well represented and sufficient*

*-6.0 – 0 = red*

*0.10- - 0.50 = purple*

*0.51 – 0.60 = yellow*

*0.61-0.70 = orange*

*0.71 -1.0 = green*

*The KGE and NSE figure that do not contain a class -6.0 – 0 (<0) do not contain values <0 (fig.3, 4 in the manuscript) and figures that contains -6.0 – 0 (<0) has a model performance that is less than 0 (fig. 5 and fig.6 in the manuscript)*

*We do not think is necessary to include a class in the legend without having its value in the subbasin polygon.*

*RC21:* Page 10 line 15: equation is not correct please revise it.

*AC21: Many thanks for the point. We will correct and insert it in the revised manuscript.*

*RC22:* Page 11: figure 3, 4, 5, and 6 please add the North symbol and the scale bar in the maps.

*AC22: Many thanks for the suggestion. We will include North symbol and the scale bar in the revised manuscript.*

*RC23:* Page 11 line 23: "The results of global sensitivity analysis revealed that the SCS runoff": please specify from where the reader can see this.

AC23: Thank you. We will include Table 2 at the end of the statement to read as follow:

"The results of global sensitivity analysis revealed that the SCS runoff curve number (CN2.mgt) is the most sensitive parameter to SWAT simulations of AET for all the six calibrations (Table 2)"

*RC24:* Page 12: all the page can be summarized by just one figure reporting on the x-axis the model configurations and on the y axis the percentage of sub-basin for a given class of the goodness of fit index (NSE, R2, etc).

AC24: Many thanks for the suggestion. The results presented in page 12 are already summarized in figure 3, 4,5,6,7, and 8. We believe this figures depicts and summarized well the SWAT model performance for each subbasin and the percentage covered (using pie chart). We believe having another figure to summarize page 12 is not needed.

We agree to reduce the presentation of results described on page 12 because the figures already shown them.

RC25: Page 13 the paper goes from section 3 to subsection 3.3: 3.1 and 3.2 are missing.

AC25: Thank you for highlighting this point. We will include subsection 3.1 and 3.2 in the revised manuscript

RC26: Page 14: increases in February from 55mm to 76mm as the space between 55 and mm

AC26: Thank you for highlighting this point. We will make the corrections in the revised manuscript

RC27: Page 15 line 5: "Using the guidelines in Moriasi et al. (2007, 2015) and Kouchi et al. (2017) for" probably these guidelines were drawn for runoff? Is it correct to use it for others hydrological processes? Is this been done in the past? If yes, please add a citation otherwise just clarify this aspect.

AC27: *Many thanks for raising this point for further clarification.*

*the general hydrologic model performance ratings for recommended statistics (NSE, PBIAS, $R^2$) performed at a monthly time and recommended by Moriasi et al. (2007, 2015) and Gupta et al are mostly drawn for runoff, sediment and nutrients because when these articles were published, the use of remotely sensed evapotranspiration datasets for hydrologic model calibration/validation did not gain much ground.*

*In this paper, we also conducted reviewed literatures on model evaluation methods and ratings for model calibration using satellite or non-satellite derived evapotranspiration. Ha et al. (2018) presented a study of calibration of spatially distributed hydrological processes and model parameters in SWAT using remote sensing data and an autocalibration procedure. NSE, $R^2$, and KGE criteria were used to assess the model performances.*

*Djman (2016) in their study of evaluation, calibration and validation of six reference $ET_0$ equation for Senegal River Delta using Penman-Monteith derived $ET_0$ obtained at saint louis station (1960-2012), uses $R^2$ and other statistical measures to perform the evaluation. Lopez et al. (2017), calibrated a large-scale hydrological model using satellite-based soil moisture and evapotranspiration products. They evaluated the model performance using NSE, PBIAS, KGE and R. Samadi et al. (2016) presented a study on assessing the sensitivity of SWAT physical parameters to potential evapotranspiration estimation methods over a coastal plain watershed in the southeastern United States, NSE, and KGE statistical measure were used for the model performance assessment.*

*All the reviewed literatures set their performance ratings for recommended statistics (NSE, PBIAS, R, KGE, $R^2$) based on Moriasi et al. (2007, 2015) guidelines.*

*In this study, we follow Lopez et al. (2017) and others as reviewed above to base our performance rating criteria for judging the model performance by using NSE, $R^2$, PBIAS and KGE.*

*RC28:* Page 15 line 20: "From our results, we agree that the AET from MOD16 tends to overestimate AET". Overestimate against what? This is a strong statement mainly because there is not direct comparison against measured AET data

*AC28: Thank you for highlighting this point. We will make the corrections in the revised manuscript*

*We agree this is a strong statement that needs to be revised. Actually, we meant that AET from MOD 16 values are higher than that of GLEAM AET and SWAT simulated AET and that this finding agrees with other studies carried out in tropical regions. Since we are using the satellite based MOD 16 to calibrate the SWAT AET simulation, the statement needs correction and we agree to write it to read as follows:*

*"From our results, we agree that AET from SWAT tends to underestimate AET, when calibrated the model with MOD16 AET and this finding agrees with other studies carried out in the tropical regions (Ruhoff et al., 2013)".*

Reference

Arnold, J. G., Srinivasan, R., Muttiah, R. S. and Williams, J. R.: Large area hydrologic modeling and assesment Part I: Model development, JAWRA J. Am. Water Resour. Assoc., 34(1), 73–89, doi:10.1111/j.1752-1688.1998.tb05961.x, 1998.

Bicknell, B. R., Imhoff, J. C., Kittle, J. L. J., Donigian, A. S. J. and Johanson, R. C.: Hydrological Simulation Program--Fortran:  User's manual for version 11, , 755, doi:EPA/68/C-01/037, 1997.

Djaman,K., Tabari, H., Baide,A.B., Diop,L., Futakuchi,K., and Irmak,S.: Analysis, calibration, and validation of evapotranspiration model to predict grass reference evapotranspiration in Senegal River Delta. Journal of Hydrology: Regional Studies 8, 82-94,2016.

Ewen, J., Parkin, G. and O'Conell, P. E.: SHETRAN : Distributed River Basin Flow Modeling System, J. Hydrol. Eng., 5(JULY), 250–258, doi:doi:10.1061/(ASCE)1084-0699(2000)5:3(250), 2000.

Franco, A. L. and Bonumá, N.B.: Multi-variable SWAT model calibration with remotely sensed evapotranspiration and observed flow. *RBRH*, *22*, e35, 2017.

Ha,L.T, Bastiaanssen, W.G.M, van Griensven, A, van Dijk, A.I.J.M, Senay, G.B.: Calibration of Spatially Distributed Hydrological Processes and Model Parameters in SWAT Using Remote Sensing Data and an Auto-Calibration Procedure: A Case Study in a Vietnamese River Basin. *Water*. 10(2):212, doi: 10.3390/w10020212, 2018.

Hargreaves, G. H. and Samani, Z. A.: Reference Crop Evapotranspiration from Temperature, Appl. Eng. Agric., 1(2), 96–99, doi:10.13031/2013.26773, 1985.

Kouchi, D. H., Esmaili, K., Faridhosseini, A., Sanaeinejad, S. H., Khalili, D. and Abbaspour, K. C.: Sensitivity of calibrated parameters and water resource estimates on different objective functions and optimization algorithms, Water (Switzerland), 9(6), 1–16, doi:10.3390/w9060384, 2017.

Lopez Lopez, P., Sutanudjaja, E., Schellekens, J., Sterk, G. and Bierkens, M.: Calibration of a large-scale hydrological model using satellite-based soil moisture and evapotranspiration products, Hydrol. Earth Syst. Sci. Discuss., (January), 1–39, doi:10.5194/hess-2017-16, 2017.

Martens, B., Miralles, D., Lievens, H., Fernández-Prieto, D., and Verhoest, N.: Improving terrestrial evaporation estimates over continental Australia through assimilation of SMOS soil moisture, Int. J. Appl. Earth Obs., 48, 146–162, 2016.

Moriasi, D. N., Arnold, J. G., Van Liew, M. W., Binger, R. L., Harmel, R. D. and Veith, T. L.: Model evaluation guidelines for systematic quantification of accuracy in watershed simulations, Trans. ASABE, 50(3), 885–900, doi:10.13031/2013.23153, 2007.

Moriasi, D. N., Gitau, M. W., Pai, N. and Daggupati, P.: Hydrologic and Water Quality Models: Performance Measures and Evaluation Criteria, Trans. ASABE, 58(6), 1763–1785, doi:10.13031/trans.58.10715, 2015.

Ruhoff, A. L., Paz, A. R., Aragao, L. E. O. C., Mu, Q., Malhi, Y., Collischonn, W., Rocha, H. R. and Running, S. W.: Assessment of the MODIS global evapotranspiration algorithm using eddy covariance measurements and hydrological modelling in the Rio Grande basin, Hydrol. Sci. J., 58(8), 1658–1676, doi:10.1080/02626667.2013.837578, 2013

Samadi,S.Z.: Assessing the sensitivity of SWAT physical parameters to potential evapotranspiration estimation methods over a coastal plain watershed in the southeastern United States. IWA, 48 (2) 395-415, doi: 10.2166/nh.2016.034, 2017.

Wang, X., Melesse, A. M. and Yang, W.: Influences of potential evapotranspiration estimation methods on SWAT's hydrologic simulation in a northwestern Minnesota watershed, Trans. ASABE, 49(6), 1755–1771, doi:10.13031/2013.22297, 2006.

---

## Author Response (AR1)

**Response to the Editor for hess-2018-170**

Dear Editor, thank you very much for the opportunity to resubmit a revised copy of our paper on 'Multi-site calibration and validation of SWAT with satellite-based evapotranspiration in a data sparse catchment in southwestern Nigeria'. The constructive criticism, comments and suggestions offered by the reviewers have been immensely helpful. We greatly appreciate their insightful comments on revising the paper. The manuscript has been revised to address the reviewer's concerns and a point-by-point reply to the reviewer's comments has been made. The changes arising from the comments have clearly improved our manuscript, which you find uploaded alongside this document. All the modifications are highlighted in yellow in the paper as you requested. We look forward to hearing from you in due time regarding our submission and to respond to any further questions and comments you may have.

**Response to the Reviewers Comments for hess-2018-170**

We thank the anonymous referees #1 and #2 for reviewing our manuscript. We are especially grateful for the many insightful and constructive comments and their valuable suggestions; these changes have clearly improved the quality of the manuscript. We have to the best of our abilities responded to them and address the referees' comments in the following point by point response. Note the following conventions: RC = referee comments, AC = authors comments (replies) printed in italic. All the modifications are highlighted in yellow in the revised manuscript.

**Reply to comments of Anonymous Referee #1**

**Major comments**

RC 1: This paper calibrates the SWAT model using 2 available ET global products, a simple remote sensing ET equation (MOD16) and a more complex water balance model forced by remote sensing data (GLEAM). MOD16 does not explicitly account for transient water stress (as, say, derived from TIR data); how does this impact the results?

*AC1: We agree with the referee that MOD16 AET does not explicitly account for transient water stress because it is not directly derived from Thermal Infrared Remote sensing (TIR) data. Some of the reasons for not fully using TIR data at the global scale for the MOD16 product are: a) a changing relationship between TIR based land surface temperature (LST) and NDVI when moving from mid to high latitudes; b) LST as derived from TIR is not equal to the aerodynamic surface temperature (which is driving the sensible heat flux), potentially leading to non-accurate AET estimations under various conditions (Mu et al., 2007, 2013). MOD16 applies the Penman-Monteith (PM) equation to calculate AET on a global scale by using variables and parameters needed from VIS/NIR remote sensing (land cover, LAI, albedo, FPAR) and from daily meteorological reanalysis data (radiation, $T_{air}$, pressure, rel. humidity; NASA's global modeling and assimilation office, GMAO). In principle, the surface resistance ($r_s$) parameter in the PM equation accounts for any direct effect on AET due to limitations in available water. The MOD16 AET scheme however, does not include any soil water content data directly. The way $r_s$ is derived in the MOD16 AET scheme only considers an indirect effect via a non-linear dependency of $r_s$ with the water vapor pressure deficit (VPD) in the atmosphere. VPD under daytime conditions often represents a proxy for soil moisture conditions and therefore $r_s$. The impact of the not-explicit consideration of transient water stress in the MOD16 AET product on our SWAT-model calibration is difficult estimate.*

*Transient water stress is not a main challenge in the present study area, which is located in the humid region of south western Nigeria with a mean Aridity Index of 0.75 from the period 1989 to 2012 (A.I. > 0.65 value, which is considered to be a humid region; UNEP, 1997). We have added this paragraph to indicate minimal impact of non-explicit consideration of transient water stress of MOD16 AET on our results in the revised manuscript (Page 18, line 24 – 30))*

*We also reviewed the literature of the MOD16 AET with measured (EC) flux data at sites climatically similar to our catchment and found an agreement between our catchment and the result obtained between MOD16 AET and the measurements for a study area in the tropical region, covered by natural savannah vegetation site in the Rio Grande Basin, Brazil conducted by Ruhoff et al. (2013) also located in a tropical region. While not being comprehensive, the comparisons were an indication that MOD16 AET behave similarly having a positive PBIAS (MOD 16 overestimating AET in both sites). Also, Trambauer et al. (2014) compared different evaporation products for Africa, in their paper they stated that MOD16 evaporation do not show a good agreement with other products in most part of Africa,*

*while the rest (GLEAM, ECMWF reanalysis ERAL-Land and PCR-GLOBWB hydrological model simulated AET) are more consistence.*

*We have fully indicated this comparison and have discussed this point in more detail in the revised version of the manuscript (page 18, line 31 – 34 and page 19, line 1- 11) and have also provided the same level of detail for the description of GLEAM AET product (page 5, line 1-7 and page 19, line 22 - 24*

RC2: It is unclear to me whether the SWAT model used here uses the plant growth model.

*AC2: The Soil and Water Assessment Tool (SWAT) is an eco-hydrological model that uses at its core the plant growth model EPIC (Williams et al., 1989) that is able to simulate the growth (including nutrient and water uptake) of many types of crops and trees as land cover. The plant growth component of SWAT is a simplified version of the EPIC plant growth model. We have included this sentence in the revised manuscript (page 7, line 6 - 13).*

RC2b: How is the vegetation taken into account?

*AC2b: SWAT is a physically based model that requires a land use map as one input data source that represents the spatial distribution of vegetation in the watershed. SWAT categorizes plants into seven different types: warm season annual legume, cold season annual legume, perennial legume, warm season annual, cold season annual, perennial and trees.*

*Plant growth is modeled by simulating leaf area development, light interception and conversion of intercepted light into biomass assuming a plant species-specific radiation-use efficiency. Hence, In SWAT, phenological plant development is based on daily accumulated heat units.*

*The plant growth model is used to assess removal of water and nutrients from the root zone, transpiration, and biomass/yield production. In SWAT the plant growth can be inhibited by a minimum or maximum temperature, available water, nitrogen and phosphorus stress. The potential biomass is based on a method developed by Monteith in 1977 (a radiation model, that uses solar radiation as its decisive factor of crop production, while temperature and water are two other important factors); and a harvest index is used to calculate the final yield. We have included this sentence in the revised manuscript for more clarity (page 7, line 6- 13).*

RC3: Two additional important performance metrics are needed as a reference for the six calibrations:

1. A reference run with default (uncalibrated) parameters – this is needed absolutely!
2. A focus on stressed /unstressed periods as defined by GLEAM ET product, with metric specific for each periods; this would allowed help analyze whether model improvement comes from better ETP formulation or a better simulation of stress.

*AC3: 1. We agree with the referee and we have added the reference scenario results in the revised manuscript (page 15, line 8 – 12 and page 33 (Figure 3))*

*AC3: 2. it is true that stressed /unstressed periods should be considered. We included both dry and the wet periods of growth in our calibration and validation. In this paper, we followed the split-sample test as presented by Klemes (1986), which is a model calibration and validation approach that consists of equally splitting the available data, when the record is sufficiently long to represent different climatic*

*conditions. Also, Gan et al. (1997) stated that data are most frequently split by time periods, carefully ensuring that the climate data used for both calibration and validation are not substantially different, i.e., wet, moderate, and dry years occur in both periods. In this study, this we achieved by comparing the absolute AET values of the calibration and validation years, with the aim of having a minimal difference by including wet, moderate, and dry years in both periods. We further checked the correlation of the calibration and the validation years which shows high value. With these two indices, we successfully included wet, moderate and dry years in each calibration and the validation periods. Since the now newly added reference run has included both dry and wet periods, we believe having another separate performance metric accounting for stressed/unstressed periods will not be necessary.*

RC4: The description of the calibrated parameters (which, I assume, follow the SWAT terminology) is lacking: there is only a Table; equations showing where those parameters appear should be provided in, say, an annex, to improve the paper standalone readability.

*AC4: We believe that the list of the calibrated parameters and where they can be found, as well as the description of the selected parameters and their calibrated optimal values, are important and these are provided in Table 2. Including detailed equations showing where these parameters appear will be too ambitious, because different equations are formulated for different hydrological conditions for most of the parameters, and this information will be difficult to present in a tabular form. To this effect, we have added in the revised manuscript some equations (equations taken from SWAT theoretical documentation) showing where the 11 most sensitive parameters used in the calibration appear in SWAT (Appendix C in page 45 - 47)*

**Minor comments:**

RC5: Figure 2: why use a half-half split sample for MOD16 but only a 1/11 split sample for GLEAM?

*AC5: The referee is right to point out this issue for clarification. MOD16 is a global dataset spanning the 13-year period 2000-2012 and the splitting of calibration period (2000-2006) and validation period (2007-2012) followed the split-sample test as presented by Klemes (1986) and Gan et al. (1997). While the GLEAM_v3.0a is a global dataset spanning the 35-year period 1980-2014. For this study, we used GLEAM_v3.0a dataset spanning 24-year period 1984-2012 because the SWAT simulation output was from 1989-2012. The splitting of calibration period (1989-2000) and validation period (2001-2012) for GLEAM_v3.0a also followed the split-sample test as presented by Klemes (1986) and Gan et al. (1997). The splitting by time periods was carefully done by ensuring that both MOD16 and GLEAM AET available years dataset used for calibration and validation are not substantially different, i.e., wet, moderate, and dry years occur in both periods. we have rightly clarified this point in the revised manuscript (page 12, line 6 -9, page 12, line 28-33 and page 14, line 2-3)*

RC6: Equation 5: the square root should extend to the third quadratic term.

*AC6: We agreed with the referee and we have made the suggested changes in page 43: KGE*

RC7: Page 10 line 18: use the term "ratio"

*AC7: Thank you! We have made the changes in page 43, line 8-9.*

RC8: Page 13 line 22: predicted>predict

*AC8: Thank you! We have made the changes in **page 16, line 11***

RC9: Page 15 line 11: Runoff > Ruhoff?

*AC9: Ruhoff et al. (2013) cited in the manuscript is correct. They author a Journal we cited in our manuscript.*

RC10: Page 15 line 33: Therefore, the Hargreaves……periods": I don't understand this sentence

*AC10: We agree with the referee that the sentence needs clarification. We have made the clarification and the changes **in page 19, line 16-21** to read in the following way:*

*"The better SWAT model performance in GS1 is attributed to the selection of the Hargreaves equation, which is based on available observed precipitation and maximum and minimum temperature to obtained AET, while the Penman-Monteith and the Priestly-Taylor equations are driven by simulated variables (wind speed, relative humidity and solar radiation) in this study. Also the complex water balance model algorithm of GLEAM takes into account soil-water balance, bare-soil evaporation and open water evaporation, evaporative stress factor and rainfall interception, all of which assist in simulating the dynamic hydrological components, especially the AET"*

**Reply to comments of Anonymous Referee #2**

**Major comments**

RC (a): Literature Review: it lacks significant contributions in the context of large-scale hydrological model simulation in data scarce area and it mainly focused on previous studies based on SWAT. It could be good to mention and discuss other approaches even if performed in different study areas but with the same problems (data scarce areas) (Kim et al., 2008; Kim and Kaluarachchi, 2009; Gebremicael et al., 2013; Tekleab et al., 2011; Abera et al., 2016 which applied a different hydrological modeling approach (Formetta et al., 2014))

*AC (a): Agreed. We have added other related references and discussed other approaches in the context of large-scale hydrological model simulation in data scarce area as suggested **in page 3, line 31 – 34 and page 4, line 1 - 13.***

RC (b): I feel that the authors should acknowledge explicitly that the analysis presented needs to be tested against observed data and that the satellite data are them self-based on modelling assumptions, which may or may not be plausible in some areas. Of course, they provide a huge help and the way in which they are used in the paper nicely show it, but probably assuming them as "measured" can be misleading. At least can be specified once in the text that "measured AET" doesn't mean proper eddy-covariance data

*AC (b): We thank the reviewer for this suggestion. In the revised manuscript, we have strengthened the fact that we only make use of two satellite derived AET products. We have also emphasized that we do not have any e.g. EC-based local measurements within our catchment. However, the satellite products have been tested elsewhere and we have briefly summarised study results that are relevant to our study (similar climate conditions) in a revised version of the manuscript **in page 5, line 18 – 20 and page 13, line 20 – 22.***

RC (c): In the paper is claimed the importance of the Curve Number parameter but nothing is said about soil moisture evolution and runoff. I wonder why the authors do not use runoff-measured data as independent validation. This will show the effects of the different ET calibration on the runoff dynamic. The two processes are strongly related, and the sensitivity of the CN parameter confirms

this. This will be an important added value to the paper. Again, the authors claim: "The average long-term annual of the water balance at the outlet of the study area shows a satisfactory percentage error of closure". Is this referred to modelled data or modelled and measured? The use of measured streamflow data would help to better understand this part as well.

*RC (c): It is true that we emphases importance on the curve number parameter because it is found to be sensitive for all the six calibrations. As suggested by the referee, we have mentioned and discussed other parameters relating to soil moisture and runoff considered during the calibration **in page 17, line 20-32 and in page 18, line 1-16.***

*We agree that AET calibration and runoff dynamics are strongly related. In the manuscript, we do not consider runoff-measured data as independent validation because it is not available for the study area and that is main reason we considered AET derived from satellite products as an alternative option for SWAT hydrologic model calibration. We believe using a freely available AET products (GLEAM & MOD16), that have been heavily tested in the past in calibration and validation studies undertaken by a number of scientists is one solution in setting up a hydrological model that will be used as a decision support tool in such a data scarce region. These points have been added and emphased in the revised manuscript **in page 13, line 13 to 18**. The average long-term annual water balance at the outlet of the study area is referred to the SWAT modelled data with only precipitation and temperature. as measured input data. We assessed the water balance component of the model inorder to ascertain, examine and verify that, SWAT numerical technique and computer code truly represents the conceptual model and that there are no inherent numerical problems with obtaining a solution. We have clarified this in the manuscript that the long-term annual water balance is referred to the SWAT modelled data. The added sentence can be found **in page 14, line 22 to 24 and page 30**.*

RC (d): Because one of the main points in discussion/conclusion is the fact that: "Hargreaves equation had a superior model performance of the Penman Monteith and the Priestly-Taylor" the authors should add their equations in the text. This would help to visualize the variables in input for each method, the variables that have been chosen for calibration and the variables that have been excluded.

*AC (d): We have included the Hargreaves, Penman-Monteith and Priestley-Taylor PET equations in the paper for visualizing of each variables input into each **method in page 8, Eq.2, Eq.3 and Eq.4***

**Specific comments:**

RC1: Page 1 line 20: remove space in the number: River Basin (20 292 km2)

*AC1: We have left this unchanged. By checking the HESS manuscript preparation guidelines for authors, it is mentioned in the last sentence (h.) under heading "physical dimensions and units", that:*

*Numerals should also be typeset using upright fonts. The symbol for the decimal marker is the dot. To facilitate reading, numbers may be divided in groups of three using a thin space (e.g. 12 345.6), starting with the ten-thousand digit. Neither dots nor commas are permitted as group separators.*

RC2: Page 1 line 21: "The novelty of the study is the use of freely available satellite derived AET data for calibration/validation of each of the SWAT delineated subbasins, thereby obtaining a better performing model at the local scale as well as at the whole watershed level": sounds like this is the

first time the gleam dataset have been used to validate/calibrate swat, which is a strong sentence. May be in the study area?

*AC2: We agree with the referee that it is a strong sentence and we acknowledge that the novelty of the study needs further clarification. To this effect we have changed the statement in the revised manuscript **in page 2, line 2- 4** to read as follows:*

*"The novelty of the study is the use of these freely available satellite derived AET datasets to calibrate and validate three different SWAT simulated AET for each of the delineated subbasins, to improve the hydrological model performance at both the local and watershed scales for a data-scarce catchment."*

*We have also added the paragraph below in the reversed manuscript **in page 5, line 26 to 30** because we believe this is a new contribution both to research community and in the study area:*

*"Although the three different PET equations and the corresponding AET simulations from SWAT have been tested for their performance before (Wang et al. (2006); Franco and Bonumá (2017); Samadi (2017); Ha et al. (2018)), the study of calibrating each of the three SWAT simualated AET variables with two different available remotely sensed derived AET products for each delineated subbasin within SWAT to determine the highest performing model for a particular region has not been undertaken"*

RC3:  Page 1 line 24: "Three different structures of the SWAT model were used in which each model structure was a set-up of SWAT with a different potential evapotranspiration (PET) equation": I would say that three different PET equations are tested: the model setup (in term of all the single components is the same except the pet).

*AC3: We agree with the referee. As suggested, we have modified the sentence in **page 1, line 20-23, page 9, line 29 - 30** and **page 17, line 17** and throughout the manuscript.*

RC4:  Page 2: mechanistic, what the authors mean? Please explain.

*AC4: We agree with the referee that the terminology needs further clarification in the manuscript. we have simplified the terminology in **page 2, line 22 - 23** to read as follows:*

*"Numerous physically based distributed (PBD), continuous models that aim to describe which driving processes are present in a system and are able to make detailed predictions in both time and space"*

RC5:  Page 3 line 25: results showed a good Nash-Sutcliffe efficiency (NSE) and Coefficient of determination (R2) value for mothly average: quantify what good means for the authors and the values obtained.

*AC5: We agree with the referee that the statement needs further quantification and clarification. To this effect we have changed the statement in the revised manuscript **in page 3, line 25 - 28**, to read as follows:*

*"The model results showed a high Nash-Sutcliffe efficiency (NSE) of 0.72 and Coefficient of determination ($R^2$) of 0.76 during the calibration periods. For the validation periods, a high model performance result showing $R^2$ of 0.71 and NSE of 0.78 for monthly average streamflow were also obtained"*

RC6:  Page 5: The mean annual rainfall for the watershed is 1224 mm year-1 and the mean annual temperature is about 27o C. Mean annual potential evapotranspiration (PET) estimated by Hargreaves method (Hargreaves and Samani, 1985) is 1720 mm year-1 and the mean AET is about 692 mm year-1. Are this value based on measured or modeled data? Please specify it.

*AC6: Many thanks for highlighting this points for clarification. We have changed the statement in the revised manuscript **in page 6, line 8-12** to read as follows:*

*The mean annual rainfall (1984-2012) obtained from measured data of Ogun watershed is 1224 mm $yr^{-1}$ and the mean annual temperature(1984-2012) obtained from measured data is about $27^o C$. Mean annual potential evapotranspiration (PET) estimated by Hargreaves method (Hargreaves and Samani, 1985) using measured minimum and maximum temperature is 1720 mm $yr^{-1}$ and the mean AET obtained from SWAT output(1989-2012) for this study area is 692 mm $yr^{-1}$.*

RC7: Page 5 typos: is 1224 mm year-1

*AC7: Thank you. We have made the correction **in page 6, line 9** and throughout the manuscript.*

RC8: Page 5: of 4103 ha, and please convert in $km^2$ because all the other areas are in km

*AC8: Thank you. We have made the changes as suggested in the revised manuscript in **page 6, line 21**.*

RC9: 30 m spatial resolution digital elevation model (DEM), 17 soil classes, 17 landuse classes, 3 slope categories, meteorological data and landuse with its management (Table 1). Please specify if those data are available, from which web-site, and the accessed date

*AC9: Thank you. We have added the additional information to **Table 1 in page 28** and in **the references in page 23  line 13 - 14, page 23 line 33 -  34, page 24 line 1 -  2 and page 26 line 27 -  28** in the revised manuscript*

RC10: Page 7 line 10: "The topHRU program allows the identification of a pareto-optimal threshold which minimizes the spatial error to 0.01 ha for a given number of HRUs and thereby minimizes the trade-off between SWAT computation time and number of HRUs. In this case, topHRU determined the optimum number of HRUs to be 1397 for the Ogun River basin. Thresholds of 0 ha for landuse, 150 ha for soil and 250 ha for slope were used in the SWAT set-up". What are the physical consequences of the thresholds? What happens if you use larger or lower values? How you define them?

*AC10: As explained in the manuscript SWAT uses HRUs, whereby a watershed is subdivided into homogenous hydrologic response units having unique soil, slope, and land use properties as the basic unit of all SWAT model calculations. For SWAT, threshold specification of landcover, soil and slope is allowed and the physical consequences of the thresholds is to improve the computational efficiency of simulations while keeping key landscape features and information of the watershed in the hydrologic modelling. In the paper, we selected thresholds of 150 ha for soil and 250 ha for the slope. This means that HRUs should be created for all the area occupied by the landuse classes. For soil it means that any homogenous soil class occupying less than 150 ha should not be considered when determining HRUs. For slope, it means that any homogenous slope class that occupies less than 250 ha should not be considered. This allows us to define how detail the watershed will be represented by selecting the desired threshold values.*

*If we select larger values, then we eliminate key landscape features and their processes out of the system which may lead to considerable loss of information about the watershed landscape, resulting in model output that are less representative of the watershed as a whole and if we select lower values then we retain as many landscape features (spatial data) in the model thereby increasing the computational time of SWAT.*

*Therefore, we defined this threshold by selecting the desired threshold values using topHRU tool and its concept as explained in the paper while minimizing the spatial error to 0.01 ha for a given number*

*of HRUs. Not selecting a threshold for landuse was based on our desire to retain all of the landuse classes for future landuse change research needs.*

*We have explained the process of threshold selection more clearly **in page 9, line 22-26 in** the revised manuscript.*

RC11: Pae 7: "delineated into 53 subbasins, with the main outlet in Abeokuta". Can you please give some summary statistics about them: min max average area, elevation, etc. Daily precipitation 5 data (1984-2012) and minimum and maximum temperature data (1984-2012) at four weather stations (Fig. 1) were used as observed input data. Are you only using 4 stations for the whole basin (20292 km2)? Why not considering satellite products for a variable (precipitation), which sometimes could be even more important than etp? The authors should include this in the discussion. The missing values of daily precipitation and minimum and maximum temperatures, along with solar radiation, wind speed and relative humidity were simulated by the ArcSWAT CSFR_World weather generator: it is clear that the ArcSWAT CSFR_World is used for gap filling of precipitation and temperature. The authors should specify: 1) how did you use the dataset for solar radiation, wind speed and relative humidity? 2) At which time resolutions are you specifying that input? 3) For which hydrological processes did you use these "simulated" forcing variables and how this affects your results?

*AC11: Many thanks for raising this comment for clarification.*

*In SWAT, the Ogun River Basin was delineated into 53 subbasin for this study. The summary statistics of the 53 delineated subbasins is as follows; the minimum and maximum elevation are 23 m and 624 m respectively, while the mean elevation is 289.1 m. The minimum and maximum subbasin area are 72.4 km$^2$ and 853.1 km$^2$ respectively, while the mean is 382.8 km$^2$. We have added this information **in page 9 line 2, 3, 6, and 7** in the manuscript.*

*Daily precipitation data (1984-2012) and minimum and maximum temperature data (1984-2012) obtained from the Nigerian Meteorological Agency for four weather stations (Fig. 1) were used as observed input data. Since the weather stations are more or less evenly distributed in or around the watershed, and the weather data obtained from stations located in the same proximity show the same rise and fall dynamics we were satisfied with the data. No orographic effect correction is needed for correcting the precipitation values. The reasons for using only 4 weather stations in this study have been added **in page 9, line 8 - 11** in the revised manuscript.*

*SWAT requires daily values of solar radiation, relative humidity and wind speed in addition to the daily precipitation, minimum and maximum temperature as weather input in SWAT. One out of many options in SWAT to generate this input variables, is to use "WGEN_CFSR_World (ArcSWAT CSFR_World weather generator)" which is an MS Access file containing long-term monthly weather statistics covering the entire globe. In this study, the long-term monthly weather statistics developed using The National Centres for Environmental Prediction (NCEP) Climate Forecast System Reanalysis (CFSR) global dataset were used to simulate daily solar radiation, relative humidity and wind speed using the WGEN CFSR World. It is also used for filling gaps in measured climate data.*

*The simulated variables were used as input variables into Penman-Monteith and Priestly-Taylor equations for obtaining the different PET estimates from SWAT.*

*The simulated variables allow options for different evaporation estimates which actually affect the results of the model performance during the calibration and validation period as shown in the manuscript. We have added this points **in page 9 line 13 to 18** in the revised manuscript*

RC12: Page7 line22-27: it sounds slightly repetitive: please consider to write the full sentence only of one model structure and to generalize for the other 2.

*AC12: We have made the suggested changes in **page 9, line 29 – 31** and we further explain the meaning of the **acronyms used in figure 2 in page 9, line 30 -32 and page 10, line 1 – 13** (The figure 2 has also been updated to effect the changes **in page 32**) in the revised manuscript to read as follows:*

*The SWAT model was set-up once for the entire Ogun River Basin and then run three times, where each model run is composed of a different PET equation available in SWAT (HG, P-M or P-T). Figure 2 shows the framework in which the three SWAT model runs (SWAT_HG, SWAT_P-T, and SWAT_P-M) were used to evaluate the model performance by:*

*(i)*      *comparing the three uncalibrated SWAT simulations of AET with the two global AET products (GLEAM and MOD16), thus allowing for six reference runs of SWAT (RGS1 through RMS6). SWAT_HG represents the SWAT run using the Hargreaves PET equation to simulate uncalibrated AET, these results were compare with the AET from GLEAM_V3.0a (RGS1) and MOD16 (RMS4). SWAT_P-T represents the SWAT run using the Priestley-Taylor PET equation to simulate uncalibrated AET and the results were compared with the AET from GLEAM_V3.0a (RGS2) and MOD16 (RMS5). SWAT_P-M represents the SWAT run using the Penman-Monteith PET equation to simulate uncalibrated AET and the results were compared with GLEAM_v3.0a (RGS3) and MOD16 (RMS6) and,*

*(ii)*      *(ii) comparing the calibrations/validations implemented with two global AET products (GLEAM and MOD16), thus allowing for six calibration results of SWAT (GS1 through MS6). SWAT_HG represents the SWAT run using the Hargreaves PET equation to simulate AET and that was calibrated and validated with the AET from GLEAM_v3.0a (GS1) and MOD16 (MS4). SWAT_P-T represents the SWAT run using the Priestley-Taylor PET equation to simulate AET and that was calibrated and validated with the AET from GLEAM_v3.0a (GS2) and MOD16 (MS5). SWAT_P- M represents the SWAT run using the Penman-Monteith PET equation to simulate AET and that was calibrated and validated with the AET from GLEAM_v3.0a (GS3) and MOD16 (MS6).*

*This procedure enabled the SWAT model run with the highest performing simulated AET to be chosen for further.*

RC13: Page 8 line 10: please explain what are the main difference s between the two dataset GLEAM_v3.0a and GLEAM_v3.0b and justify why you selected one of the two.

*AC13: Many thanks for raising this comment for more clarification.*

*The two datasets differ in their forcing variables and their temporal coverage (Martens et al., 2016).*

*GLEAM_v3.0a is a global dataset that is based on reanalysis net radiation and air temperature, satellite-based vegetation optical length, and a combination of gauge-based, reanalysis satellite-based precipitation. It is a dataset spanning the 35-year period 1980-2014. For this study, we preferred and selected GLEAM_v3.0a dataset spanning 24-year period 1984-2012 because of its long-term availability that allows reasonably selection and splitting of calibration and validation periods that are not substantially different in climatic condition i.e., wet, moderate, and dry years occur in both periods and which covers our SWAT simulation output period (1989-2012).*

*GLEAM_v3.0b is a global dataset driven by satellite data only and spanning 13-year period 2003-2015. We considered this dataset for the verification of SWAT simulated AET because there are no ground truth AET data in the study area and, because of its different forcing variable, which categories it as an independent dataset not considered in the calibration and validation period.*

*We have clarified the differences between GLEAM_v3.0a and GLEAM_v3.**0b in page 5, line 3 - 7** and justified why we selected one for model calibration/validation **in page 12, line 28 - 33** and the other for model verification **in page 14, line 19 - 21** in the revised manuscript.*

RC14: Page 8 line 25: "was implemented in SWAT-CUP. SWAT-CUP (Abbaspour, 2015)" move the citation when you firstly introduce SWAT-CUP.

*AC14: Thank you. We have made the change in **page 12, line 3** as suggested.*

RC15: Pages 8 line 28 to page 9 line 6: Please specify the parameter set that you started the sensitivity analysis with, at least the processes to which they are related. Moreover, specify the list of the parameters that resulted sensitive and how you define a parameter as "sensitive".

*AC15: We started the global sensitivity analysis for each of the six-calibration run with the same 50 parameters as shown in the table below. The table below only represent the result of the GS1 parameter sensitivity analysis but the same 50 parameters were used for GS2 through MS6 sensitivity analysis.*

*In this study as described in the manuscript, the parameter sensitivity is determined by numerous rounds of Latin Hypercube sampling and we defined a parameter as being sensitive when (considering the absolute values) large values of t-stat and smaller values of p-value were obtained, then, the more sensitive the parameter.*

*The processes to which the parameters are related hydrological are runoff, evaporation, interception, transpiration. In short, each of the hydrological process are represented in the 50 parameters we started the initial global sensitivity with, as shown in table 1 below.*

*Table 1: The 50 parameters consider in the initial global sensitivity analysis and their relative significance. The table below only show the global sensitivity analysis result of the GS1*

| Parameter Name | t-Stat | P-Value |
|---|---|---|
| 45:V__BMX_TREES{..}.plant.dat | 0.68 | 0.62 |
| 23:R__SOL_CBN(..).sol | -0.99 | 0.50 |
| 47:V__TMPMX(..).wgn | 1.84 | 0.32 |
| 48:V__PCPMM(..).wgn | -2.27 | 0.26 |
| 35:V__LAI_INIT.mgt | 2.28 | 0.26 |
| 31:V__SURLAG.bsn | -3.03 | 0.20 |
| 28:R__ALPHA_BF_D.gw | 3.21 | 0.19 |
| 19:A__GWQMN.gw | -3.32 | 0.19 |
| 24:R__SOL_ALB(..).sol | -3.82 | 0.16 |
| 37:V__FLOWFR.mgt | -3.86 | 0.16 |
| 40:R__SOL_ZMX.sol | -4.06 | 0.15 |
| 50:R__SOL_Z(..).sol | 4.08 | 0.15 |
| 39:V__TLAPS.sub | 4.27 | 0.15 |
| 15:V__ESCO.hru | -4.35 | 0.14 |
| 27:V__RCHRG_DP.gw | 4.55 | 0.14 |
| 42:V__ALPHA_BF_D.gw | -4.70 | 0.13 |
| 25:R__USLE_K(..).sol | -5.73 | 0.11 |
| 26:V__SOLARAV(..).wgn | -5.96 | 0.11 |

| | | |
|---|---|---|
| 38:V__TDRAIN.mgt | 6.02 | 0.10 |
| 34:V__BIO_MIN.mgt | 6.03 | 0.10 |
| 46:V__TMPMN(..).wgn | -6.18 | 0.10 |
| 3:V__REVAPMN.gw | -6.19 | 0.10 |
| 43:V__RADINC(..).sub | -6.20 | 0.10 |
| 36:V__BIO_INIT.mgt | 6.25 | 0.10 |
| 13:V__EVRCH.bsn | -6.91 | 0.09 |
| 20:R__HRU_SLP.hru | 6.94 | 0.09 |
| 21:V__GW_DELAY.gw | 6.96 | 0.09 |
| 14:V__GW_REVAP.gw | -7.01 | 0.09 |
| 44:V__HUMINC(..).sub | -7.22 | 0.09 |
| 29:V__SHALLST.gw | -7.34 | 0.09 |
| 16:V__CH_N2.rte | 7.59 | 0.08 |
| 22:V__ALPHA_BNK.rte | 7.66 | 0.08 |
| 17:V__CH_K2.rte | -7.66 | 0.08 |
| 41:V__CH_N1.sub | 7.70 | 0.08 |
| 49:V__DEEPST.gw | 7.98 | 0.08 |
| 12:V__EVLAI.bsn | 8.03 | 0.08 |
| 11:V__OV_N.hru | 8.12 | 0.08 |
| 18:V__SFTMP.bsn | 8.49 | 0.07 |
| 33:V__GWHT.gw | 8.74 | 0.07 |
| 10:V__FFCB.bsn | -8.78 | 0.07 |
| 7:R__SOL_BD(..).sol | -8.91 | 0.07 |
| 6:V__EVRSV.res | -8.95 | 0.07 |
| 32:V__GSI{..}.plant.dat | -9.78 | 0.06 |
| 2:V__EPCO.hru | 10.14 | 0.06 |
| 4:R__SOL_K(..).sol | 10.51 | 0.06 |
| 9:V__ALPHA_BF.gw | -10.52 | 0.06 |
| 1:V__ESCO.hru | -10.71 | 0.06 |
| 30:V__CANMX.hru | 11.52 | 0.06 |
| 5:R__SOL_AWC(..).sol | 12.37 | 0.05 |
| 8:R__CN2.mgt | 15.79 | 0.04 |

*After choosing the 11 most sensitive parameters based on the t-stat and p-values. We tried many combinations of the most sensitive analysis e.g. we started with P < 0.09 (37 parameters) the result was not as good at that of the 11 parameters combination. So, we decided to take the 11 most sensitive parameters and we run another global sensitivity analysis to further identify the relative significance of each parameter before calibration. Using 11 parameters gave the most reasonable results. This methodology was extended to all the remaining calibration runs.*

*We have further clarified the number of parameters we started with, the process to which they are related and how we define the parameter sensitive in the revised manuscript on page **in page12, line 11 - 19.***

*Table 2: The 11 most sensitive parameters consider in the final calibration and validation of all the six calibration runs. The table below only show the sensitivity analysis of the GS1 calibration results.*

| Parameter Name | t-Stat | P-Value |
|---|---|---|
| 9:R__SOL_AWC(..).sol | 0.01 | 0.99 |
| 2:V__EPCO.hru | -0.21 | 0.83 |
| 7:V__FFCB.bsn | -0.27 | 0.79 |
| 4:V__GSI{..}.plant.dat | 0.40 | 0.69 |
| 6:V__EVRSV.res | -0.54 | 0.59 |
| 10:R__SOL_K(..).sol | -0.89 | 0.37 |
| 5:V__ALPHA_BF.gw | 1.43 | 0.15 |
| 11:R__SOL_BD(..).sol | -2.01 | 0.04 |
| 3:V__CANMX.hru | 2.19 | 0.03 |
| 1:V__ESCO.hru | -2.86 | 0.00 |
| 8:R__CN2.mgt | -23.93 | 0.00 |

*RC16:* Page 9 line 16: A metric among the six can be considered an objective function if it is optimized in the calibration procedure; it can be considered as goodness of fit metric if it is used to quantify how well or bad the model reproduces the measured data. Are those goodness of fit metrics? Which one of these six metrics has been optimized in the calibration procedure? Have you used all of them also as objective function? This is not fully clear.

*AC16: Many thanks for raising this comment for more clarification. The Nash-Sutcliffe is the selected objective function that was optimized during the calibration process. This statement has been added for clarity **in page 14, line 2** in the revised manuscript.*

*RC17  Page* 9 line 20 –Page 10 line 20: Consider to: i) just spell in the text the statistics used, their ranges and their optimal values and ii) move in appendix the explanation of each statistics because they are well known.

*AC17: Thank you. We have made the changes as suggested in **page 13, line 28 – 34 and page 14, line 1**. The equations and their description has been moved to  **Appendix A in page 43.***

*RC18:* Page 10 line 23-26: please specify how the uncertainty is quantified: what are the parameters that are changed/sampled the LHS, what are their ranges?

*AC18: Thank you for highlighting this point for further clarification. We have emphasized on how the uncertainty was quantified in **page 14, line 7 – 13.***

*As described in the manuscript the uncertainty was quantified by: 1.) assessing the percentage of GLEAM_3.0a AET data bracketed by the model output 95% predictive uncertainty band, the index used for the quantification is P-factor, and 2.) assessing the ratio of the average width of the 95ppu and the standard deviation of the GLEAM_3.0a, the index used for the quantification is R-factor.*

*The parameters that are changed are listed in Table 2 above by implementing numerous rounds of Latin Hypercube sampling. The optimum values are presented in the revised manuscript **in Table 2 (page 29)**. In this study, the ranges in which we started the calibration with are presented in **Appendix B in the revised manuscript in page 44.***

*Table 3a: Eleven parameters and their minimum and maximum range used in this study for the 1st iteration (1000 simulations) for all the six calibration runs.*

| Parameter name | Minimum range | Maximum range |
|---|---:|---:|
| v__ESCO.hru | 0.00 | 1.00 |
| v__EPCO.hru | 0.00 | 1.00 |
| v__CANMX.hru | 0.00 | 100.00 |
| v__GSI{2,4,5}.plant.dat | 0.00 | 5.00 |
| v__ALPHA_BF.gw | 0.00 | 1.00 |
| v__EVRSV.res________17,50 | 0.00 | 1.00 |
| v__FFCB.bsn | 0.00 | 1.00 |
| r__CN2.mgt | -0.25 | 0.85 |
| r__SOL_AWC().sol | 0.23 | 0.95 |
| r__SOL_K().sol | -0.06 | 0.95 |
| r__SOL_BD().sol | -0.41 | 0.95 |

*RC19:* Page 11: please show a figure of the river basin with the subbasin polygons and the pixel of MODIS and GLEAM. This will help the reader to understand how many pixels of MODIS and GLEAM cover your basins. Each sub-basins has its own model AET. How did you choose the MODIS or GLEAM pixel to compare with and compute the NSE, R², etc.

*AC19: Thanks for the suggestion, we prepared a figure showing the river basin with the subbasin polygon and the pixel of MOD16 and GLEAM AET. We have inserted the figures as suggested in* **Appendix D in page 48.**

*To compare MODIS pixel value (Fig.1) to SWAT simulated AET values from each subbasin for computing the NSE, R², PBIAS, KGE, an area-weighted averaging scheme was performed in ArcGIS to create aggregated monthly time-series of AET data of each subbas*

[Figure]

**MOD16 AET (1km)**

MOD16AET_intersect_ogun53_subbasin

• Polygon_to_Points_2000-20012

*Figure 1: Ogun River Basin with its 53-subbasin polygons intersected by the pixel of MOD16 AET.*

*The GLEAM_3.0a and GLEAM_3.0b AET is provided in netcdf format, with one file per year and variable. The datasets are available on a $0.25^0$ latitude- longitude regular grid and at daily temporal resolution.*

[Figure]

*Figure 2: Ogun River Basin with its 53-subbasin polygons intersected by the pixel of GLEAM AET*

*We used "make NetCDF raster layer" tool in ArcGIS to convert the NetCDF file into a raster layer to view how many pixels of GLEAM cover our subbasins (Fig 2). We realized some points from which the data will be extracted from the pixel are not located in some subbasins (Fig.2). Therefore, we decided to create a point at the center of each subbasin in ArcGIS (Fig. 3). The coordinates of each point at the center of the subbasins were obtained.*

[Figure]

*Figure 3: Ogun River Basin with its 53-subbasin polygons intersected by the pixel of GLEAM AET with a point at the center of each subbasin.*

*To enable us to compare GLEAM pixel value to SWAT simulated AET values for each subbasin for computing the NSE, $R^2$, PBIAS, KGE, we went back to the NetCDF file of the daily time series (1989-2012) and extracted the GLEAM AET value for each subbasin using "Make NetCDF table view. The values extracted were aggregated to monthly values. The same procedure was carried out for GLEAM_3.0b version. All the clarification has been added in the revised manuscript **in page 12, line 22 - 28 and page 13, line 3 – 6.***

*RC20:* Page 11: figure 3, 4, 5, and 6 please consider to add a 4th class for the range of KGE and NSE (<0). This indicates where just the mean of the observed data will be more performing than the model itself

*AC20: Many thanks for the suggestion that needs further clarification.*

*We divided the results into 5 classes. Wish we think the performance rating classes represented with figure 3,4,5,6 for each subbasins are well represented and sufficient*

*-6.0 – 0 = red*

*0.10- - 0.50 = purple*

*0.51 – 0.60 = yellow*

*0.61-0.70 = orange*

*0.71 -1.0 = green*

*The KGE and NSE figure that do not contain a class -6.0 – 0 (<0) do not contain values <0 (fig.3, 4 in the manuscript) and figures that contains -6.0 – 0 (<0) has a model performance that is less than 0 (fig. 5 and fig.6 in the manuscript)*

*We do not think is necessary to include a class in the legend without having its value in the subbasin polygon.*

*RC21:* Page 10 line 15: equation is not correct please revise it.

*AC21: Many thanks for the point. We have made the changes in the revised manuscript in page 43: KGE.*

*RC22:* Page 11: figure 3, 4, 5, and 6 please add the North symbol and the scale bar in the maps.

*AC22: Many thanks for the suggestion. We have included North symbol and the scale bar in the revised manuscript **in page 34 – 37.***

*RC23:* Page 11 line 23: "The results of global sensitivity analysis revealed that the SCS runoff": please specify from where the reader can see this.

AC23: Thank you. We have included Table 2 at the end of the statement in **page 17, line 17** in the manuscript

*RC24:* Page 12: all the page can be summarized by just one figure reporting on the x-axis the model configurations and on the y axis the percentage of sub-basin for a given class of the goodness of fit index (NSE, R$^2$, etc).

AC24: Many thanks for the suggestion.  The results presented in page 12 are already summarized in Figures 3, 4, 5,6,7,8 and 9 for clarity. We believe having figures that summarize well the calibration/validation results of the Ogun River Basin 53 subbasins and that of the whole catchment is necessary for detailed information. We believe having just one figure to report all the results of the calibration/validation procedure might not be sufficient in this case.

 To this effect, we have reduced the length of the result presentation described in page **15 and 16** because the figures already shown them.

*RC25:* Page 13 the paper goes from section 3 to subsection 3.3: 3.1 and 3.2 are missing.

AC25: Thank you for highlighting this point. We have included the subsection 3.1 and 3.2 in the revised manuscript in **page 16, line 9 and page 16, line 22**

*RC26:* Page 14: increases in February from 55mm to 76mm as the space between 55 and mm

AC26: Thank you for highlighting this point. We have made the corrections in the revised manuscript **in page 16, line 27**.

*RC27:* Page 15 line 5: "Using the guidelines in Moriasi et al. (2007, 2015) and Kouchi et al. (2017) for" probably these guidelines were drawn for runoff? Is it correct to use it for others hydrological processes? Is this been done in the past? If yes, please add a citation otherwise just clarify this aspect.

AC27: *Many thanks for raising this point for further clarification. The general hydrologic model performance ratings for recommended statistics (NSE, PBIAS, R$^2$) performed at a monthly time and recommended by Moriasi et al. (2007, 2015) and Gupta et al are mostly relevant for runoff, sediment and nutrients because when these articles were published, the use of remotely sensed evapotranspiration datasets for hydrologic model calibration/validation has not gained much ground.*

*In this paper, we also conducted reviewed literatures on model evaluation methods and ratings for model calibration using satellite or non-satellite derived evapotranspiration.  Ha et al. (2018) presented a study of calibration of spatially distributed hydrological processes and model parameters in SWAT using remote sensing data and an autocalibration procedure. NSE, R$^2$, and KGE criteria were used to assess the model performances.*

*Djman (2016) in their study of evaluation, calibration and validation of six reference ET$_0$ equation for Senegal River Delta using Penman-Monteith derived ET$_0$ obtained at saint louis station (1960-2012), uses R$^2$ and other statistical measures to perform the evaluation. Lopez et al. (2017), calibrated a large-scale hydrological model using satellite-based soil moisture and evapotranspiration products. They evaluated the model performance using NSE, PBIAS, KGE and R. Samadi et al. (2016) presented a study on assessing the sensitivity of SWAT physical parameters to potential evapotranspiration estimation*

*methods over a coastal plain watershed in the southeastern United States, NSE, and KGE statistical measure were used for the model performance assessment.*

*All the reviewed literatures set their performance ratings for recommended statistics (NSE, PBIAS, R, KGE, R²) based on Moriasi et al. (2007, 2015) and kouchi et al. (2017) guidelines.*

*In this study, we follow Lopez et al. (2017) and others as reviewed above to base our performance rating criteria for judging the model performance by using NSE, R², PBIAS and KGE. These points have added in the revised manuscript in page 13, line 23 - 31.*

*RC28:* Page 15 line 20: "From our results, we agree that the AET from MOD16 tends to overestimate AET". Overestimate against what? This is a strong statement mainly because there is not direct comparison against measured AET data

*AC28: Thank you for highlighting this point. We agree this is a strong statement that needs to be revised. Actually, we meant that AET from MOD 16 values are higher than that of GLEAM AET and SWAT simulated AET and that this finding agrees with other studies carried out in tropical regions. Since we are using the satellite based MOD16 to calibrate the SWAT AET simulation, the statement needs correction and we have made the corrections in page 18, line 24 – 34 and page 19, line 1 – 11.*

[revised manuscript text omitted]

---

## Author Response (AR2)

**Response to the Editor for hess-2018-170**

Dear Editor, thank you very much for the opportunity to resubmit a revised copy of our paper entitled 'Multi-site calibration and validation of SWAT with satellite-based evapotranspiration in a data sparse catchment in southwestern Nigeria'. The constructive comments and suggestions offered by the reviewers have been immensely helpful. We greatly appreciate their insightful comments on revising the paper. The manuscript has been revised again to address the reviewer's concerns and a point-by-point reply to the reviewer's comments has been made. The changes arising from the comments have clearly improved our manuscript. The new version of the manuscript is uploaded alongside this document. As requested, all the modifications are highlighted (**in Turquoise**) in the manuscript.. We look forward to hearing from you in due time regarding our submission and to respond to any further questions and comments you may have.

**Response to the Reviewers Comments for hess-2018-170**

We thank the anonymous referees #1 and #2 for reviewing our manuscript. We are especially grateful for the many insightful and constructive comments and their valuable suggestions; these changes have clearly improved the quality of the manuscript. We have to the best of our abilities responded to them and address the referees' comments in the following point by point response. Note the following conventions: RC = referee comments, AC = authors comments (replies) printed in italic. All the modifications are highlighted **in Turquoise** in the revised manuscript.

**Reply to comments of Anonymous Referee #1**

**RC 1:** Precise what satellite data is used by GLEAM (I think there is also surface soil moisture from scaterrometer data)

*AC1: The Global Land-surface Evaporation Amsterdam (GLEAM) combines a wide range of remote sensing observations from **different satellites** to separately estimate the different components of terrestrial evaporation and surface soil moisture through a process-based methodology at a global scale and 0.25-degree spatial resolution (Martens et al., 2017).*

*This additional statement of GLEAM using a process-based methodology to estimate land-surface evaporation from multi-satellite information is included in page 10 line 15-27 in the revised manuscript as suggested but for detail information on the different satellite data used by GLEAM the reader is referred to Martens et al., 2017 in page 11 lines 15-17 of the revised manuscript.*

*The different satellite data used by GLEAM is briefly stated below for GLEAMv3.0a and v3.0b used in this study.*

*GLEAMv3.0a (used for AET calibration and validation in this paper)*

 a. *Radiation fluxes from the current reanalysis of the European Centre for Medium-Range Weather Forecasts (ECMWF), ERA-Interim (Dee et al., 2011), were processed and used.*
 b. *For the precipitation forcing, the Multi-Source Weighted Ensemble Precipitation (MSWEP) data set was used. MSWEP is based on a merger of selected satellite-, reanalysis-, and gauge-based products.*
 c. *Air temperature estimates from ERA-Interim are also selected for the long-term GLEAM data set (GLEAMv3.0a).*
 d. *The phenological controls on transpiration are derived from observations of microwave VOD (Vegetative Optical Depth). The 0.25∘ product from Liu et al. (2011) was used, which is based on retrievals from several passive microwave sensors using the Land Parameter Retrieval Model (LPRM, Owe et al. (2008)). To cover the period 1980–2015, it was merged with LPRM-based VOD retrievals from SMOS (van der Schalie et al., 2015, 2016) using a similar cumulative distribution functions (CDF) matching approach to the one used by Liu et al. (2011).*
 e. *For the assimilation of microwave surface soil moisture, the SMOS Level 3 soil moisture product (Jacquette et al., 2010) and the ESA Climate Change Initiative soil moisture (ESA CCI SM v2.3) data set (Liu et al., 2012; Wagner et al., 2012) were selected. The ESA CCI SM v2.3 data is a blended product of soil moisture retrievals from several active and passive microwave sensors, available for the period 1978–2015 at the global scale.*

*GLEAMv3.0b (used for AET verification in this paper)*

   a. *Radiation inputs are based on measurements from the Clouds and Earth's Radiant Energy System (CERES) onboard Terra and Aqua (Wielicki, 1996).*
   b. *For the precipitation forcing, the Tropical Rainfall Measurement Mission (TRMM) Multi-satellite Precipitation Analysis (TMPA) 3B42v7 product was used.*
   c. *Air temperatures are derived from measurements of the Atmospheric Infrared Sounder (AIRS, Aumann et al., 2003).*
   d. *The phenological controls on transpiration are derived from observations of microwave VOD (Vegetative Optical Depth). The 0.25∘ product from Liu et al. (2011) was used, which is based on retrievals from several passive microwave sensors using the Land Parameter Retrieval Model (LPRM, Owe et al. (2008)). To cover the period 1980–2015, it was merged with LPRM-based VOD retrievals from SMOS (van der Schalie et al., 2015, 2016) using a similar cumulative distribution functions (CDF) matching approach to the one used by Liu et al. (2011).*
   e. *For the assimilation of microwave surface soil moisture, the ESA Climate Change Initiative soil moisture (ESA CCI SM v2.3) data set (Liu et al., 2012; Wagner et al., 2012) were selected. The ESA CCI SM v2.3 data is a blended product of soil moisture retrievals from several active and passive microwave sensors, available for the period 1978–2015 at the global scale.*

**Table 1** *provides summary information on the forcing variables used to produce the GLEAMv3.0a and v3.0b datasets together with type of satellite, reanalysis and gauge-based dataset used by GLEAM.*

| GLEAM Data set | Variable | Data set | Type |
|---|---|---|---|
| v3a | Radiation | ERA-Interim | Reanalysis |
| v3b | | CERES L3 SYN1deg | Satellite |
| v3a | Air Temperature | ERA-Interim | Reanalysis |
| v3b | | AIRS L3 RetStd v6.0 | Satellite |
| v3a | Precipitation | MSWEP v1.0 | Merge |
| v3b | | TRMM 3B42 | Merge |
| v3a | Snow Water Equivalents | GLOBSNOW L3Av2 & NSIDC v01 | Satellite |
| v3b | | GLOBSNOW L3Av2 & NSIDC v01 | Satellite |
| v3a | Vegetation Optical Depth | LPRM(MERGE) | Satellite |
| v3b | | LPRM(MERGE) | Satellite |
| v3a | Surface Soil Moisture | ESA-CC1v2.3 | Satellite |
| v3b | | ESA-CC1v2.3 | Satellite |

*Source: Martens et al.2017*

**Reply to comments of Anonymous Referee #2**

**Major comments**

**RC1**: On the authors reply to reviewer 2 comment C: If it is completely understandable that runoff measurements are not available, how about assess the model calibrated on etp using soil moisture data (potentially satellite retrieved)? All the SWAT applications studies the authors used in the reference have verified their simulations against streamflow measurement. Please could you refer to studies that used hydrological models forced by GLEAM and do not account for runoff or soil moisture in the validation process? I would at least point out the conclusion that this step (of independent model verification) is missing and need further developments. This has consequences on the quantification of the water balance discussed in the paper.

*AC1: Many thanks for the suggestion. We have included the validation of our model calibrated on etp using ESA ICC soil moisture data (potentially satellite retrieved) as suggested in the revised manuscript (e.g, page 12 lines 4-19, page 15 lines 21-31 and page 44 figure 13-14)*

**Minor comments**

RC1: pag 14 lines 22-24: could you please specify the units of the terms in the equation 7 and 8.

*AC1: Thank you! We have specified the units of the terms in the equation 7 and 8 in page 15 lines 14-19*

RC2: pag 17 line 20-32: a 5 lines sentence probably is too long please consider to split it in 2.

*AC2: We agree with the referee and we have made the changes in page 18 lines 24-31.*

RC3: Figures 3 to 6: looking at the figure as it is now the reader cannot understand that classes of the NSE or KGE are 5. For example figure 5 NSE: the legend shows three classes of which the last is NSE <= 0.5 which can include also NSE<0 (at it is). The same olds for the KGE (KGE<0.5 includes KGE<0). Please consider to show in the legend all the classes and all the colors you have used for each of the indicators. For Pbias, what happens to the class >+-25?

*AC3: We agree with the referee and we have made the changes in Figures 4 to 7 to show in the pie chart legend all the classes and all the colours we have used for each of the indicators of Ogun River subbasins performance in page 35 to 38 as suggested. The Pbias for GS1 and MS6 in both calibration and validation period do not exceed ±20. The legend has been reset to reflect c(-20,0,+20).*

RC4: On the authors reply to reviewer 2 comment 19: Please report the way in which you compared the results in the main text and then refer to the appendix D for the figures, now there is no reference in the main text. The background of the figures in appendix D is probably too dark especially for the gleam application. What the grey shadow means, it is not specified in the legend.

*AC4: We agree with the referee. We have reported how we compared the results of our model calibration using GLEAM AET and MOD16 AET and we have referred to the appendix D figures in the main text as suggested in page 13 lines 12-13 and page 15 line 26. The background of the figure in appendix D has been changed and the mean monthly GLEAM (1989-2012) and MOD 16 AET (2000-2012) values has been added in the legend in page 50.*

**Reference**

Dee, D. P., Uppala, S. M., Simmons, A. J., Berrisford, P., Poli, P., Kobayashi, S., Andrae, U., Balmaseda, M. A., Balsamo, G., Bauer, P., Bechtold, P., Beljaars, A. C. M., van de Berg, L., Bidlot, J., Bormann, N., Delsol, C., Dragani, R., Fuentes, M., Geer, A. J., Haimberger, L., Healy, S. B., Hersbach, H., Hólm, E. V., Isaksen, L., Kallberg, P., Köhler, M., Matricardi, M., McNally, A. P., Monge-Sanz, B. M., Morcrette, J.-J., Park, B.-K., Peubey, C.,deRosnay,P.,Tavolato,C.,Thépaut,J.-N.,andVitart,F.:The ERA-Interim reanalysis: configuration and performance of the data assimilation system, Q. J. Roy. Meteor. Soc., 137, 553–597, doi:10.1002/qj.828, 2011.

Liu, Y. Y., de Jeu, R. A. M., McCabe, M. F., Evans, J. P., and van Dijk, A. I. J. M.: Global long-term passive microwave satellitebased retrievals of vegetation optical depth, Geophys. Res. Lett., 38, L18402, doi:10.1029/2011GL048684, 2011.

Liu, Y. Y., Dorigo, W. A., Parinussa, R. M., de Jeu, R. A. M., Wagner, W., McCabe, M. F., Evans, J. P., and van Dijk, A. I. J. M.: Trend-preserving blending of passive and active microwave soil moisture

retrievals, Remote Sens. Environ. 123, 280–297, doi:10.1016/j.rse.2012.03.014, 2012.

Martens, B., Miralles, D. G., Lievens, H., van der Schalie, R., de Jeu, R. A. M., Fernández-Prieto, D., Beck, H. E., Dorigo, W. A., and Verhoest, N. E. C.: GLEAM v3: satellite-based land evaporation and root-zone soil moisture, Geosci. Model Dev., 10, 1903-1925, https://doi.org/10.5194/gmd-10-1903-2017, 2017.

Owe, M., de Jeu, R., and Holmes, T.: Multisensor historical climatology of satellite-derived global land surface moisture, J. Geophys. Res.-Earth, 113, F01002, doi:10.1029/2007JF000769, 2008.

van der Schalie, R., Parinussa, R., Renzullo, L. J., van Dijk, A. I. J. M., Su, C.-H., and de Jeu, R. A. M.: SMOS soil moisture retrievals using the Land Parameter Retrieval Model: Evaluation over the Murrumbidgee catchment, southeast Australia, RemoteSens.Environ.,163,70–79,doi:10.1016/j.rse.2015.03.006, 2015.

van der Schalie, R., Kerr, Y. H., Wigneron, J.-P., RodríguezFernández, N. J., Al-Yaari, A., and de Jeu, R. A. M.: Global SMOS soil moisture retrievals from The Land Parameter Retrieval Model, Int. J. Appl. Earth Obs., 45, 125–134, doi:10.1016/j.jag.2015.08.005, 2016

Wagner, W., Dorigo,W.A.,de Jeu, R.A. M.,Fernández-Prieto,D., Benveniste, J., Haas, E., and Ertl, M.: Fusion of active and passive microwave observations to create an essential climate variable data record on soil moisture, in: Proceedings of the XXII International Society for Photogrammetry and Remote Sensing (ISPRS) Congress, Melbourne, Australia, vol. 25, 2012.

Wielicki, B. A.: Clouds and the Earth's Radiant Energy System (CERES): An earth observing system experiment, B. Am. Meteorol. Soc., 77, 853–868, doi:10.1175/15200477(1996)077<0853:CATERE>2.0.CO;2, 1996.